# A VAE-based Framework for Learning Multi-Level Neural Granger-Causal Connectivity

**Jiahe Lin**
*Machine Learning Research, Morgan Stanley*

**Huitian Lei**
*Lyft, Inc.*

**George Michailidis** *⋆*
*Department of Statistics and Data Science*
*University of California, Los Angeles*

**Reviewed on OpenReview:** *https://openreview.net/forum?id=kNCZ95mw7N*

## Abstract

Granger causality has been widely used in various application domains to capture lead-lag relationships amongst the components of complex dynamical systems, and the focus in extant literature has been on a single dynamical system. In certain applications in macroeconomics and neuroscience, one has access to data from a collection of *related* such systems, wherein the modeling task of interest is to extract the shared common structure that is embedded across them, as well as to identify the idiosyncrasies within individual ones. This paper introduces a Variational Autoencoder (VAE) based framework that *jointly* learns Granger-causal relationships amongst components in a collection of related-yet-heterogeneous dynamical systems, and handles the aforementioned task in a principled way. The performance of the proposed framework is evaluated on several synthetic data settings and benchmarked against existing approaches designed for individual system learning. The method is further illustrated on a real dataset involving time series data from a neurophysiological experiment and produces interpretable results.

## 1 Introduction

The concept of Granger causality introduced in Granger (1969) leverages the temporal ordering of time series data. It is defined in terms of predictability of future values of a time series; namely, whether the inclusion of past information (lag values) of other time series as well as its own (self lags) leads to a reduction in the variance of the prediction error of the time series under consideration. Since its introduction, it has become a widely-used approach in the analysis of economic (Stock & Watson, 2001), financial (Hong et al., 2009) and neuroimaging (Seth et al., 2015) time series data. The standard setting in these applications is that one is interested in estimating Granger causal relationships in a dynamical system (e.g., a national economy, a brain) comprising of $p$ variables.

Granger causality can also be expressed through the language of graphical models (Dahlhaus & Eichler, 2003; Eichler, 2012). The node set of the graph corresponds to the $p$ variables at different time points; *directed* edges between nodes at past time points to those at the present time capture Granger causal relationships (for more details and a pictorial illustration see Section C.1). Traditionally, Granger causality was operationalized through linear vector autoregressive (VAR) models (Granger, 1969), in which case the entries of the estimated transition matrices correspond precisely to the edges of the Granger causal graph.

---

⋆ Corresponding Author. ⟨gmichail@ucla.edu⟩

More recent work has explored how Granger causal relationships can be learned through nonlinear models; e.g., see review paper Shojaie & Fox (2022) and references therein.

In certain application domains, one has access to data from a collection of *related* dynamical systems. A motivating example is described next. Consider electroencephalography (EEG) recordings obtained from $p$ electrodes placed on the scalp of a subject (e.g., a patient or an animal). The resulting time series data constitute measurements from a complex neurophysiological dynamical system (Stam, 2005). On many instances, one has access to such measurements for a collection of *M related* subjects (or "entities", equivalently); for example, they may be performing the same cognitive task (e.g., visual counting, geometric figure rotation) or exhibit a similar neurological disorder (e.g., epilepsy, insomnia, dementia). In such a setting, one can always opt to perform separate analyses on each subject's data; however, it would be useful to develop methodology that models the data from all subjects *jointly*, so as to simultaneously extract the embedded structure shared across subjects and identify the idiosyncrasies (heterogeneity) in any single one. In other words, if one views all subjects as belonging to a common group, the quantities of interest are the shared group-level connectivity structure (amongst nodes) and the entity-level ones.

Conceptually, the above-mentioned modeling task is not difficult to accomplish in a linear setting where one can decompose the transition matrices into a "shared" component and an idiosyncratic (entity-specific) one, with some orthogonality-type constraint to enforce identifiability of the parameters (more details provided in Section C.3). However, the task becomes more challenging and involved in non-linear settings where one hopes to use flexible models to capture the underlying complex dynamics. In particular, a decomposition-based approach, which requires the exact specification of the functional form of the shared component or how the shared and the idiosyncratic components interact, would be rather restrictive. To this end, we adopt a generative model-based approach, which circumvents the issue by encoding the Granger causal relationships through graphs. By postulating a model with a hierarchical structure between the shared and entity-specific components, the problem can be addressed in a flexible, yet principled manner.

**Summary of contributions.** We develop a two-layer Variational Autoencoder (VAE) based framework for estimating Granger-causal connections amongst nodes in a collection of related dynamical systems — jointly for the common group-level and the entity-level ones — in the presence of entity-specific heterogeneity. Depending on the assumed connection type (continuous or binary) amongst the nodes, the proposed framework can accommodate the scenario accordingly by imposing a commensurate structure on the encoded/decoded distributions, leveraging conjugacy between pairs of distributions. The proposed model enables extracting the embedded common structure in a principled way, without resorting to any ad-hoc or post-hoc aggregation. Finally, the framework can be generalized to the case where multiple levels of nested groups are present and provides estimates of the group-level connectivity for all levels of groups.

The remainder of the paper is organized as follows. In Section 2, we provide a review of related literature on Granger-causality estimation, with an emphasis on neural network-based methods. The main building block used in the proposed framework, namely, a multi-layer VAE is also briefly introduced. Section 3 describes in detail the proposed framework, including the encoder/decoder modules and the training/inference procedure. In Section 4, model performance is assessed on synthetic datasets and benchmarked against several existing methods. An application to a real dataset involving EEG signals from 22 subjects is discussed in Section 5. Finally, Section 6 concludes the paper.

## 2 Related Work and Preliminaries

In this section, we review related work on inferring Granger causality based on time series data, with an emphasis on deep neural network-based approaches. Further, as the proposed framework relies on variational autoencoders (VAE) with a hierarchical structure, we also briefly review VAEs in the presence of multiple latent layers.

## 2.1 Inference of Granger causality

Linear VAR models have historically been the most popular approach for identifying Granger causal relationships. Within the linear setting, hypothesis testing frameworks with theoretical guarantees have been developed (Granger, 1980; Geweke, 1984), while more recently regularized approaches have enabled the estimation in the high-dimensional setting (Basu et al., 2015). Recent advances in neural network techniques have facilitated capturing non-linear dynamics and identifying Granger causality accordingly, as discussed next.

Note that estimation of Granger causality is an *unsupervised* task, in the sense that the connectivity as captured by the underlying graph is *not observed* and thus cannot serve as the supervised learning target. Depending on the model family that the associated estimation procedure falls into, existing approaches suitable for estimating Granger causality based on neural networks (Montalto et al., 2015; Nauta et al., 2019; Wu et al., 2020; Khanna & Tan, 2020; Tank et al., 2021; Marcinkevičs & Vogt, 2021; Löwe et al., 2022) can be broadly categorized into prediction-based and generative model-based ones. We selectively review some of them next. In the remainder of this subsection, we use $x_{i,t}$ to denote the value of node $i$ at time $t$, $\mathbf{x}_t := (x_{1,t}, \cdots, x_{p,t})$ the collection of node values of the dynamical system, and $\mathbf{x} := \{\mathbf{x}_1, \cdots, \mathbf{x}_T\}$ the trajectory over time.

Within the predictive modeling framework, recent representative works include Khanna & Tan (2020); Tank et al. (2021); Marcinkevičs & Vogt (2021), where the Granger-causal relationship is inferred from coefficients that govern the dynamics of the time series, and the coefficients are learned by formulating prediction tasks that can be generically represented as $\mathbf{x}_t = f(\mathbf{x}_{t-1}, ..., \mathbf{x}_{t-q}) + \boldsymbol{\varepsilon}_t$, with $\mathbf{x}_t \in \mathbb{R}^p$ being the multivariate time series signal and $\boldsymbol{\varepsilon}_t$ the noise term. In Tank et al. (2021), coordinates of the response are considered separately, that is, $x_{i,t} = f_i(\mathbf{x}_{t-1}, ..., \mathbf{x}_{t-q}) + \varepsilon_{i,t}$, and $f_i$ is parameterized using either multi-layer perceptrons (MLP) or LSTM (Hochreiter & Schmidhuber, 1997). In the case of an $L$-layer MLP,

$$\widehat{x}_{i,t} = W^L \mathbf{h}_t^{L-1} + \mathbf{b}^L; \quad \mathbf{h}_t^l = \sigma\Big(W^l \mathbf{h}_t^{l-1} + b^l\Big),\ l = 2, \cdots, L; \quad \mathbf{h}_t^1 = \sigma\Big(\sum_{k=1}^q W^{1k} \mathbf{x}_{t-k} + \mathbf{b}^1\Big);$$

the Granger-causal connection from the $j$th node to the $i$th node is then encoded in some "summary" (e.g., Frobenius norm) of $\{W_{:j}^{11}, \cdots, W_{:j}^{1q}\}$, with each component corresponding to the first hidden layer weight of lags $x_{j,t-1}, \cdots, x_{j,t-q}$. Various regularization schemes are considered and incorporated as penalty terms in the loss function, to encourage sparsity and facilitate the identification of Granger-causal connections. The case of LSTM-based parameterization is handled analogously. Marcinkevičs & Vogt (2021) parameterizes $f$ as an additive function of the lags, i.e., $\mathbf{x}_t = \sum_{k=1}^q \Psi_k(\mathbf{x}_{t-k})\mathbf{x}_{t-k} + \boldsymbol{\varepsilon}_t$; the output of $\Psi_k : \mathbb{R}^p \mapsto \mathbb{R}^{p \times p}$ contains the generalized coefficients of $\mathbf{x}_{t-k}$, whose $(i, j)$ entry corresponds to the impact of $x_{j,t-k}$ on $x_{i,t}$ and $\Psi_k$ is parameterized through MLPs. The Granger causal connection between the $j$th node and the $i$th node is obtained by aggregating information from the coefficients of all lags $\{\Psi_k(\mathbf{x}_{t-k})_{ij}\}$, i.e., $\max_{1 \leq k \leq q}\{\text{median}_{q+1 \leq t \leq T}(|\Psi_k(\mathbf{x}_{t-k})_{ij}|)\}$. Finally, an additional stability-based procedure where the model is fit to the time series in the reverse order is performed for the final selection of the connections.[1] It is worth noting that for both of the above-reviewed approaches, the ultimately desired node $j$ to node $i$ ($\forall\ i, j \in \{1, \cdots, p\}$) Granger causal relationship is depicted by a scalar value, whereas in the modeling stage, such a connection is collectively captured by multiple "intermediate" quantities—$\{W_{:j}^{1k}, k = 1, \cdots q\}$ in Tank et al. (2021) and $\{\Psi_k(\mathbf{x}_{t-k})_{ij}, k = 1, \cdots, q\}$ in Marcinkevičs & Vogt (2021); hence, an information aggregation step becomes necessary to summarize the above-mentioned quantities to a single scalar value.

For generative model-based approaches, the starting point is slightly different. Notable ones include Löwe et al. (2022) that builds upon Kipf et al. (2018), and the focus is on *relational inference*. The postulated generative model assumes that the trajectories are collectively governed by an underlying latent graph $\mathbf{z}$, which effectively encodes Granger-causal connections:

$$p(\mathbf{x}|\mathbf{z}) = p(\{\mathbf{x}_{T+1}, \cdots, \mathbf{x}_1\}|\mathbf{z}) = \prod_{t=1}^T p(\mathbf{x}_{t+1}|\mathbf{x}_t, \cdots, \mathbf{x}_1, \mathbf{z}).$$

---

[1]This stability-based step amounts to finding an optimal thresholding level for the "final" connections: the same model is fit to the time series in the reverse order, and "agreement" is sought between the Granger causal connections obtained respectively based on the original and the reverse time series, over a sequence of thresholding levels; the optimal one is determined by the one that maximizes the agreement measure. We refer interested readers to the original paper and references therein for more details.

Specifically, in their setting, $x_{i,t} \in \mathbb{R}^d$ is vector-valued and $z_{ij}$ corresponds to a categorical "edge type" between nodes $i$ and $j$. For example, it can be a binary edge type indicating presence/absence, or a more complex one having more categories. To simultaneously learn the edge types and the temporal dynamics, the model is formalized through a VAE that maximizes the evidence lower bound (ELBO), given by $\mathbb{E}_{q_\phi(\mathbf{z}|\mathbb{x})}(\log p_\theta(\mathbb{x}|\mathbf{z})) - \mathrm{KL}(q_\phi(\mathbf{z}|\mathbb{x}) \,\|\, p_\theta(\mathbf{z}))$, where $q_\phi(\mathbf{z}|\mathbb{x})$ is the probabilistic encoder, $p_\theta(\mathbb{x}|\mathbf{z})$ the decoder, and $p_\theta(\mathbf{z})$ the prior distribution. Concretely, the probabilistic encoder is given by $q_\phi(\mathbf{z}|\mathbb{x}) = \mathrm{softmax}\big(f_{\mathrm{enc},\phi}(\mathbb{x})\big)$ and it infers the type for each entry of $\mathbf{z}$; the function $f_{\mathrm{enc},\phi}$ is parameterized by neural networks. The decoder $p_\theta(\mathbb{x}|\mathbf{z}) = \prod_{t=1}^{T} p_\theta(\mathbf{x}_{t+1}|\mathbf{z}, \mathbf{x}_\tau : \tau \leq t)$ projects the trajectory based on past values and $\mathbf{z}$—specifically, the distributional parameters for each step forward. For example, if a Gaussian distribution is assumed, in the Markovian case, $p_\theta(\mathbf{x}_{t+1}|\mathbf{x}_t, \mathbf{z}) = \mathcal{N}(\mathrm{mean}, \mathrm{variance})$, where $\mathrm{mean} = f_{\mathrm{dec},\theta}^1(\mathbf{x}_t, \mathbf{z})$, $\mathrm{variance} = f_{\mathrm{dec},\theta}^2(\mathbf{x}_t, \mathbf{z})$ and $f_{\mathrm{dec},\theta}^1, f_{\mathrm{dec},\theta}^2$ are parameterized by some neural networks. Finally, maximizing the ELBO loss can be alternatively done by minimizing

$$-\mathbb{E}_{q_\phi(\mathbf{z}|\mathbb{x})}(\log p_\theta(\mathbb{x}|\mathbf{z})) + \mathrm{KL}(q_\phi(\mathbf{z}|\mathbb{x}) \,\|\, p_\theta(\mathbf{z})) := \text{negative log-likelihood} + H\big(q_\phi(\mathbf{z}|\mathbb{x})\big) + \mathrm{const};$$

the negative log-likelihood corresponds to the reconstruction error of the *entire* trajectory coming out of the decoder, and the KL divergence term boils down to the sum of entropies denoted by $H(\cdot)$ if the prior $p_\theta(\mathbf{z})$ is assumed to be a uniform distribution over edge types.

In summary, at the formulation level, generative model-based approaches treat Granger-causal connections (relationships) as a latent graph and learn it jointly with the dynamics, whereas predictive ones extract Granger-causal connections from the parameters that govern the dynamics in a post-hoc manner. The former can readily accommodate vector-valued nodes whereas for the latter, it becomes more involved and further complicates how the connections can be extracted/represented based on the model parameters. At the task level, to learn the model parameters, prediction-based approaches rely on tasks where the predicted values of the future one-step-ahead timestamp are of interest, whereas generative approaches amount to reconstructing the observed trajectories; prediction and reconstruction errors constitute part of the empirical risk minimization loss and the ELBO loss, respectively.

## 2.2 Multi-layer variational autoencoders

With a slight abuse of notation, in this subsection, we use $\mathbf{x}$ to denote the observed variable and $\mathbf{z}_l, l = 1, \cdots, L$ the latent ones for $L$ layers.

A "shallow" VAE with one latent layer is considered in the seminal work of Kingma & Welling (2014), where the generative model is given by $p_\theta(\mathbf{x}, \mathbf{z}_1) = p_\theta(\mathbf{x}|\mathbf{z}_1)p_\theta(\mathbf{z}_1)$, with $p_\theta(\mathbf{z}_1)$ denoting the prior distribution. Later works (Kingma et al., 2014; Burda et al., 2016; Sønderby et al., 2016) consider the extension into multiple latent layers, where the generative model can be represented through a cascading structure as follows:

$$p_\theta(\mathbf{x}, \{\mathbf{z}_l\}_{l=1}^L) = p_\theta(\mathbf{x}|\mathbf{z}_1)\Big(\prod_{l=1}^{L-1} p_\theta(\mathbf{z}_l|\mathbf{z}_{l+1})\Big)p_\theta(\mathbf{z}_L);$$

the corresponding inference model (encoder) is given by $q_\phi(\mathbf{z}_1, \cdots, \mathbf{z}_L|\mathbf{x}) = q_\phi(\mathbf{z}_1|\mathbf{x})\prod_{i=1}^L q_\phi(\mathbf{z}_l|\mathbf{z}_{l-1})$. The variational lower bound on $\log p(\mathbf{x})$ can be written as

$$\mathbb{E}_{q_\phi(\{\mathbf{z}\}_{l=1}^L|\mathbf{x})}\Big(\log p_\theta(\mathbf{x}|\{\mathbf{z}\}_{l=1}^L)\Big) - \mathrm{KL}\Big(q_\phi(\{\mathbf{z}\}_{l=1}^L|\mathbf{x}) \,\|\, p_\theta(\{\mathbf{z}\}_{l=1}^L)\Big), \tag{1}$$

with the first term corresponding to the reconstruction error.

**Conjugacy adjustment.** Under the above multi-layer setting, Sønderby et al. (2016) considers an inference model that recursively merges information from the "bottom-up" encoding and "top-down" decoding steps. Concretely, in the case where each layer is specified by a Gaussian distribution, the original distribution at layer $l$ after encoding is given by $q_\phi(\mathbf{z}_l|\mathbf{z}_{l-1}) \sim \mathcal{N}(\mu_{q,l}, \sigma_{q,l}^2)$ and the distribution at the same layer after decoding is given by $p_\theta(\mathbf{z}_l|\mathbf{z}_{l+1}) \sim \mathcal{N}(\mu_{p,l}, \sigma_{p,l}^2)$. The adjustment amounts to a precision-weighted combination that combines information from the decoder distribution into the encoder one, that is, $q_\phi(\mathbf{z}_l|\cdot) \sim \mathcal{N}(\tilde\mu_{q,l}, \tilde\sigma_{q,l}^2)$, where $\tilde\mu_{q,l} = (\mu_{q,l}\sigma_{q,l}^{-2} + \mu_{p,l}\sigma_{p,l}^{-2})/(\sigma_{q,l}^{-2} + \sigma_{p,l}^{-2})$ and $\tilde\sigma_{q,l}^2 = 1/(\sigma_{q,l}^{-2} + \sigma_{p,l}^{-2})$. This information-sharing mechanism

leads to richer latent representations and improved approximation of the log-likelihood function. A similar objective is also considered in Burda et al. (2016) and operationalized through importance weighting.

It is worth noting that the precision-weighted adjustment in Sønderby et al. (2016) is in the spirit of the conjugate analysis in Bayesian statistics. In particular, in Bayesian settings where the data likelihood is assumed Gaussian with *a fixed variance parameter* and the prior distribution is also assumed Gaussian, the posterior distribution possesses a closed-form Gaussian distribution (and hence conjugate w.r.t. the prior)[2]. For this reason, we term such an adjustment as the "conjugacy adjustment", which will be used later in our technical development.

Finally, utilizing multiple layers possessing a hierarchy as discussed above resembles the framework adopted in Bayesian hierarchical modeling. We provide a brief review of the topic in Section C.2. We also sketch in Section C.3 a modeling formulation under this framework for collection of linear VARs.

## 3 The Proposed Framework

Given a collection of trajectories for the same set of $p$ variables (nodes) from $M$ dynamical systems (entities), we are interested in estimating the Granger causal connections amongst the nodes in each system (i.e., entity-level connections), as well as the common "backbone" connections amongst the nodes that are shared across the entities (i.e., group-level connections).

To this end, we propose a two-layer VAE-based framework, wherein Granger-causal connections are treated as latent variables with a hierarchical structure, and they are learned jointly with the dynamics of the trajectories. In Section 3.1, we present the posited generative process that is suitable for the modeling task of interest, and give an overview of the proposed VAE-based formulation; the details of the components involved and their exact modeling considerations are discussed in Section 3.2. Section 3.3 provides a summary of the end-to-end training process and the inference tasks that can be performed based on the trained model.

The generalization of the proposed framework to the case of multiple levels of grouping across entities is deferred to Appendix F, where the *grand* common and the *group* common structures can be simultaneously learned with those of the entities.

### 3.1 An overview of the formulation

Consider a setting where there are $M$ entities, each of them having the same set of $p$ nodes, that evolve as a dynamical system. Let $x_{i,t}^{[m]}$ denote the value of node $i$ of entity $m \in \{1, \cdots, M\}$ at time $t$. It can be either scalar or vector-valued, with scalar node values being prevalent in traditional time-series settings; in the latter case, the nodes can be thought of as being characterized by vector-valued "features"[3]. Let $\mathbf{x}_t^{[m]} := (x_{1,t}^{[m]}, \cdots, x_{p,t}^{[m]})$ be the collection of node values at time $t$ for entity $m$, and $x^{[m]} := \{\mathbf{x}_1^{[m]}, \cdots, \mathbf{x}_T^{[m]}\}$ the corresponding trajectory over time. Further, let $\mathbf{z}^{[m]} \in \mathbb{R}^{p \times p}$ denote the Granger-causal connection matrix of entity $m$ and $\bar{\mathbf{z}} := [\bar{z}_{ij}] \in \mathbb{R}^{p \times p}$ the common structure embedded in $\mathbf{z}^{[1]}, \cdots, \mathbf{z}^{[M]}$, and note that it does *not* necessarily correspond to the arithmetic mean of the $\mathbf{z}^{[m]}$'s. In the remainder of this paper, we may refer to these matrices as "graphs" interchangeably.

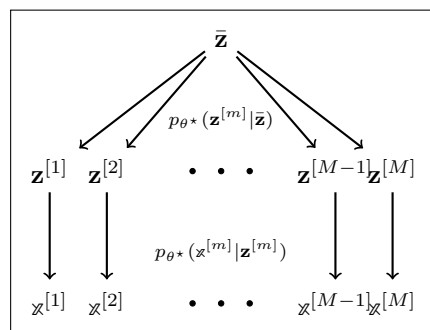

Figure 1: Diagram for the postulated top-down generative process.

Depending on the modeling scenario, the entity-level Granger-causal connections $\mathbf{z}^{[m]}$ can either be binary or continuous. In the former case, it corresponds precisely to the

---

[2]Note that in the case where the data likelihood is Gaussian, but the variance is no longer a fixed parameter, the Normal-Normal conjugacy does not necessarily go through.

[3]For example, in the Springs experiment in Kipf et al. (2018) (which is also considered in this paper; see Appendix B.1), the features correspond to a 4-dimensional vector, with the first two coordinates being the 2D velocity and the last two being the 2D location

*aggregate Granger causal graph* defined in Dahlhaus & Eichler (2003); in the latter case, its $(i,j)$-entry (scalar value) reflects the strength of the relationship between the past value(s) of node $j$ the present value of node $i$; see Appendix C.1 and Remark 6 for a detailed discussion.

The posited generative process, whose true parameters are denoted by $\theta^\star$, is given by:

$$
\begin{aligned}
p_{\theta^\star}\Big(\{\mathbb{x}^{[m]}\}_{m=1}^M, \{\mathbf{z}^{[m]}\}_{m=1}^M, \bar{\mathbf{z}}\Big) &= p_{\theta^\star}\Big(\{\mathbb{x}^{[m]}\}_{m=1}^M | \{\mathbf{z}^{[m]}\}_{m=1}^M\Big) \cdot p_{\theta^\star}\Big(\{\mathbf{z}^{[m]}\}_{m=1}^M | \bar{\mathbf{z}}\Big) \cdot p_{\theta^\star}(\bar{\mathbf{z}}) \\
&= \prod_{m=1}^M p_{\theta^\star}(\mathbb{x}^{[m]} | \mathbf{z}^{[m]}) \prod_{m=1}^M p_{\theta^\star}(\mathbf{z}^{[m]} | \bar{\mathbf{z}}) \prod_{1 \le i,j \le p} p_{\theta^\star}(\bar{z}_{ij}).
\end{aligned}
\tag{2}
$$

The decomposition is based on the following underlying assumptions (see also Figure 1 for a pictorial illustration):

- conditional on the entity-specific graphs $\mathbf{z}^{[m]}$, their trajectories $\mathbb{x}^{[m]}$'s are *independent* of the grand common $\bar{\mathbf{z}}$, and they are conditionally independent from each other given their respective entity-specific graphs $\mathbf{z}^{[m]}$'s

- the entity-specific graphs $\mathbf{z}^{[m]}$ are conditionally independent given the common graph $\bar{\mathbf{z}}$

- the prior distribution $p_{\theta^\star}(\bar{\mathbf{z}})$ factorizes over the edges.

The proposed model creates a hierarchy between the common graph and the entity-specific ones, which in turn naturally provides a coupling mechanism amongst the latter. The grand common structure can be estimated as one learns all the latent components jointly with the dynamics of the system through a VAE. Let $\mathcal{X} := \{\mathbb{x}^{[1]}, \cdots, \mathbb{x}^{[m]}\}$, $\mathcal{Z} := \{\bar{\mathbf{z}}, \mathbf{z}^{[1]}, \cdots, \mathbf{z}^{[m]}\}$, $q_\phi(\mathcal{Z}|\mathcal{X})$ denote the encoder, $p_\theta(\mathcal{X}|\mathcal{Z})$ the decoder and $p_\theta(\mathcal{Z})$ the prior distribution. Then, the ELBO is given by

$$
\mathbb{E}_{q_\phi(\mathcal{Z}|\mathcal{X})}\Big(\log p_\theta(\mathcal{X}|\mathcal{Z})\Big) - \mathrm{KL}\Big(q_\phi(\mathcal{Z}|\mathcal{X}) \,\big\|\, p_\theta(\mathcal{Z})\Big),
$$

and serves as the objective function for the end-to-end encoding-decoding procedure as depicted in Figure 2.

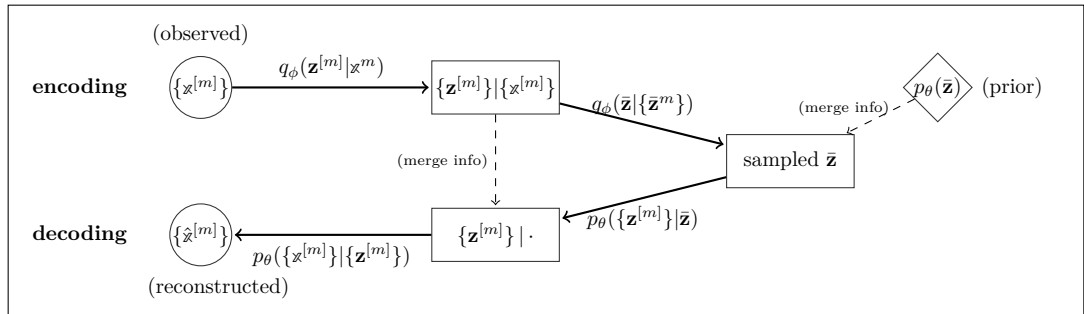

Figure 2: Diagram for the end-to-end encoding-decoding procedure. Solid paths with arrows denote modeling the corresponding distributions during the encoding/decoding process; dashed paths with arrows correspond to information merging based on (weighted) conjugacy adjustment. Quantities obtained after each step are given inside the circles/rectangles. $\{\mathbb{x}^{[m]}\}$ is short for the collection $\{\mathbb{x}^{[m]}\}_{m=1}^M$; $\{\mathbf{z}^{[m]}\}$ is analogously defined.

*Remark* 1 (On the proposed formulation). (1) Depending on the assumption on the entity-level Granger-causal connections $\mathbf{z}^{[m]}$—either binary or continuous—encoder/decoder distributions can then be selected accordingly. In particular, distributions that form conjugate pairs (e.g., Gaussian-Gaussian for the continuous case and Beta-Bernoulli for the binary case) can facilitate computations. (2) The proposed framework naturally allows estimation of positive/negative connections in a principled way without resorting to ad-hoc aggregation schemes. It also enables incorporation of external information pertaining to the presence/absence of connections through the decoder. (3) In settings where a large collection of entities is available, but each entity has limited sample size, the joint learning framework can be advantageous over an individual entity learning one.

## 3.2 Modeling details

Next, we provide details on the specification of the encoder and the decoder, the sampling steps, and the loss function calculations for model (2).

### 3.2.1 Encoder

The goal of the encoder is to infer the latent graphs $\mathcal{Z} := \{\bar{\mathbf{z}}, \mathbf{z}^{[1]}, \cdots, \mathbf{z}^{[M]}\}$ based on the observed trajectories $\mathcal{X} := \{\varkappa^{[1]}, \cdots, \varkappa^{[M]}\}$.

Let $\phi$ denote the collection of parameters in the encoder $q_\phi(\mathcal{Z}|\mathcal{X})$. To delineate the dependency between the trajectories and the graphs, the following assumptions are imposed:

- conditioning on $\{\mathbf{z}^{[m]}\}_{m=1}^M$, $\bar{\mathbf{z}}$ is independent of $\{\varkappa^{[m]}\}_{m=1}^M$ and the conditional probability $q_\phi\big(\bar{\mathbf{z}}\big|\{\mathbf{z}^{[m]}\}_{m=1}^M\big)$ factorizes across edges $(i,j)$;
- the entity-specific graphs are conditionally independent given their corresponding trajectories, i.e., $q_\phi\big(\{\mathbf{z}^{[m]}\}_{m=1}^M\big|\{\varkappa^{[m]}\}_{m=1}^M\big)$ factorizes across entities.

These assumptions are in line with the structure of the model in (2), in that the conditional dependencies posited in the generative model are respected during the "bottom-up" encoding process.

Consequently, the encoder can be decomposed into the following product components:

$$q_\phi\big(\mathcal{Z}|\mathcal{X}\big) = q_\phi\Big(\bar{\mathbf{z}}\,\big|\,\{\mathbf{z}^{[m]}\}_{m=1}^M\Big) \prod_{m=1}^M q_\phi\Big(\mathbf{z}^{[m]}|\varkappa^{[m]}\Big) = \prod_{1 \le i,j \le p} q_\phi\Big(\bar{z}_{ij}\,\big|\{z_{ij}^{[m]}\}_{m=1}^M\Big) \prod_{m=1}^M q_\phi\Big(\mathbf{z}^{[m]}|\mathbf{x}^{[m]}\Big).$$

There are two types of terms in the above expression: $q_\phi(\mathbf{z}^{[m]}|\varkappa^{[m]})$ that infers each entity's latent graph based on its trajectory, and $q_\phi(\bar{z}_{ij}|\{z_{ij}^{[m]}\}_{m=1}^M)$ that obtains the grand common based on the entity-level graphs, in an edge-wise manner. Note that for $q_\phi(\bar{z}_{ij}|\{z_{ij}^{[m]}\}_{m=1}^M)$, together with modeling $p_\theta(z_{ij}^{[m]}|\bar{z}_{ij})$, resembles prior-posterior calculations in Bayesian statistics using conjugate pairs of distributions; hence, depending on the underlying structural assumptions (continuous or binary) on the $\mathbf{z}^{[m]}$'s, one can choose emission heads (or equivalently, the output functional form) accordingly.

At the high level, the encoder can be abstracted into 3 modules, parameterized through $f_{x \to h}, f_{h \to z}$ and $f_{z \to \bar{z}}$, respectively:

(enc-a) trajectory to hidden representation $\varkappa^{[m]} \to \mathbf{h}^{[m]} := f_{x \to h}(\varkappa^{[m]})$, with $\mathbf{h}_{ij}^{[m]}$ corresponding to the edge-specific one;

(enc-b) hidden representation to the entity-specific graph: $\mathbf{h}^{[m]} \to \mathbf{z}^{[m]} := f_{h \to z}(\mathbf{h}^{[m]})$;

(enc-c) entity-level graphs to the grand common (edge-wise): $\{z_{ij}^{[m]}\}_{m=1}^M \to \bar{z}_{ij} := f_{z \to \bar{z}}(\{z_{ij}^{[m]}\}_{m=1}^M)$.

Modules (enc-a) and (enc-b) combined, model $q_\phi(\mathbf{z}^{[m]}|\varkappa^{[m]})$ and correspond to "Trajectory2Graph" operations, while module (enc-c) models $q_\phi(\bar{z}_{ij}|\{z_{ij}^{[m]}\}_{m=1}^M)$ and captures the "Entity2Common" one. On the other hand, given the above-mentioned conjugate pair consideration, the choices of $f_{h \to z}$ and $f_{z \to \bar{z}}$ are considered jointly.

Formally, for $f_{x \to h}$, we use a similar approach to that in Kipf et al. (2018), where $f_{x \to h}$ entails message-passing operations that are widely adopted in the literature related to graph neural networks (Scarselli et al., 2008; Gilmer et al., 2017). At a high level, these operations entail "node2edge" (concatenating the representation of the node stubs) and "edge2node" (aggregating the representation of incoming edges) iteratively and non-linear functions (e.g., MLPs) in between. The operation ultimately leads to $\{\mathbf{h}_{ij}^{[m]}\}$, with $\mathbf{h}_{ij}^{[m]} \in \mathbb{R}^{n_{\text{hid}}}$ being a $n_{\text{hid}}$-dimensional hidden representation corresponding to $z_{ij}^{[m]}$. Full details are provided in Appendix A.1 wherein we also provide a pictorial illustration for the operations.

Once the $\mathbf{h}_{ij}^{[m]}$'s are obtained, subsequent modeling in modules (enc-b) and (enc-c) can be generically represented as

$$z_{ij}^{[m]}\,|\,\mathbf{h}_{ij}^{[m]} \sim q_z(\,\cdot\,; \delta_{q,ij}^{[m]}), \qquad \text{and} \qquad \bar{z}_{ij}\,|\{z_{ij}^{[m]}\} \sim q_{\bar{z}}(\,\cdot\,; \bar{\delta}_{ij}),$$

where $q_z(\,\cdot\,;\delta_{q,ij}^{[m]})$ is some distribution with parameter $\delta_{q,ij}^{[m]} := f_{h\to z}(\mathbf{h}_{ij}^{[m]})$ being the function output of $f_{h\to z}$. Similarly, $q_{\bar z}(\,\cdot\,;\bar\delta_{q,ij})$ is some distribution with parameter $\bar\delta_{q,ij} := f_{z\to\bar z}(\{z_{ij}^{[m]}\})$ being the function output of $f_{z\to\bar z}$. The exact choices for $f_{h\to z}$ and $f_{z\to\bar z}$ bifurcate depending on the scenario:

- Case 1, $\mathbf{z}^{[m]}$'s entries being continuous: in this case, we consider a Gaussian-Gaussian emission head pair. Consequently, $\delta_{q,ij}^{[m]} = \{\mu_{q,ij}^{[m]}, (\sigma^{[m]})_{q,ij}^2\}$, $\bar\delta_{q,ij} = \{\bar\mu_{q,ij}, \bar\sigma_{q,ij}^2\}$;

$$q_z \sim \mathcal{N}\Big(\mu_{q,ij}^{[m]}, (\sigma^{[m]})_{q,ij}^2\Big); \quad \mu_{q,ij}^{[m]} := f_{h\to z}^1(\mathbf{h}_{ij}^{[m]}),\ (\sigma^{[m]})_{q,ij}^2 := f_{h\to z}^2(\mathbf{h}_{ij}^{[m]}); \tag{3}$$

$$q_{\bar z} \sim \mathcal{N}\Big(\bar\mu_{q,ij}, \bar\sigma_{q,ij}^2\Big); \quad \bar\mu_{q,ij} := f_{z\to\bar z}^1(\{z_{ij}^{[m]}\}),\ \bar\sigma_{q,ij}^2 := f_{\bar z\to z}^2(\{z_{ij}^{[m]}\}). \tag{4}$$

$f_{h\to z}^1, f_{h\to z}^2$ are component functions of $f_{h\to z}$, each with an $n_{\text{hid}}$-dimensional input and a scalar output; they can be simple linear functions with $f_{h\to z}^2$ having an additional softplus operation to ensure positivity. Similarly, $f_{z\to\bar z}^1, f_{z\to\bar z}^2$ comprise $f_{z\to\bar z}$, each with an $m$-dimensional input and a scalar output; in practice their functional form can be as simple as taking the sample mean and standard deviation, respectively.

- Case 2, $\mathbf{z}^{[m]}$'s entries being binary: in this case, we consider a Beta-Bernoulli emission head pair, i.e.,

$$q_z \sim \text{Ber}\Big(\delta_{q,ij}^{[m]}\Big); \quad \delta_{q,ij}^{[m]} := f_{h\to z}(\mathbf{h}_{ij}^{[m]}), \tag{5}$$

$$q_{\bar z} \sim \text{Beta}\Big(\bar\alpha_{q,ij}, \bar\beta_{q,ij}\Big); \quad \bar\alpha_{q,ij} := f_{z\to\bar z}^1(\{z_{ij}^{[m]}\}),\ \bar\beta_{q,ij} := f_{z\to\bar z}^2(\{z_{ij}^{[m]}\}). \tag{6}$$

The output of $f_{h\to z}$ corresponds to the Bernoulli success probability and it is parameterized with an MLP with the last layer performing sigmoid activation to ensure that the output lies in $(0,1)$. $f_{z\to\bar z}^1$ and $f_{z\to\bar z}^2$ are component functions of $f_{z\to\bar z}$. Similar to the Gaussian case, their choice need not be complicated and is chosen based on moment-matching.

Note that the prior distribution $p_\theta(\bar z_{ij})$ is also selected according to the underlying scenario, with a standard Normal distribution used in the continuous case and a $\text{Beta}(1,1)$ in the binary case. Once the distribution parameters for $\bar z_{ij}$ are obtained based on (4) or (6), we apply conjugacy adjustment to incorporate also the information from the prior, before the sampling step takes place.

### 3.2.2 Decoder

The goal of the decoder $p_\theta(\mathcal{X}|\mathcal{Z})$ is to reconstruct the trajectories based on the entity and group level graphs, and its components follow from the generative process described in (2), that is,

$$p_\theta(\mathcal{X}|\mathcal{Z}) = p_\theta\Big(\{\varkappa^{[m]}\}_{m=1}^M | \{\mathbf{z}^{[m]}\}_{m=1}^M\Big) \cdot p_\theta\Big(\{\mathbf{z}^{[m]}\}_{m=1}^M | \bar{\mathbf{z}}\Big) = \prod_{m=1}^M p_\theta(\varkappa^{[m]}|\mathbf{z}^{[m]}) \prod_{m=1}^M p_\theta(\mathbf{z}^{[m]}|\bar{\mathbf{z}}),$$

where $\theta$ denotes the collections of parameters in the decoder. The two components $p_\theta(\mathbf{z}^{[m]}|\bar{\mathbf{z}})$ and $p_\theta(\varkappa^{[m]}|\mathbf{z}^{[m]})$, respectively capture the dependency between the entity-specific graphs $\mathbf{z}^{[m]}$'s and their grand common $\bar{\mathbf{z}}$, and the evolution of the trajectories given $\mathbf{z}^{[m]}$. Consequently, the decoder can be broken into two modules, parameterized through $g_{\bar z\to z}$ and $g_{z\to x}$:

(dec-a) $p_\theta(\mathbf{z}^{[m]}|\bar{\mathbf{z}})$, the grand common to entity-specific graphs $\mathbf{z} \to \mathbf{z}^{[m]} := g_{\bar z\to z}(\bar{\mathbf{z}})$, with $g_{\bar z\to z}(\cdot)$ acting on the sampled $\bar{\mathbf{z}}$ (edge-wise). Samples drawn from this distribution will be used to guide the evolution of the trajectories of the corresponding entity;

(dec-b) $p_\theta(\varkappa^{[m]}|\mathbf{z}^{[m]})$, graph to trajectory $\mathbf{z}^{[m]} \to \varkappa^m$; concretely,

$$p_\theta(\varkappa^{[m]}|\mathbf{z}^{[m]}) = p_\theta(\mathbf{x}_1^{[m]}|\mathbf{z}^{[m]}) \prod_{t=2}^T p_\theta\Big(\mathbf{x}_t^{[m]} \mid \mathbf{x}_{t-1}^{[m]}, ..., \mathbf{x}_1^{[m]}, \mathbf{z}^{[m]}\Big),$$

with $p_\theta(\mathbf{x}_t^{[m]} \mid \mathbf{x}_{t-1}^{[m]}, ..., \mathbf{x}_1^{[m]}, \mathbf{z}^{[m]})$ modeled through $g_{z\to x}(\mathbf{x}_{t-1}^{[m]}, \cdots, \mathbf{x}_{t-q}^{[m]}, \mathbf{z}^{[m]})$ assuming a fixed context length of $q$ (or $q$-lag dependency, equivalently).

We refer to these two modules as "Common2Entity" and "Graph2Trajectory", respectively.

**Common2Entity.** We consider a weighted conjugacy adjustment that merges the information from the encoder distribution into the decoder one, so that it contains both the grand common and the entity-specific information. Concretely, for some pre-specified weight $\omega \in [0, 1]$,

- Case 1, in the continuous case, let $p_\theta(z_{ij}^{[m]}|\bar{z}_{ij}) \sim \mathcal{N}(\mu_{p,ij}^{[m]}, (\sigma^{[m]})_{p,ij}^2)$ with $\mu_{p,ij}^{[m]} := g_{\bar{z}\to z}^1(\bar{\mathbf{z}}_{ij}^{[m]})$ and $(\sigma^{[m]})_{p,ij}^2 := g_{\bar{z}\to z}^2(\bar{\mathbf{z}}_{ij}^{[m]})$; $g_{\bar{z}\to z}^1, g_{\bar{z}\to z}^2 : \mathbb{R} \mapsto \mathbb{R}$ are component functions of $g_{\bar{z}\to z}$. This gives the "unadjusted" distribution that contains only the grand common information. With $\mu_{q,ij}^{[m]}$ and $(\sigma^{[m]})_{q,ij}^2$ obtained in (3), the weighted adjustment gives $p_\theta(z_{ij}^{[m]}|\cdot) \sim \mathcal{N}\left(\tilde{\mu}_{p,ij}^{[m]}, (\tilde{\sigma}^{[m]})_{p,ij}^2\right)$, where

$$\tilde{\mu}_{p,ij}^{[m]} := \frac{\omega\mu_{q,ij}^{[m]}(\sigma^{[m]})_{q,ij}^{-2} + (1-\omega)\mu_{p,ij}^{[m]}(\sigma^{[m]})_{p,ij}^{-2}}{\omega(\sigma^{[m]})_{q,ij}^{-2} + (1-\omega)(\sigma^{[m]})_{p,ij}^{-2}}, \quad (\tilde{\sigma}^{[m]})_{p,ij}^2 := \frac{1}{\omega(\sigma^{[m]})_{q,ij}^{-2} + (1-\omega)(\sigma^{[m]})_{p,ij}^{-2}}. \quad (7)$$

- Case 2, in the binary case, let $p_\theta(z_{ij}^{[m]}|\bar{z}_{ij}) \sim \text{Ber}(\delta_{p,ij}^{[m]})$, where $\delta_{p,ij}^{[m]} := g_{\bar{z}\to z}(\bar{z}_{ij})$. With $\delta_{q,ij}^{[m]}$ obtained in (5), the weighted adjustment gives

$$p_\theta(z_{ij}^{[m]}|\cdot) \sim \text{Ber}\left(\tilde{\delta}_{p,ij}^{[m]}\right); \quad \tilde{\delta}_{p,ij}^{[m]} = \frac{1}{\omega/\delta_{q,ij}^{[m]} + (1-\omega)/\delta_{p,ij}^{[m]}}. \quad (8)$$

Similar to the function $f_{z\to\bar{z}}$ in the encoder, here $g_{\bar{z}\to z}$ corresponds to $f_{z\to\bar{z}}$'s "reverse-direction" counterpart and its choice can be rather simple[4].

*Remark* 2 (On the role of $\omega$). It governs the mixing percentage of the entity-specific and the common information: when $\omega = 1$, the "tilde" parameters of the post-adjustment distribution effectively collapse into the encoder ones (e.g., $\tilde{\delta}_{p,ij} \equiv \delta_{q,ij}^{[m]}$ and analogously for $\tilde{\mu}_{p,ij}, \tilde{\sigma}_{p,ij}^2$); correspondingly, samples drawn from $p_\theta(z_{ij}^{[m]}|\cdot)$ essentially ignore the sampled $\bar{\mathbf{z}}$ and hence they can be viewed as entirely entity-specific. At the other extreme, for $\omega = 0$, the tilde parameters coincide with the unadjusted ones; therefore, apart from the grand common information carried in the sampled $\bar{\mathbf{z}}$, no entity-specific one is passed onto the sampled $\mathbf{z}^{[m]}$. By varying $\omega$ between $(0, 1)$, one effectively controls the level of heterogeneity and how strongly the sampled entity-specific graphs deviate from the grand common one.

**Graph2Trajectory.** Module (dec-b) pertains to modeling the dynamics of the trajectory $\mathbb{x}^{[m]}$ given the sampled $\mathbf{z}^{[m]}$. Here, we focus on one-step Markovian dependency, i.e., $q = 1$ and thus $p_\theta(\mathbf{x}_t^{[m]}|\mathbf{x}_{t-1}^{[m]}, ..., \mathbf{x}_1^{[m]}, \mathbf{z}^{[m]}) \approx g_{z\to x}(\mathbf{x}_{t-1}^{[m]}, \mathbf{z}^{[m]})$. The extension to longer lag dependencies ($q > 1$) can be readily obtained by pre-processing the input accordingly, as discussed in Appendix A.2.

We consider the following parameterization of $g_{z\to x}$. At the high level, given that $z_{ij}^{[m]}$ corresponds to the Granger-causal connection from node $j$ to node $i$, it should serve as a "gate" controlling the amount of information that can be passed from $x_{j,t-1}^{[m]}$ to $x_{i,t}^{[m]}$. To this end, each response coordinate $x_{i,t}^{[m]}$ is modeled as follows:

$$u_{i,t-1}^{[m],j} := \check{x}_{j,t-1}^{[m]} \circ z_{ij}^{[m]} \text{ (gating)}, \quad \mathbf{u}_{i,t-1}^{[m]} = \{u_{i,t-1}^{[m],1}, \cdots, u_{i,t-1}^{[m],p}\}, \quad \text{and} \quad \check{\mathbf{u}}_{i,t-1}^{[m]} := \text{MLP}(\mathbf{u}_{i,t-1}^{[m]}); \quad (9)$$

$$x_{i,t}^{[m]} \sim \mathcal{N}(\mu_{x,it}^{[m]}, (\sigma^{[m]})_{x,it}^2), \quad \text{where} \quad \mu_{x,it}^{[m]} := \text{Linear}(\check{\mathbf{u}}_{i,t-1}^{[m]}), \quad (\sigma^{[m]})_{x,it}^2 = \text{Softplus}\left(\text{Linear}(\check{\mathbf{u}}_{i,t-1}^{[m]})\right). \quad (10)$$

Note that in the gating operation in (9), we use $\check{x}_{j,t-1}^{[m]}$ to denote the output after some potential numerical embedding step (e.g., Gorishniy et al. (2022)) of $x_{j,t-1}^{[m]}$; in the absence of such embedding, $\check{x}_{j,t-1}^{[m]} \equiv x_{j,t-1}^{[m]}$. Through the gating step[5], $x_{j,t-1}^{[m]}$ exerts its impact on $x_{i,t}^{[m]}$ entirely through $u_{i,t-1}^{[m],j}$. The continuous case and the binary case $z_{ij}^{[m]}$ can be treated in a unified manner: in the former case, the value of $z_{ij}^{[m]}$ corresponds to the strength; in the latter case, it performs masking. Subsequently, $\mathbf{u}_{i,t-1}^{[m]}$ collects the $u_{i,t-1}^{[m],j}$'s of all nodes $j = 1, \cdots, p$, and serves as the predictor for $x_{i,t}^{[m]}$. Finally, if one simply sums all $u_{i,t-1}^{[m],j}$'s to obtain the mean of $x_{i,t}^{[m]}$, then it effectively coincides with the operation in a linear VAR system, with $z_{ij}^{[m]}$ corresponding precisely to the entries in the transition matrix.

---

[4]In our experiments, we use an identity function and it has been effective across the settings considered.

[5]Note that $z_{ij}^{[m]}$ is a scalar and is applied to all coordinates of $\check{x}_{j,t-1}^{[m]}$ in the case the latter is a vector.

*Remark* 3. The above-mentioned choice of $g_{z \to x}$ can be viewed as a "node-centric" one, wherein entries $z_{ij}^{[m]}$ control the information passing directly through the nodes. As an alternative, one can consider an "edge-centric" one, which leverages the idea of message-passing in GNNs and entails "node2edge" and "edge2node" operations. This resembles the technology adopted in Kipf et al. (2018); Löwe et al. (2022) that consider primarily having graph entries corresponding to categorical edge types, which, after some adaptation, can be used to handle the numerical case. In practice, we observe that the edge-centric graph2trajectory decoder can lead to instability for time series signals[6]. A more detailed comparison can be found in Appendix A.2, where additional illustrations are provided for the two.

### 3.2.3 Sampling

Given the stochastic nature of the sampled quantities, drawing samples from the encoded/decoded distributions requires special handling to enable the gradient to back propagate. Depending on whether entries of $\mathbf{z}^{[m]}$ are continuous or binary, there are three possible types of distributions involved; for notational simplicity, here we use $z$ to represent generically the random variable under consideration.

- Normal $z \sim \mathcal{N}(\mu, \sigma^2)$. In this case, the "standard" reparameterization trick (Kingma & Welling, 2014) can be used, that is, $z = \mu + \sigma \circ \epsilon$, $\epsilon \sim \mathcal{N}(0, 1)$.

- Bernoulli $z \sim \mathrm{Ber}(\delta)$. In this case, the discrete distribution is approximated by its continuous relaxation (Maddison et al., 2017). Concretely, $z = \mathrm{softmax}((\log(\boldsymbol{\pi}) + \boldsymbol{\epsilon})/\tau)$ where $\boldsymbol{\epsilon} \in \mathbb{R}^2$ whose coordinates are i.i.d. samples from $\mathrm{Gumbel}(0, 1)$, $\boldsymbol{\pi} = (1 - \delta, \delta)$ is the binary class probability and $\tau$ is the temperature.

- Beta $z \sim \mathrm{Beta}(\alpha, \beta)$. In this case, implicit reparameterization of the gradients (Figurnov et al., 2018) is leveraged and the construction of the reparameterized samples becomes much more involved. We refer interested readers to Figurnov et al. (2018); Jankowiak & Obermeyer (2018) for an in-depth discussion on how parameterized random variables can be obtained and become differentiable.

### 3.2.4 Loss function

The loss function is given by the negative ELBO, that is,[7]

$$-\mathbb{E}_{q_\phi(\mathcal{Z}|\mathcal{X})}\Big(\log p_\theta(\mathcal{X}|\mathcal{Z})\Big) + \mathrm{KL}\Big(q_\phi(\mathcal{Z}|\mathcal{X}) \,\big\|\, p_\theta(\mathcal{Z})\Big) =: \text{reconstruction error} + \text{KL};$$

the first term corresponds to the reconstruction error that measures the deviation between the original trajectories and the reconstructed ones, while the KL term measures the "consistency" between the encoded and the decoded distributions, and can be viewed as a type of regularization.

Let $\boldsymbol{\mu}_{x,t}^{[m]} := (\mu_{x,1t}^{[m]}, \cdots, \mu_{x,pt}^{[m]})^\top$ and $\Sigma_{\mathbf{x}_t^{[m]}} := \mathrm{diag}((\sigma^{[m]})_{x,1t}^2, \cdots, (\sigma^{[m]})_{x,pt}^2)^\top$ with the components defined in (10). The reconstruction error is the negative Gaussian log-likelihood loss given by

$$\sum_{m=1}^{M} \Big( \sum_{t=2}^{T} \big(\mathbf{x}_t^{[m]} - \boldsymbol{\mu}_{x,t}^{[m]}\big)^\top \Sigma_{\mathbf{x}_t^{[m]}}^{-1} \big(\mathbf{x}_t^{[m]} - \boldsymbol{\mu}_{x,t}^{[m]}\big) + \log |\Sigma_{\mathbf{x}_t^{[m]}}| \Big). \tag{11}$$

The KL term can be simplified after some algebra to (see Appendix A.3 calculation):

$$\mathbb{E}_{q_\phi(\mathcal{Z}|\mathcal{X})}\Big[\mathrm{KL}\Big(q_\phi(\bar{\mathbf{z}}|\{\mathbf{z}^{[m]}\}) \,\big\|\, p_\theta(\bar{\mathbf{z}})\Big)\Big] + \mathbb{E}_{q_\phi(\mathcal{Z}|\mathcal{X})}\Big[\mathrm{KL}\Big(q_\phi(\{\mathbf{z}^{[m]}\}|\{\mathbf{x}^{[m]}\}) \,\big\|\, p_\theta(\{\mathbf{z}^{[m]}\}|\bar{\mathbf{z}})\Big)\Big]; \tag{12}$$

both terms can be viewed as "consistency matching" terms that measure the divergence between the distributions obtained in the encoder pass and that from the decoder pass. Finally, note that in the implementation, the quantities involved are replaced by their conjugacy adjusted counterparts wherever applicable, and this is similar to the treatment in Sønderby et al. (2016).

---

[6]to contrast with the physical system (e.g., Springs) considered in the experiments of Kipf et al. (2018).
[7]Recall that $\mathcal{X} := \{\mathbf{x}^{[m]}; m = 1, \cdots, M\}$ and $\mathcal{Z} := \{\bar{\mathbf{z}}, \mathbf{z}^{[m]}; m = 1, \cdots, M\}$.

### 3.3 Training and inference

The functions in the encoder ($f_{x \to h}$, $f_{h \to z}$ and $f_{z \to \bar{z}}$) and those in the decoder ($g_{\bar{z} \to z}$ and $g_{z \to x}$) are shared across all entities $m = 1, \cdots, M$, and thus the model is trained based on the "pooled" data of all entities, while keeping track of the entity id that each data block is associated with. The steps involved in the end-to-end training under the proposed framework are summarized in Exhibit 1.

---

**Exhibit 1:** Outline of steps for training under the two-layer VAE-based framework

---

**Input:** observed trajectories $\{x^{[1]}, \cdots, x^{[M]}\}$, hyperparameters. Let $\langle M \rangle := \{1, \cdots, M\}$.

– **Forward pass, encoder:** $\{x^{[m]}\} \to \{\mathbf{z}^{[m]}\} \to \bar{\mathbf{z}}$
   0. `[Traj2Graph]` $m \in \langle M \rangle$: obtain the encoded distribution for entity-specific graphs $q_\phi(\mathbf{z}^{[m]}|x^{[m]})$;
   1. $m \in \langle M \rangle$: sample $\mathbf{z}^{[m]}$ from $q_\phi(\mathbf{z}^{[m]}|x^{[m]})$;
   2. `[Entity2Common]` based on $\{\mathbf{z}^{[m]}\}_{m=1}^M$, obtain the encoded distribution for the common graph
     $q_\phi(\bar{\mathbf{z}}|\{\mathbf{z}^{[m]}\})$;
– **Forward pass, decoder:** $\bar{\mathbf{z}} \to \{\mathbf{z}^{[m]}\} \to \{x^m\}$
   3. merge prior info $p_\theta(\bar{\mathbf{z}})$ into $q_\phi(\bar{\mathbf{z}}|\{\mathbf{z}^{[m]}\})$ then sample $\bar{\mathbf{z}}$;
   4. `[Common2Entity]` $m \in \langle M \rangle$: obtain the decoded distribution for entity-specific graphs $p_\theta(\mathbf{z}^{[m]}|\bar{\mathbf{z}})$;
   5. $m \in \langle M \rangle$: merge entity-specific encoded info $q_\phi(\mathbf{z}^{[m]}|x^{[m]})$ into $p_\theta(\mathbf{z}^{[m]}|\bar{\mathbf{z}})$, then sample $(\mathbf{z}^{[m]}|\cdot)$;
   6. `[Graph2Traj]` $m \in \langle M \rangle$: using $\mathbf{z}^{[m]}$ and the lag info $\mathbf{x}_{t-1}^{[m]}$, decode to get $\hat{\mathbf{x}}_t^{[m]}$; $t = 2, \cdots, T$.
– **Loss calculation**
   7. calculate the ELBO loss by summing up (11) and (12);
– **Backward pass**: update neural network parameters based on gradients (back-propagation)

**Output:** Trained encoder and decoder

---

Several pertinent remarks follow. (1) The data typically consist of "long" trajectories that contain all the available observations (time points); one needs to partition them to "short" ones of length $T$ (that are typically between 20-50), which constitute the samples used in model training. See Appendix A.5 for additional illustration. (2) In the case where one has external information regarding presence or absence of edges in the $\mathbf{z}^{[m]}$'s, it can be incorporated by enforcing the corresponding entries to zero after the former are sampled in Step 5. (3) Once the encoder (inference model) and the decoder (generative model) are trained, the latent graphs can be obtained by applying the trained encoder on the trajectories. For entity-specific graphs $\mathbf{z}^{[m]}$'s, the inference model gives the encoded distribution $q_\phi(\mathbf{z}^{[m]}|\mathbf{x}^{[m]})$'s. In practice, the graph of interest is extracted by calculating the "mode" of the distribution; the grand common graph $\bar{\mathbf{z}}$ can be analogously handled. It is worth noting that for continuous $\mathbf{z}^{[m]}$'s, the proposed framework naturally provides signed estimates and thus positive/negative Granger causal connections can be readily differentiated (see Appendix E for a detailed discussion). (4) The trained decoder can be utilized to quantify also the *predictive* strength of the Granger-causal connection, as discussed in Appendix A.4.

## 4 Synthetic Data Experiments

We evaluate the performance of the proposed framework, together with benchmarking methods on several synthetic data settings. For all experiments, we start from a common graph that corresponds to $\bar{\mathbf{z}}$, add perturbations to it for individual entities to produce heterogeneous Granger-causal connections (i.e., the $\mathbf{z}^{[m]}$'s), then simulate trajectories $\{x^{[m]}\}$ corresponding to each entity based on their respective $\mathbf{z}^{[m]}$'s and the specified dynamics. The estimated entity-specific and grand common graphs are then evaluated against the underlying truth, for both the proposed and competing methods.

Prediction model-based competitors[8] include `NGC` (Tank et al., 2021), `GVAR` (Marcinkevičs & Vogt, 2021) and `TCDF` (Nauta et al., 2019), and a regularized linear VAR model based estimator (`Linear`; e.g., Basu & Michailidis (2015)). For generative model-based ones, we consider variations of Löwe et al. (2022). Note

---

[8]The selection of these competitors is based on the results reported in Marcinkevičs & Vogt (2021). Specifically, we picked the ones that were demonstrated to be competitive. The code implementations for these competitors (except for the regularized Linear VAR) are directly taken from the repositories accompanying the papers.

that the original paper and the accompanying code implementation only handles the case where each entry in the latent graph is a categorical variable denoting the "edge type". Consequently, we adapt the method and make necessary modifications to the code, so that it can handle numerical values[9]. Besides using the edge-centric graph2trajectory decoder adopted in Kipf et al. (2018); Löwe et al. (2022), we also consider another variant based on the proposed node-centric one. These two benchmarks are referred to as `One-edge` and `One-node`. Note that none of the above-mentioned methods readily handles the multi-entity setting where all graphs are estimated jointly; hence, for comparison purposes, the estimated grand common graph for the competitors is simply obtained by averaging the estimated entity ones.

## 4.1 Data generating mechanisms

The data generating mechanisms used are based on: (1) a linear VAR, (2) a non-linear VAR, and (3) multi-species Lotka-Volterra systems. Two additional mechanisms corresponding to the Lorenz96 and the Springs systems are also considered; their description and results are presented in Appendix B. Consistent with extant notation, $p$ denotes the number of nodes and $M$ the number of entities.

**Linear VAR.** The dynamics of a linear VAR(1) model are determined by $\mathbf{x}_t = A\mathbf{x}_{t-1} + \boldsymbol{\varepsilon}_t$, $\mathbf{x}_t \in \mathbb{R}^p$, wherein $A \in \mathbb{R}^{p \times p}$ is the transition matrix and coincides with the Granger-causal graph; for notational convenience, let $\bar{A} := \bar{\mathbf{z}}$ denote the grand common and $A^{[m]} := \mathbf{z}^{[m]}$ the entity-specific graphs. For this mechanism, we set $p = 30$ and $M = 20$, while the noise term $\boldsymbol{\varepsilon}_t$ has i.i.d entries drawn from a standard Gaussian distribution.

We first discuss the generation of the "initial" common graph $\bar{A}^{(0)}$, whose skeleton $\mathcal{S}_{\bar{A}^{(0)}}$ (i.e., support set) is determined by independent draws from $\mathrm{Ber}(0.1)$; nonzero entries are first drawn from $\mathrm{Unif}(-2,-1) \cup (1,2)$, then scaled so that the spectral radius (i.e., the maximum in absolute value eigenvalue) of $\bar{A}^{(0)}$ is 0.5. Next, we generate perturbations of $\bar{A}^{(0)}$ by "relocating" 10% of the entries (denote their index set by $\mathcal{S}_{\mathrm{ptrb}}$) in $\mathcal{S}_{\bar{A}^{(0)}}$ to random locations in the non-support set $\mathcal{S}^c_{\bar{A}^{(0)}}$. This step generates the corresponding $A^{[m]}$'s. Note that the perturbation mechanism ensures that $\mathcal{S}_{\mathrm{ptrb}} \subset \mathcal{S}_{\bar{A}^{(0)}}$. Further, the positions of the 10% of entries selected at random remain fixed for all $M$ entities, and only the "new" locations are randomly selected and hence differ across the entities, thus inducing heterogeneity across the $A^{[m]}$'s. As a result of the perturbation, for $\bar{A}^{(0)}$, entries in $\mathcal{S}_{\mathrm{ptrb}}$ are essentially "flipped" to zero, and this gives rise to the final grand common graph $\bar{A}$; see also Figure 3a.

**Non-Linear VAR.** For this mechanism, we set $p = 20$ and $M = 10$. We first describe how $\bar{\mathbf{z}}$ and $\mathbf{z}^{[m]}$ are generated, as they dictate the connections and determine how the dynamics are specified. First, let $\bar{\mathbf{z}}^{(0)}$ be the "initial" common graph, set to a banded matrix that has non-zero entries on the diagonal and the adjacent upper and lower diagonals. Next, we perturb $\bar{\mathbf{z}}^{(0)}$ as follows: for all rows not divisible by 3 (e.g., rows, 1, 2, 4, etc.), the two off-diagonal entries are relocated to other positions at random within the same row. This is repeated for all $m$'s to generate $\mathbf{z}^{[m]}$'s. The perturbation creates a zigzag pattern for the final $\bar{\mathbf{z}}$, since whenever a perturbation is present, the original off-diagonal entries on the $\pm 1$ band are guaranteed to get flipped to zero – see Figure 3b for an illustration. Within any entity $m$, response nodes indexed by $i = 2, \cdots, p-1$ have 3 parents; denote their indices by $k_i^1 < k_i^2 < k_i^3$ with subscript $i$ corresponding to the response node id and superscript the parent id, and $k_i^2 \equiv i$ by construction.

The trajectories are generated as follows. For $i = 2, \cdots, p-1$, let $x_{i,t} = 0.25 x_{i,t-1} + \sin(x_{k_i^1,t-1} \cdot x_{k_i^3,t-1}) + \cos(x_{k_i^1,t-1} + x_{k_i^3,t-1}) + \varepsilon_{i,t}$, $\varepsilon_{i,t} \sim \mathcal{N}(0,0.25)$. For the first node $i$ and the last node $p$, their dynamics are slightly different given that they only have one "neighbor"[10]. The choice of such dynamics (in particular, using sine/cosine functions) is somewhat ad-hoc, but aim to induce non-linearities, while ensuring that the system is stable given that these functions are uniformly bounded. Finally, note that we omit the superscript $[m]$ that indexes the entities, as the dynamic specification applies to the dynamical systems of all entities; the parent set for each response node $i$ of entity $m$ is dictated by row $i$ of $\mathbf{z}^{[m]}$.

---

[9]see Appendix A.2 for how the adaptation can be conducted.

[10]For $i = 1$, the dynamics is given by $x_{1,t} = 0.4 x_{1,t-1} - 0.5 x_{2,t-1} + \varepsilon_{1,t}$; for $i = p$, the dynamics is given by $x_{p,t} = 0.4 x_{p,t-1} - 0.5 x_{p-1,t-1} + \varepsilon_{p,t}$

**Multi-species Lotka-Volterra system.** It comprises of coupled ordinary different equations (ODE) that model the population dynamics of multiple predators and preys based on their interactions, specified by the corresponding Granger causal graphs. We consider $p = 20$ and $M = 10$. The $p$ nodes are separated equally into preys and predators (i.e., $\frac{p}{2}$ preys and predators each). Let $\mathbf{x}_t := (\mathbf{u}_t^\top, \mathbf{v}_t^\top)^\top$ with $\mathbf{u}_t := (u_{1,t}, \cdots, u_{p/2,t})^\top \in \mathbb{R}^{p/2}$ and $\mathbf{v}_t := (v_{1,t}, \cdots, v_{p/2,t})^\top \in \mathbb{R}^{p/2}$ denoting the population size of the preys and the predators at time $t$, respectively; $\mathbb{u}_i := \{u_{i,t}\}$ corresponds to the continuous-time trajectory for the $i$th coordinate and $\mathbb{v}_j$ is analogously defined. The dynamics for each coordinate are specified through the following ODE system:

$$\frac{\mathrm{d}\mathbb{u}_i}{\mathrm{d}t} = \alpha\mathbb{u}_i - \beta\mathbb{u}_i(\sum_{j \in \mathcal{P}_i} \mathbb{v}_j) - \alpha(\mathbb{u}_i/\eta)^2; \qquad \frac{\mathrm{d}\mathbb{v}_j}{\mathrm{d}t} = \delta\mathbb{v}_j(\sum_{i \in \mathcal{P}_j} \mathbb{u}_i) - \gamma\mathbb{v}_j. \tag{13}$$

The parameters are set to $\alpha = 1.1$, $\beta = 0.2$, $\gamma = 1.1$, $\delta = 0.2$ and $\eta = 200$. Once again, we omit superscript $[m]$ as this specification applies to all $m = 1, \cdots, M$. The heterogeneity at the entity level is contingent on their graphs $\mathbf{z}^{[m]}$'s that dictate the coupling mechanism; in particular, $\mathcal{P}_i$ and $\mathcal{P}_j$ are the parent set of nodes $i$ and $j$, and are respectively dictated by the support set of the $i$th and $j$th rows of the corresponding $\mathbf{z}^{[m]}$. The generation mechanism of $\bar{\mathbf{z}}$ and $\mathbf{z}^{[m]}$ are described next. The common graph $\bar{\mathbf{z}}$ is generated identically to the one considered in Marcinkevičs & Vogt (2021), where the 20 nodes can be separated into 5 decoupled systems, each containing 2 predators and 2 preys. We add random perturbations to $\bar{\mathbf{z}}$ to arrive at the $\mathbf{z}^{[m]}$'s, by adding additional entries. These additional entries in the upper right/lower left blocks need to be symmetric w.r.t. the diagonal so that the predator-prey correspondence is respected, and they also provide coupling across the originally decoupled $5 \times 4$ systems – see also Figure 3c for an illustration.

## 4.2 Performance evaluation

For all settings, we consider sample sizes of 10K. We run 5 data replicates and report the mean and standard deviation of the AUROC and AUPRC metrics for the competing methods considered. Given that the underlying true Granger-causal graphs in the examined settings are sparse, we also report the best attainable F1 score for each method after thresholding the entries of the group and entity-specific graphs. Results for two other experimental settings, —the Lorenz96 and the Springs systems—, are presented in Appendix B.1. Additional metrics such as true positive rate (TPR), true negative rate (TNR) and accuracy (ACC) based on different thresholding levels are deferred to Appendix B.2, together with visual illustrations of the estimates obtained by good performing competitors.

Table 1 displays the results for all methods. The proposed framework is referred to as Multi-node and Multi-edge, corresponding to the multi-entity joint learning approaches using the node- and edge-centric decoders, respectively; a visualization of the estimated $\bar{\mathbf{z}}$ and $\mathbf{z}^{[1]}, \mathbf{z}^{[2]}$ for illustration purposes is provided in Figure 3 for the former.

The main findings are as follows: (1) the proposed joint-learning approach clearly outperforms its individual learning counterpart (e.g., Multi-node vs. One-node), both at the entity level and the group level (i.e., the common graph). (2) The node-centric decoder consistently outperforms its edge-centric counterpart (e.g., `Multi-node` vs. `Multi-edge`). (3) If one focuses only on individual learning methods, the ones based on prediction models tend to exhibit superior performance (e.g., `GVAR/NGC` vs. `One-node`). In addition, despite the presence of non-linear dynamics, the regularized linear VAR model exhibits surprisingly good performance, especially for the common structure. (4) For practical purposes, post-hoc averaging of the entity-specific Granger causal graphs is reasonably effective for extracting the common structure.

*Remark* 4 (On the robustness with respect to sample size). The proposed joint-learning framework is adequately robust to sample sizes. In particular, in the case where the training sample size reduces to 3000, `Multi-node` shows little degradation in its performance in recovering $\bar{\mathbf{z}}$ (within 1% across all settings in AUROC), and its performance degradation in recovering the entity-level $\mathbf{z}^{[m]}$'s are within 2% for the same metric. On the other hand, `One-node` shows a material deterioration in performance especially for the estimated $\bar{\mathbf{z}}$ (as large as 5% for more challenging settings such as the Lotka-Volterra system), although at the individual entity level, the deterioration is of a smaller magnitude at around 3%. In Appendix B.4 additional comments on the "minimum sample size required" are provided from a practitioner's perspective.

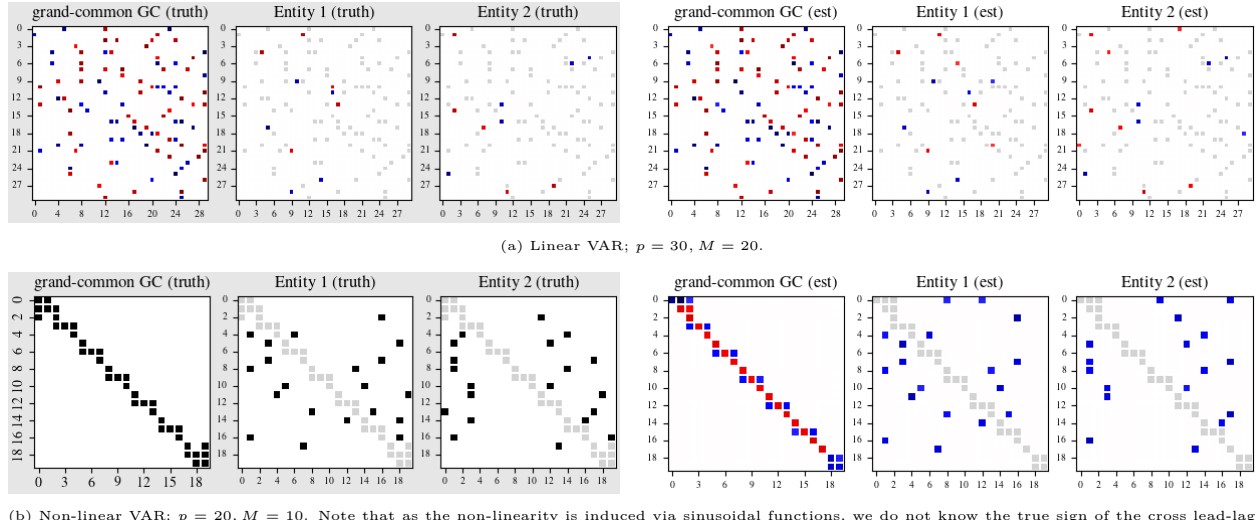

(a) Linear VAR; $p = 30$, $M = 20$.

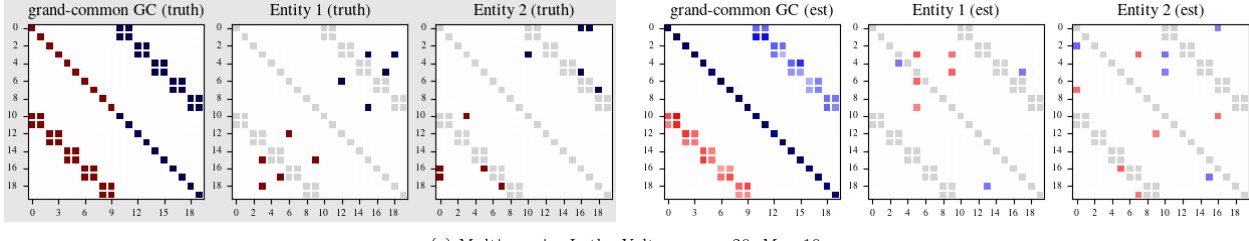

(b) Non-linear VAR; $p = 20$, $M = 10$. Note that as the non-linearity is induced via sinusoidal functions, we do not know the true sign of the cross lead-lag dependency; as such, the entries corresponding to edges that are present are colored in black.

(c) Multi-species Lotka-Volterra; $p = 20$, $M = 10$.

Figure 3: True (shaded panel on the left) and estimated (non-shaded panel on the right) Granger-causal connections using the proposed framework with node-centric decoder (Multi-node); from left to right: $\bar{\mathbf{z}}$, $\mathbf{z}^{[1]}$ and $\mathbf{z}^{[2]}$ and their estimated counterparts. Nonzero entries in $\mathbf{z}^{[1]}, \mathbf{z}^{[2]}$ (and $\widehat{\mathbf{z}}^{[1]}, \widehat{\mathbf{z}}^{[2]}$, resp.) that overlap with those in $\bar{\mathbf{z}}$ ($\widehat{\bar{\mathbf{z}}}$) have been grayed-out so that the idiosyncratic ones stand out.

Table 1: Performance evaluation for the estimated $\bar{\mathbf{z}}$ and $\mathbf{z}^{[m]}$'s: "common" corresponds to $\bar{\mathbf{z}}$ and "entity(avg)" the $\mathbf{z}^{[m]}$'s after averaging the performance metric across $m = 1, \cdots, M$. Numbers are in % and rounded to integers, and correspond to the mean results based on 5 data replicates; standard deviations are reported in the parenthesis.

| | | Generative model-based | | | | Prediction model-based | | | |
|---|---|---|---|---|---|---|---|---|---|
| | | Multi-node | Multi-edge | One-node | One-edge | NGC-cMLP | GVAR | TCDF | Linear |
| **Linear VAR** | | | | | | | | | |
| common | AUROC | 100(0.0) | 100(0.0) | 95(6.6) | 98(4.8) | 100(0.4) | 100(0.0) | 79(2.0) | 100(0.0) |
| | AUPRC | 100(0.0) | 100(0.0) | 83(20.4) | 91(15.9) | 99(1.3) | 100(0.0) | 50(7.6) | 100(0.0) |
| | F1(best) | 100(0.0) | 100(0.0) | 81(17.4) | 88(15.9) | 96(3.5) | 100(0.0) | 52(5.1) | 100(0.0) |
| entity (avg) | AUROC | 100(0.1) | 99(0.6) | 100(0.1) | 100(0.1) | 96(1.8) | 100(0.0) | 77(1.4) | 100(0.0) |
| | AUPRC | 99(0.3) | 95(2.4) | 99(0.2) | 98(0.4) | 86(4.4) | 99(0.1) | 36(5.5) | 100(0.0) |
| | F1(best) | 97(0.8) | 90(3.5) | 96(0.6) | 95(1.0) | 79(4.7) | 99(0.4) | 44(3.4) | 100(0.0) |
| **Non-linear VAR** | | | | | | | | | |
| common | AUROC | 99(0.2) | 82(1.7) | 97(0.2) | 93(0.8) | 90(0.7) | 99(0.1) | 75(1.0) | 99(0.1) |
| | AUPRC | 96(0.9) | 58(1.1) | 80(0.8) | 80(8.0) | 64(1.1) | 98(0.2) | 53(0.5) | 98(0.1) |
| | F1(best) | 94(0.6) | 60(0.7) | 74(1.0) | 83(6.9) | 61(0.9) | 98(0.7) | 56(1.2) | 98(0.7) |
| entity (avg) | AUROC | 98(0.3) | 85(0.9) | 94(0.4) | 95(0.5) | 94(0.5) | 99(0.3) | 73(0.9) | 96(0.7) |
| | AUPRC | 93(1.0) | 75(0.8) | 76(0.2) | 89(0.6) | 87(0.6) | 96(0.6) | 44(1.8) | 96(0.7) |
| | F1(best) | 86(1.5) | 73(1.0) | 70(0.3) | 86(0.8) | 82(0.4) | 91(0.8) | 50(1.5) | 97(0.6) |
| **Lotka-Volterra** | | | | | | | | | |
| common | AUROC | 100(0.0) | 100(0.0) | 97(1.1) | 87(8.4) | 100(0.0) | 100(0.0) | 79(0.8) | 100(0.1) |
| | AUPRC | 100(0.0) | 100(0.1) | 92(3.0) | 73(10.5) | 100(0.0) | 100(0.0) | 58(1.2) | 100(0.4) |
| | F1(best) | 100(0.7) | 99(0.8) | 87(5.4) | 69(9.0) | 100(0.4) | 97(1.2) | 53(1.4) | 94(3.5) |
| entity (avg) | AUROC | 89(1.0) | 84(1.3) | 83(1.6) | 75(1.3) | 92(1.0) | 93(0.6) | 72(0.8) | 77(1.0) |
| | AUPRC | 80(1.5) | 70(2.0) | 69(1.8) | 51(2.6) | 87(1.2) | 89(1.0) | 41(1.0) | 71(1.2) |
| | F1(best) | 74(1.4) | 65(2.0) | 63(1.4) | 53(2.2) | 84(0.8) | 84(0.7) | 46(0.3) | 71(0.7) |

Finally, we remark that `GVAR` exhibits consistently strong performance amongst the methods under consideration. On the other hand, it is observed during evaluation time that given the magnitude of the estimated entries, the quality of the graph skeleton is sensitive to the exact choice of the thresholding level, whereas the proposed framework is more robust. This has implications on the difficulty of choosing a good threshold in practice — see also Table 4 and additional discussion and remarks in Appendix B.2.

## 5 Application to a Multi-Subject EEG Dataset

The dataset in consideration corresponds to electroencephalogram (EEG) measurements obtained from 72 active electrodes placed on the scalp of 22 subjects (entities), and they are publicly available; see Trujillo et al. (2017). Prior investigation on this dataset primarily centers around understanding the information provided by different connectivity measures that are available in the literature, rather than the connectivity patterns themselves.

The EEG experiment pertains to a stimulus procedure performed on the subjects comprising of 1-min interleaved sessions with eyes open (EO) or closed (EC). Such experiments aim to provide insights into the brain's functional segregation and integration (Barry et al., 2007; Rubinov & Sporns, 2010; Miraglia et al., 2016). Note that (1) the experiment is integrated, but the data are collated separately for the eyes-open and the eyes-closed interleaving sessions, which results in two datasets (EO and EC, respectively); and (2) due to the design of the experiment, the dynamics governing the data within the EO sessions (respectively, EC sessions) are stable and stay largely unchanged.

We select to analyze the data from 31 specific EEG channels (and hence $p = 31$) located at the back of the scalp (see Figure 4), where the primary visual cortex is located. For both datasets, we restrict the analysis to entities that have at least 40000 observations (total number of time points)[11], and the whole trajectory is further partitioned into training/validation data, with the latter having 2000 time points. Here the validation data is used to select the best hyperparameters such that the reconstruction or prediction error is minimum over the search grid, depending on the method. Four methods are considered, including the proposed joint-learning one with a node-centric decoder (`Multi-node`), its individual-learning counterpart (`One-node`) and prediction model-based `GVAR` and `NGC`.

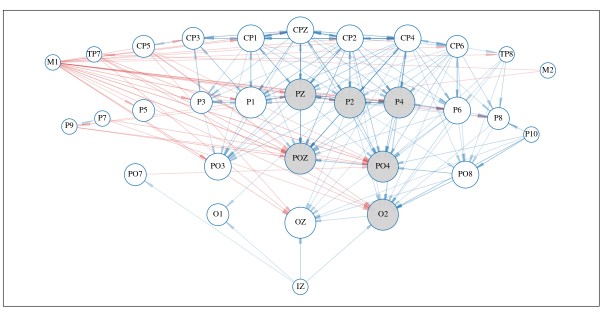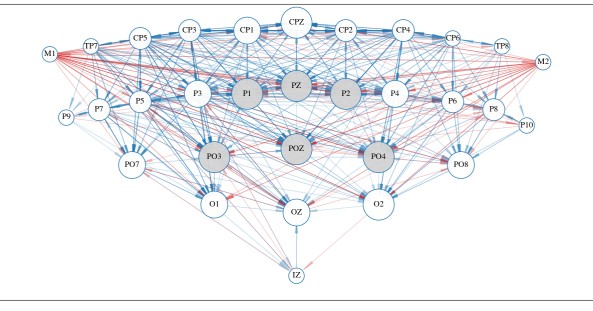

(a) Eyes Open (EO)

(b) Eyes Closed (EC)

Figure 4: `Multi-node` results: estimated **common** Granger-causal connections for EO (left panel) and EC (right panel) after normalization and subsequent thresholding at 0.15. Red edges correspond to positive connections and blue edges correspond to negative ones; the transparency of the edges is proportional to the strength of the connection. Larger node sizes correspond to **higher in-degree** (incoming connectivity), and the top 6 nodes are colored in gray.

The estimated common Granger-causal connections based on `Multi-node` and `GVAR` are depicted in Figures 4 and 5, respectively. The results based on `One-node` and `NCG` are delegated to Appendix D[12]. For all methods, we threshold the raw estimates to remove very small entries; the thresholding values are chosen so that each

---

[11]This restriction has reduced the number of entities to 21 for the EO dataset while the number of entities for the EC dataset remains at 22.

[12]Note that the Granger causal connections estimated by `Multi-node` and `One-node` are up to a "complete sign flip" (see, e.g., discussion in Appendix E); nonetheless, these methods are effective in distinguishing positive (negative) connections from negative (positive) ones. Further, `NCG` does not provide signed estimates (positive/negative) of the Granger causal connections, unlike the other three methods.

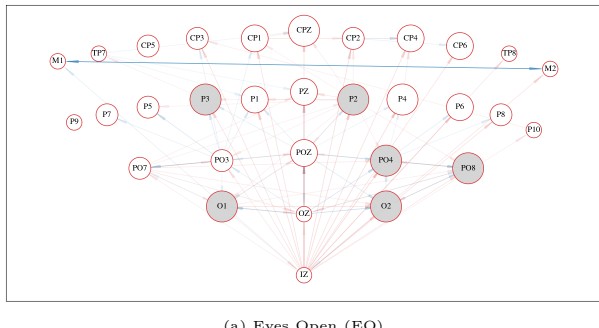 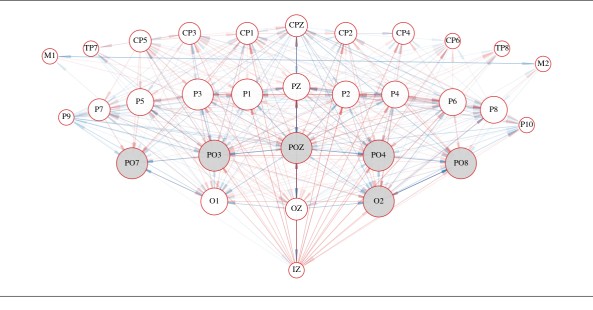

(a) Eyes Open (EO)  (b) Eyes Closed (EC)

Figure 5: `GVAR` results: estimated **common** Granger-causal connections for EO (left panel) and EC (right panel) after normalization and subsequent thresholding at 0.05. Red edges correspond to positive connections and blue edges correspond to negative ones; the transparency of the edges is proportional to the strength of the connection. Larger node sizes correspond to **higher in-degree** (incoming connectivity), and the top 6 nodes are colored in gray.

method has around 400 total number of edges for the EC session to facilitate comparisons across them. Results from these methods exhibit commonalities and differences, as discussed next.

The following common observations are noted across most methods: (1) based on the results by `Multi-node`, `GVAR` and `One-node`, the overall Granger causal connectivity level is markedly higher for the EC session compared to the EO one; this is consistent with results in studies in the literature (Barry et al., 2007; Marx et al., 2004; Das et al., 2016; Trujillo et al., 2017), albeit using different connectivity measures. On the other hand, results from `NCG` show the reverse pattern, i.e., higher connectivity for the EO session compared to the EC session. (2) For both the EO and EC sessions, the in-degree of nodes in the mid-line channels (i.e. OZ, POZ, PZ, CPZ) tends to be higher than that of the nodes to the left and right parts of the brain. This is broadly comparable to results in the literature—see, e.g., Barry et al. (2007) for adult subjects and Barry et al. (2009) for children, though the problem under consideration and thus the analysis is different in their work. (3) As it is observed in `Multi-node`, `One-Node`, and `NGC`, the OZ channel exhibits different degree of connectivity for the EO and the EC sessions; in particular, it is Granger causal for many other channels in the former, i.e., being the emitter of edges and exhibit higher node out-degree; this becomes significantly less so in the latter—see also Hatton et al. (2023).

The four methods also exhibit certain discrepancies in their results. (1) As mentioned above, the EO session exhibits an overall decrease in connectivity when compared against the EC session. The drop in connectivity, however, is not uniform across nodes on the left and the right parts of the brain. This is reported by generative model-based methods `Multi-node` and `One-node`, and a result also mentioned in the literature (Barry et al., 2007; 2009; Modarres et al., 2023). Such discrepancy—in terms of the differential change in connectivity level between the nodes on the left and those on the right—is significantly less pronounced in `GVAR` and `NGC`. (2) A strong bi-directional Granger causal link between channels M1 and M2 in both the EO and EC sessions is observed according to `GVAR`. This strong connection is somewhat harder to interpret, since these two channels correspond to mastoid (behind the ears) locations and their connectivity is customarily modulated through the midline positioned ones (OZ, PZ, POZ, CPZ) (Das et al., 2022). (3) As a minor remark, for `GVAR`, we observe strong autoregressive connections (i.e., dominant diagonals in the estimates)[13]; for `NGC`, the overall connectivity level in the raw estimates is significantly higher and thus requires stronger thresholding. Both observations are also noted in selected synthetic data experiments.

In summary, all methods with the exception of `NCG` are in agreement regarding the decrease in Granger causal connections from the EC to the EO session. There is also concordance across methods regarding the observation that this decrease is not uniform across the left and right parts of the brain. Both of these results are in accordance to previous ones in the literature, although based on different analysis techniques and connectivity measures.

---

[13]this is not shown in the plot (to avoid self-loops) for aesthetic purposes. Note also that visually, the edges are overall more "faint" in the plot, as a result of the dominant diagonals and the corresponding normalization.

## 6   Discussion

This paper proposes a multi-layer VAE-based framework for jointly estimating the group and entity-level Granger-causal graphs, in the presence of connectivity heterogeneity across entities. The framework is based on a hierarchical generative structure that couples the group and entity-specific graphs. The model is learned via an end-to-end encoding-decoding procedure that minimizes the negative ELBO loss. The results of the numerical experiments show that the performance of the proposed framework is broadly robust to sample size, especially for the common graph. Further, the joint learning paradigm has a clear advantage over its "individual learning" generative model-based counterpart, which then leads to more accurate quantification for both the common connectivity patterns and the idiosyncratic ones. This advantage becomes more pronounced in settings where one has limited sample size and large collections of related systems. In addition, the joint learning paradigm can be useful in situations, where one may be interested in detecting "outlier" dynamical systems in the collection under consideration, or in identifying clusters of such systems. These tasks can be accomplished by close examination and analysis of the entity specific graphs.

Although "prediction models plus post-hoc aggregation" heuristics can sometimes exhibit competitive performance, the embedded common structure across entities is completely neglected at the formulation level. In addition, existing models within this framework are also limited to scalar-valued nodes, partly due to their reliance on performing ad-hoc extraction/aggregation on intermediate quantities (e.g., neural network weight matrices during training) to infer the Granger causality.

In the presence of non-linearity, a key advantage of generative model-based approaches is that the Granger-causal relationships are solely encoded through the latent graph that serves as the gateway for information propagation. This provides a clean way to model relationships between connectivity patterns — either statically or dynamically. The setting considered in this work is a static one, and the type of such relationship manifests as common-idiosyncratic connectivity patterns. A potential extension to the generative process under consideration, suitable for more complex real-world dynamical systems, is to allow for time-varying connectivity patterns. For example, Graber & Schwing (2020) extends the work in Kipf et al. (2018) to a dynamic setting. With appropriate modifications to the proposed approach, such as expanding the conditional relationship of the graphs dictated in (2) so that they also depend on their past, this modeling task can be handled in a straightforward manner.

### Code and Data Availability

The code repository is available at *https://github.com/georgemichailidis/vae-multi-level-neural-GC-official*. The multi-subject EEG dataset is available at *https://dataverse.tdl.org/dataverse/rsed2017*, as provided by Trujillo et al. (2017).

### Acknowledgements

The authors thank the Action Editor and three anonymous reviewers for their careful review of the work, and their constructive comments and suggestions.

George Michailidis was supported in part by NSF grants DMS 2348640 and DMS 2334735.

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

# A    Additional Modeling and Implementation Details

In this section, we provide a description for some additional modeling details. In Sections A.1 and A.2, we omit superscript $[m]$ that indexes the entities whenever there is no ambiguity, as the descriptions therein apply to all $m$'s independently unless otherwise specified.

## A.1    Encoder

We provide details for the encoder sub-module that is abstracted as $f_{x \to h}$, wherein based on the node trajectories, one obtains the hidden representations for the edges $\{\mathbf{h}_{ij}\} := f_{x \to h}(\mathbf{x})$; see also Section 3.2, module (enc-a).

As the most basic building blocks of message-passing operations, "node2edge" and "edge2node" operate based off a *complete* graph, and can be generically represented as:

$$e_{ij} \leftarrow \mathrm{concat}(x_i, x_j) \quad \text{(node2edge)}; \qquad x_i \leftarrow \sum_j e_{ij} \quad \text{(edge2node)},$$

with $x_i$ denoting the node representation and $e_{ij}$ the edge one. $f_{x \to h}$ is then parameterized through the $L$ passes of such operations:

$$\begin{aligned}
(\text{init emb}): \quad & \check{\mathbf{x}}_i^{(0)} \leftarrow \mathrm{emb}(\mathbf{x}_i), \quad \forall i = 1, \cdots, p \\
\check{\mathbf{x}} \to \mathbf{e}: \quad & e_{ij}^{(l)} \leftarrow \mathrm{MLP}\big(\mathrm{node2edge}(\check{\mathbf{x}}_i^{(l-1)}, \check{\mathbf{x}}_j^{(l-1)})\big); \quad l = 1, \cdots, L \\
\mathbf{e} \to \check{\mathbf{x}}: \quad & \check{\mathbf{x}}_i^{(l-1)} \leftarrow \mathrm{MLP}\big(\mathrm{edge2node}(e_{ij}^{(l)}; j = 1, \cdots, p)\big); \quad l = 2, \cdots, L
\end{aligned}$$

Here $\mathbf{x}_i$ corresponds to the trajectory of node $i$ over time, that is, $\mathbf{x}_i = (x_{i,1}, \cdots, x_{i,T})$ and the final hidden representation is given by $\mathbf{h}_{ij} := e_{ij}^{(L)}$, $i, j = 1, \cdots, p$.

Concretely, the embedding module can be as simple as entailing only Linear-ReLU type operations; the input trajectory $\mathbf{x}_i$ of a node $i$, $\forall i \in \{1, \cdots, p\}$, is processed via the following steps outlined in Figure 6:

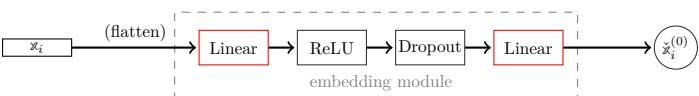

Figure 6: Example for the embedding operation in the encoder according to MLP style. Blocks with trainable parameters are outlined in red. Note that the flattening step is only required when the nodes are vector-valued.

Note that this also coincides with the `LR`-type embedding functions in Gorishniy et al. (2022). In regards to the MLP block, it is obtained by stacking the sub-blocks as illustrated in Figure 7.

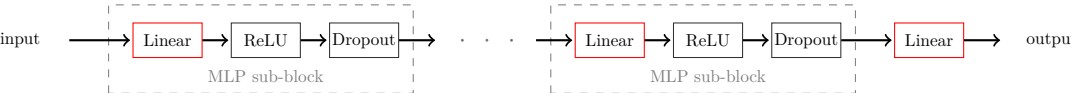

Figure 7: An MLP block obtained by stacking the sub-blocks. Constituent blocks with trainable parameters are outlined in red.

Figure 8 provides a pictorial illustration for the sequential operations entailed in the Trajectory2Graph encoder module.[14] Note that this is effectively the `MLPEncoder` used in Kipf et al. (2018) and the description is given here for the sake of completeness. We refer interested readers to Kipf et al. (2018) for some other encoders considered therein.

---

[14]In our experiments, all the MLP blocks used in the Trajectory2Graph operations are kept simple with only one single sub-block; the hidden dimension is set at 128 or 256, depending on the exact experiments.

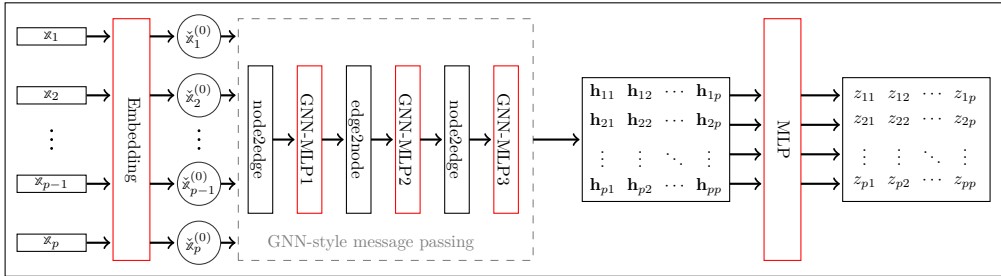

Figure 8: Diagram for the Trajectory2Graph encoder operations. Blocks with trainable parameters are outlined in red.

## A.2 Decoder

We divide this subsection into two parts, that respectively (1) discuss how the structure adopted in a node-centric Graph2trajectory module described in Section 3.2 can readily accommodate the presence of dependence on more than 1 lags; and (2) provide a brief discussion on how the original edge-centric decoder adopted in Kipf et al. (2018); Löwe et al. (2022) can be revised to adapt to the case of a numerical graph, and compare it with the node-centric one, although architectural choices are not the focus of this paper.

**Extension to multiple lag dependency.** The extension of a node-centric decoder to accommodate the presence of more than 1 lags (i.e., $q > 1$) is straightforward, largely due to the fact that the node value at time $t - 1$, denoted by $x_{j,t-1}$ is not limited to be scalar-valued in the first place. In the case of $q$-lag dependency, one can simply replace $x_{j,t-1}$ by concat$(x_{j,t-1}, \cdots, x_{j,t-q})$ and proceed with the remainder of the operations as outlined in (9) and (10). In particular, with the presence of more lags, as an alternative to a (optional) numerical embedding step, one can instead consider 1D-CNN as a preprocessing module on the "new" $x_{j,t-1}$, before an element-wise gate represented by $z_{ij}$ is applied to control the information flow.

**Adaptation of the edge-centric decoder.** The original edge-centric decoder adopted in Kipf et al. (2018) handles the case where each entry in $z_{ij}$ corresponds to an edge type (categorical), and it entails the following operations:

1. node2edge for each time step, that is $e_{ij,t-1} := \text{concat}(x_{i,t-1}, x_{j,t-1})$ to arrive at the edge representation at time $t - 1$;

2. for *each* edge type of interest, run $e_{ij,t-1}$'s through its corresponding edge type-specific function (e.g., MLP) to get the "enriched" representation $\check{e}_{ij,t-1}$;

3. aggregate the enriched edge representations back to nodes via an edge2node operation, giving rise to $\mathbf{v}_{i,t-1}$'s, $i = 1, \cdots, p$; $\mathbf{v}_{i,t-1}$ then serves as the predictor for time-$t$ response $x_{i,t}$.

In order for the above module to accommodate the case of a numeric $z_{ij}$, the following simple modification to step 2 is introduced:

2' run $e_{ij,t-1}$'s through some function (e.g., MLP) to get the "enriched" representation $\check{e}_{ij,t-1}$, and further update it through a gating mechanism as dictated by $z_{ij}$, that is, $\check{e}_{ij,t-1} \leftarrow \check{e}_{ij,t-1} \circ z_{ij}$.

The information propagation path from node $j$ to $i$ can be represented as:

$$x_{j,t-1} \overset{\text{node2edge}}{\to} e_{ij,t-1} \overset{\text{MLP}}{\to} \check{e}_{ij,t-1} \overset{\text{gating}}{\to} \check{e}_{ij,t-1} \circ z_{ij} \overset{\text{edge2node}}{\to} \mathbf{v}_{i,t-1} \to x_{i,t}; \qquad (14)$$

one can easily verify that for $z_{ij} = 0$, there is no path from $x_{j,t-1}$ to $x_{i,t}$.

As a final remark, for the node-centric decoder, the gating through $z_{ij}$ directly operates on the node representation, and the path is given by

$$x_{j,t-1} \overset{\text{emb}}{\to} \check{x}_{j,t-1} \overset{\text{gating}}{\to} \check{x}_{j,t-1} \circ z_{ij} \overset{\text{element of}}{\to} \mathbf{u}_{i,t-1} \to x_{i,t};$$

see also Figure 9 for a pictorial illustration.

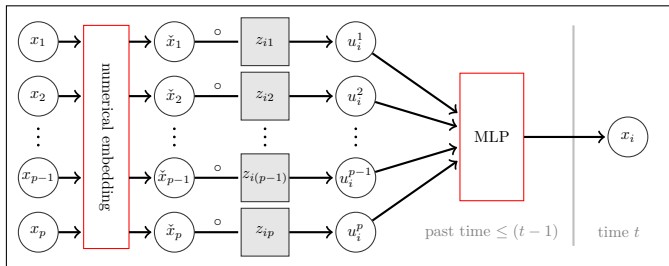

Figure 9: Diagram for the node-centric Graph2Trajectory Decoder operations, with node $i$ being the response node in this illustration. The corresponding entries of $\mathbf{z}$'s (shaded in gray, obtained by sampling and is fed into the Graph2Trajectory Decoder as input) perform the gating operation (denoted by $\circ$). Blocks with trainable parameters are outlined in red and are shared across all response nodes $i = 1, \cdots, p$.

To contrast, for the edge-centric decoder, as indicated in (14), entries in $z_{ij}$ determine the lead-lag information passing from $j \to i$ via $e_{ij,t-1}$, and therefore such a gating mechanism is somewhat circumstantial.

### A.3  Loss calculation

A derivation of (12) is given next.

$$
\mathrm{KL}\Big(q_\phi(\mathcal{Z}|\mathcal{X}) \,\big\|\, p_\theta(\mathcal{Z})\Big) = \mathbb{E}_{q_\phi(\mathcal{Z}|\mathcal{X})} \log\left[\frac{q_\phi(\mathcal{Z}|\mathcal{X})}{p_\theta(\mathcal{Z})}\right] = \mathbb{E}_{q_\phi(\mathcal{Z}|\mathcal{X})}\left[\log\frac{q_\phi(\bar{\mathbf{z}}|\{\mathbf{z}^{[m]}\})}{p_\theta(\bar{\mathbf{z}})} + \log\frac{q_\phi(\{\mathbf{z}^{[m]}\}|\{\mathbb{x}^{[m]}\})}{p_\theta(\{\mathbf{z}^{[m]}\}|\bar{\mathbf{z}})}\right]
$$

$$
= \iint q_\phi\big(\bar{\mathbf{z}}|\{\mathbf{z}^{[m]}\}\big)q_\phi\big(\{\mathbf{z}^{[m]}\}|\{\mathbb{x}^{[m]}\}\big) \log\left[\frac{q_\phi\big(\bar{\mathbf{z}}|\{\mathbf{z}^{[m]}\}\big)}{p_\theta\big(\bar{\mathbf{z}}\big)}\right] \mathrm{d}\bar{\mathbf{z}} \mathrm{d}\{\mathbf{z}^{[m]}\}
$$

$$
+ \iint q_\phi\big(\bar{\mathbf{z}}|\{\mathbf{z}^{[m]}\}\big)q_\phi\big(\{\mathbf{z}^{[m]}\}|\{\mathbb{x}^{[m]}\}\big) \log\left[\frac{q_\phi\big(\{\mathbf{z}^{[m]}\}|\{\mathbb{x}^{[m]}\}\big)}{p_\theta\big(\{\mathbf{z}^{[m]}\}|\bar{\mathbf{z}}\big)}\right] \mathrm{d}\bar{\mathbf{z}} \mathrm{d}\{\mathbf{z}^{[m]}\}
$$

$$
= \int q_\phi(\{\mathbf{z}^{[m]}\}|\{\mathbb{x}^{[m]}\}) \underbrace{\left\{\int q_\phi\big(\bar{\mathbf{z}}|\{\mathbf{z}^{[m]}\}\big) \log\left[\frac{q_\phi\big(\bar{\mathbf{z}}|\{\mathbf{z}^{[m]}\}\big)}{p_\theta\big(\bar{\mathbf{z}}\big)}\right] \mathrm{d}\bar{\mathbf{z}}\right\}}_{\mathrm{KL}\Big(q_\phi(\bar{\mathbf{z}}|\{\mathbf{z}^{[m]}\}) \,\big\|\, p_\theta(\bar{\mathbf{z}})\Big)} \mathrm{d}\{\mathbf{z}^{[m]}\}
$$

$$
+ \int q_\phi\big(\bar{\mathbf{z}}|\{\mathbf{z}^{[m]}\}, \{\mathbb{x}^{[m]}\}\big) \underbrace{\left\{\int q_\phi\big(\{\mathbf{z}^{[m]}\}|\{\mathbb{x}^{[m]}\}\big) \log\left[\frac{q_\phi\big(\{\mathbf{z}^{[m]}\}|\{\mathbb{x}^{[m]}\}\big)}{p_\theta\big(\{\mathbf{z}^{[m]}\}|\bar{\mathbf{z}}\big)}\right] \mathrm{d}\{\mathbf{z}^{[m]}\}\right\}}_{\mathrm{KL}\Big(q_\phi(\{\mathbf{z}^{[m]}\}|\{\mathbb{x}^{[m]}\}) \,\big\|\, p_\theta(\{\mathbf{z}^{[m]}\}|\bar{\mathbf{z}})\Big)} \mathrm{d}\bar{\mathbf{z}}
$$

$$
\overset{(a)}{=} \mathbb{E}_{q_\phi(\{\mathbf{z}^{[m]}\}|\mathcal{X})}\left[\mathrm{KL}\Big(q_\phi(\bar{\mathbf{z}}|\{\mathbf{z}^{[m]}\}) \,\big\|\, p_\theta(\bar{\mathbf{z}})\Big)\right] + \mathbb{E}_{q_\phi(\bar{\mathbf{z}}|\mathcal{X})}\left[\mathrm{KL}\Big(q_\phi(\{\mathbf{z}^{[m]}\}|\{\mathbb{x}^{[m]}\}) \,\big\|\, p_\theta(\{\mathbf{z}^{[m]}\}|\bar{\mathbf{z}})\Big)\right].
$$

For (a), the first term is straightforward, the second term goes through since

$$
\int p(x|y,z)\left\{\int p(y|z) \log\frac{p(y|z)}{q(y|x)} \mathrm{d}y\right\} \mathrm{d}x = \iint p(y|z)p(x|y,z) \log\frac{p(y|z)}{q(y|x)} \mathrm{d}x \mathrm{d}y
$$

$$
= \mathbb{E}_{Y|Z}\mathbb{E}_{X|Z,Y} \log\frac{p(y|z)}{q(y|x)} = \mathbb{E}_{Y|Z}\mathbb{E}_{X|Z} \log\frac{p(y|z)}{q(y|x)} = \mathbb{E}_{X|Z}\left[\mathbb{E}_{Y|Z} \log\frac{p(y|z)}{q(y|x)}\right];
$$

and the last equality holds as a result of the Fubini-Tonelli theorem.

### A.4  Evaluating the predictive strength of Granger causal relationships

Next, we briefly discuss how the trained decoder can be used to measure the predictive strength of the Granger causal connections.

Once the model is trained, using the inference procedure described in Section 3.3, one obtains estimates $\hat{z}^{[m]}$ for all entity-specific graphs. Further, a trained Graph2Trajectory module, abstracted as $\hat{g}_{z\to x}$, also becomes

available. The predictive strength of any connection entry $(i, j)$ — corresponding to the lead-lag relationship from $j$ to $i$ — can then be assessed by *nullifying* the corresponding entry. Throughout the remainder of the discussion, we omit superscript $[m]$ for ease of presentation, as the procedure is applicable to an arbitrary entity of interest.

Let $\tilde{\mathbf{z}}^{(ij)}$ be identical to $\hat{\mathbf{z}}$ except that the $(i, j)$ entry is set to zero (nullified). The reconstructed trajectories, based on the estimated and the nullified graphs are given by $\hat{\mathbf{x}} = \hat{g}_{z \to x}(\hat{\mathbf{z}}, \mathbf{x}_1)^{15}$ and $\tilde{\mathbf{x}}^{(ij)} = \hat{g}_{z \to x}(\tilde{\mathbf{z}}^{(ij)}, \mathbf{x}_1)$, respectively. The predictive strength can then be evaluated based on the difference in the residual-sum-of-squares (RSS), with the latter obtained by evaluating the reconstructed trajectory against the observed values. Concretely, $\text{RSS}(\hat{\mathbf{x}})$ can be obtained by $\frac{1}{T-1} \sum_{t=2}^{T} \|\mathbf{x}_t - \hat{\mathbf{x}}_t\|^2$ and that for $\tilde{\mathbf{x}}^{(ij)}$ can be analogously obtained; the predictive strength of the $(i, j)$ connection can then be calculated as $\text{RSS}(\hat{\mathbf{x}}) - \text{RSS}(\tilde{\mathbf{x}}^{(ij)})$. This procedure can be generalized to a set of connections, where instead of nullifying a single entry, multiple entries are nullified simultaneously and the remainder of the evaluation follows. Note that the proposed procedure resembles that of testing for the presence/absence of Granger causality in linear VAR models, where an F-test is used (Geweke, 1984). The calculated difference $\text{RSS}(\hat{\mathbf{x}}) - \text{RSS}(\tilde{\mathbf{x}}^{(ij)})$ also appears in the numerator of the aforementioned F-statistic.

## A.5 Construction of samples

We briefly explain how samples are constructed from observed data trajectories. We omit the superscript $[m]$ that corresponds to the entity ID, since the construction is generally applicable.

The available data can either correspond to a collection of long trajectories (e.g., traditional time series setting where observations for different variables are collected over time), or to multiple collections of (long) trajectories, where each collection corresponds to temporal observations over time from repeated measurements (e.g., in the context of a neurophysiological experiment, a subject is exposed to a stimulus (eyes open or eyes close) a number of times). In both cases, the trajectories are parsed into shorter ones of length $T$, which is the context window considered in the modeling. Concretely, let $\{\mathbf{x}_0, \mathbf{x}_1, \cdots, \mathbf{x}_{\widetilde{T}}\}$ be the long trajectory, with $\widetilde{T}$ denoting the total number of observations. The samples, indexed by $n$, are shorter trajectories of length $T$, with each consisting of observations $\mathcal{X}^{(n)} := \{\mathbf{x}_{sn}, \mathbf{x}_{sn+1}, \cdots, \mathbf{x}_{sn+T-1}\}$, where $s$ is the stride size that dictates the overlapping between samples with adjacent indices. A long trajectory of length $\widetilde{T}$ gives rise to $\lfloor (\widetilde{T} - T)/s + 1 \rfloor$ samples, which are then used during mini-batch training.

# B Additional Synthetic Data Experiments and Results

## B.1 Lorenz96 and Springs5 experiments

To explore the applicability of the proposed framework to selected special cases, there are two other settings considered in our synthetic data experiments: the Lorenz96 and the Springs5 systems. Unlike the settings presented in the numerical experiments in Section 4 wherein the entity-level heterogeneity manifests itself primarily in the form of perturbations to the skeleton of the shared common graph, for these two systems, the entity-specific skeletons are either identical across all $M$ entities and only the magnitude of the entries changes (Lorenz96), or they manifest their heterogeneity through a probabilistic mechanism (Springs), as explained in the sequel.

Similar to those presented earlier, for both settings, we run the experiments on 5 data replicates and report the metrics after averaging across the 5 runs, with their respective standard deviation included in the parentheses.

---

[15]Recall that throughout the main sections, we use $\mathbf{x} := \{\mathbf{x}_1, \cdots, \mathbf{x}_T\}$ to denote the trajectory; here $\hat{\mathbf{x}}$ is effectively its "reconstructed" couterpart.

### B.1.1 The Lorenz96 system

The Lorenz96 system (Lorenz, 1996) has been previously investigated in Tank et al. (2021); Marcinkevičs & Vogt (2021). The dynamics for a $p$-variable system evolve according to the following ODE:

$$\frac{d\mathbb{x}_i}{dt} = (\mathbb{x}_{i+1} - \mathbb{x}_{i-2})\mathbb{x}_{i-1} - \mathbb{x}_i + F, \qquad i = 1, \cdots, p, \tag{15}$$

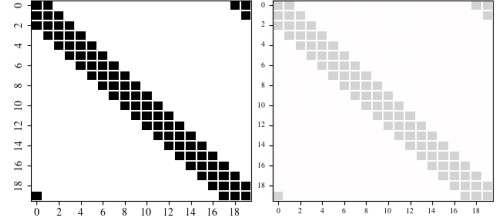

where $\mathbb{x}_i := \{x_{i,t}\}$ denotes the continuous time trajectory of node $i$ with $\mathbb{x}_0 := \mathbb{x}_p, \mathbb{x}_{-1} := \mathbb{x}_{p-1}$ and $\mathbb{x}_{p+1} := \mathbb{x}_1$. Such a system corresponds to a Granger-causal structure shown in Figure 10 that depicts its skeleton. The representation in (15) can be obtained from Kerin & Engler (2022):

$$\frac{d\mathbb{x}_i}{dt} = \alpha(\mathbb{x}_{i+1} - \mathbb{x}_{i-2})\mathbb{x}_{i-1} - \beta\mathbb{x}_i + \gamma, \tag{16}$$

Figure 10: Lorenz96: $\bar{\mathbf{z}}$ and $\mathbf{z}^{[0]}$, showing only the skeleton.

by reparameterizing $\alpha = \beta, \lambda = \alpha/\beta$ and setting $F = \alpha\gamma/\beta^2$. $F$ is the forcing constant that controls the degree of non-linearity; in particular, given the relationship between (15) and (16), as $F$ varies, the *strength* of the Granger-causality changes despite an invariant skeleton. In other words, to induce heterogeneity across entities, we can only change the parameter $F$ that induces heterogeneity in the magnitudes of the Granger causal connections, while the skeleton of the Granger causal graph remains the same. We consider a setting with $p = 20$ and $M = 5$ entities, with the forces taking the following values: $F \in \{10.0, 17.5, 25.0, 32.5, 40.0\}$.

Table 2: Performance evaluation for the estimated $\bar{\mathbf{z}}$ and $\mathbf{z}^{[m]}$'s for setting Lorenz96. Numbers are in % and rounded to integers, and correspond to the mean results based on 5 data replicates; standard deviations are reported in the parentheses.

| | | Generative model-based | | | | Prediction model-based | | | |
|---|---|---|---|---|---|---|---|---|---|
| | | Multi-node | Multi-edge | One-node | One-edge | NGC-cMLP | GVAR | TCDF | Linear |
| common | AUROC | 100(0.1) | 100(0.7) | 100(0.1) | 90(19.7) | 97(0.0) | 100(0.1) | 82(0.9) | 99(0.1) |
| | AUPRC | 100(0.4) | 99(1.6) | 100(0.3) | 82(32.5) | 87(0.1) | 100(0.2) | 65(0.9) | 97(0.5) |
| | F1(best) | 97(1.5) | 96(3.4) | 97(1.3) | 80(25.7) | 87(0.8) | 98(1.0) | 59(1.3) | 89(0.2) |
| entity | AUROC | 95(1.3) | 85(3.7) | 96(1.0) | 88(1.9) | 96(0.1) | 97(0.8) | 79(0.8) | 99(0.1) |
| (avg) | AUPRC | 89(2.3) | 76(4.6) | 91(2.0) | 78(2.9) | 85(0.3) | 90(1.5) | 62(0.7) | 96(0.3) |
| | F1(best) | 82(3.2) | 71(3.5) | 84(2.6) | 72(3.1) | 83(0.4) | 83(0.2) | 58(0.5) | 88(0.3) |

The results are shown in Table 2 and the main findings are: (1) consistent with the results in Section 4, the node-centric decoder outperforms the edge-centric one; (2) the proposed joint-learning approach `Multi-node` matches the performance of `GVAR` and outperforms all other competitors for the common graph; (3) for the entity-specific graphs, interestingly, the linear VAR exhibits a slight edge over all competing methods, while the performance of the proposed model is broadly on-par with the remaining competitors.

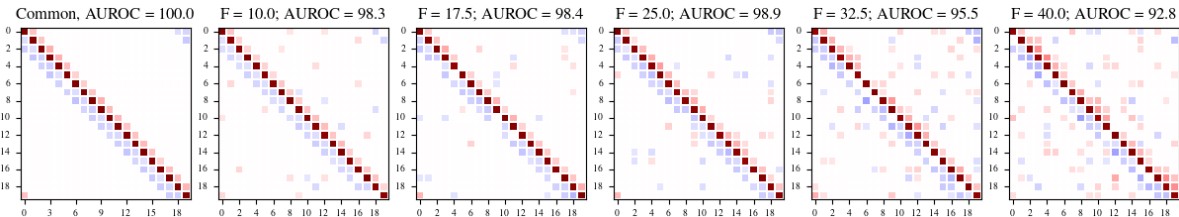

Figure 11: Estimated $\bar{\mathbf{z}}$ and $\mathbf{z}^{[m]}$'s with different $F$'s using the proposed joint-learning framework with a node-centric decoder (`Multi-node`).

Finally, the common and the five entity-specific Granger causal graphs for the `Multi-node` method are depicted in Figure 11. It can be seen that the performance deteriorates for systems with larger external force $F$.

### B.1.2 Springs5 system

This setting is investigated in Kipf et al. (2018); Löwe et al. (2022), and in this work we consider a "multi-entity" version of it. In the original setting, particles (i.e., nodes) are connected (pairwise) by springs at random with probability 0.5; in the case where the connection between particles $i$ and $j$ is present, they interact according to Hooke's law $F_{ij} = -k(r_i - r_j)$, where $F_{ij}$ is the force applied to particle $i$ by particle $j$, $k$ is the spring constant and $r_i$ is the location vector of particle $i$ in 2-dimensional space. With some initial location and velocity, the trajectories can be simulated by solving Newton's equations of motion (see also Kipf et al. (2018), Appendix B for details). Crucially, (1) the Granger-causal graph is essentially a realization of the homogeneous Erdős-Rényi graph (Erdős & Rényi,

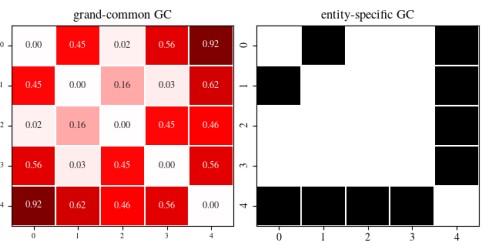

Figure 12: Springs5: $\bar{\mathbf{z}}$ and $\mathbf{z}^{[0]}$. $\mathbf{z}^{[0]}$ is binary (and symmetric) with entries generated according to Bernoulli distributions.

1959) with edge probability being 0.5, and (2) each node's trajectory is multivariate with 4 dimensions, that is, $x_{i,t} \in \mathbb{R}^d$, $d = 4$; the first 2 dimensions correspond to the velocity and the last 2 to the location in the 2-dimensional space.

The extension to the "multi-entity" case that is suitable for the setup considered in this paper is described next, and it differs primarily from the original one in how the Granger-causal connections across nodes are generated. Specifically, we start from $\bar{\mathbf{z}}$, whose entries $(i, j)$ in its upper-triangular part are generated independently from $\text{Beta}(1, 1)$; then set $\bar{\mathbf{z}}_{ji} \equiv \bar{\mathbf{z}}_{ij}, i < j$ so that it's symmetric. For the $\mathbf{z}^{[m]}$'s, let $\mathbf{z}^{[m]}_{ij} \sim \text{Ber}(\bar{\mathbf{z}}_{ij}), i < j$, and then set $\mathbf{z}^{[m]}_{ji} \equiv \mathbf{z}^{[m]}_{ij}, \forall\, m = 1, \cdots, M$. Once $\mathbf{z}^{[m]}$'s are generated, they dictate the connections between nodes in their respective systems, and one can proceed with the same procedure as in the original setting to simulate the trajectories. Note that (1) each entity's Granger-causal graph corresponds to a realization of a *heterogeneous* Erdős-Rényi graph; the edge probability differs across node pairs and depends on the corresponding entry in $\bar{\mathbf{z}}$ that is a realization from the Beta distribution, and (2) the grand common structure possesses a "probabilistic" interpretation, in that it effectively captures the *expectation* of an edge being present/absent across all entities. In this experiment, we set $p = 5$ and $M = 10$.

None of the competitors based on the prediction models can readily handle this setting[16], and therefore we only present results for those based on generative models. Note that in this experiment, despite that the underlying true graphs are symmetric, we do *not* use this information during our estimation.

Table 3 shows the results for the above-mentioned systems, using both the node- and the edge-centric decoders. A visualization of the estimates is provided in Figure 13. Overall, the proposed joint learning framework outperforms individual learning for entity-level graphs, while the performance is largely comparable for the common graph estimate. Given the physics system nature of this dataset (vis-a-vis time series signals), the edge-centric decoder has a small advantage over the node-centric one; this is manifested by the fact that under the joint learning framework, the two decoders show comparable performance, whereas the edge-centric decoder is clearly superior in the case of single-entity separate learning. Note that this points to another potential advantage of the joint-learning framework, in that it is more robust and exhibits less volatility than individual learning.

Table 3: Performance evaluation for the estimated $\bar{\mathbf{z}}$ (error in Frobenius norm) and $\mathbf{z}^{[m]}$'s (accuracy and F1 score after thresholding at 0.5, averaged across all entities) for the Springs5 system.

| quantity | metric | Multi-node | Multi-edge | One-node | One-edge |
|---|---|---|---|---|---|
| common | ERR-fnorm | 1.00(0.259) | 0.92(0.294) | 1.30(0.412) | 0.79(0.217) |
| entity(avg) | ACC% | 99.3(0.84) | 99.3(0.76) | 87.5(6.45) | 96.3(3.99) |
| entity(avg) | F1Score% | 99.5(0.79) | 99.4(0.73) | 88.2(7.45) | 96.3(4.78) |

---

[16]There are two issues that the prediction model-based competitors can not readily handle and would require major changes: (1) all of them assume that the Granger-causality to be estimated is numeric and therefore does not naturally handle the binary case, and (2) at any point in time, each node is assumed to have a scalar value, akin to classical time-series settings, whereas here each node is vector valued; consequently, the existing code does not readily handle it.

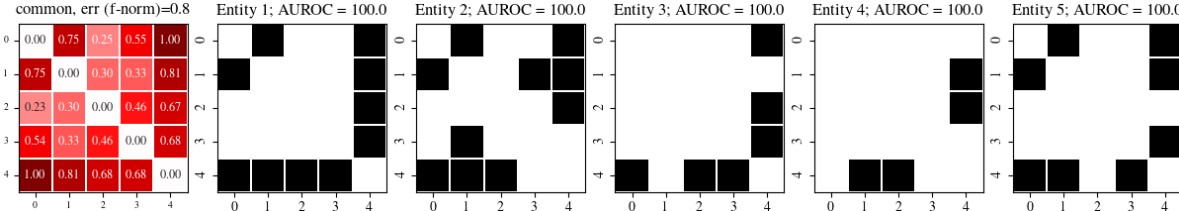

Figure 13: Estimated $\bar{\mathbf{z}}$ and $\mathbf{z}^{[m]}$'s (showing the first five) using the proposed framework with node-centric decoder (`Multi-node`).

## B.2 Additional performance evaluation results and their visualization

Table 4 presents additional evaluation metrics (TPR, TNR and ACC) for the proposed method and its strong competitors, after the estimates of the Granger causal graphs are thresholded at various levels no greater than 0.5 (after normalization). We only show the results for the estimated common graph $\bar{\mathbf{z}}$, since the results for the entity-level ones exhibit similar patterns.

As briefly mentioned in Section 4, prediction model-based methods (`NGC/GVAR`) are more sensitive to the value of the threshold, manifested by a sudden jump in accuracy once the threshold exceeds a certain level. On the other hand, the change in ACC for the ones based on generative models is more gradual. Given that in practice it is common to use a moderate threshold to eliminate small entries of the initial estimates of the Granger causal graphs to determine their skeleton, the above-mentioned susceptibility can adversely impact the quality of the final estimate used for interpretation purposes and in downstream analytical tasks.

Table 4: Performance evaluation for the support set of the estimated common graph $\bar{\mathbf{z}}$ at various threshold levels (left-most column). Numbers are in %, and correspond to the mean results based on 5 data replicates.

| | Multi-node | | | One-node | | | NGC-cMLP | | | GVAR | | | Linear | | |
|---|---|---|---|---|---|---|---|---|---|---|---|---|---|---|---|
| | TPR | TNR | ACC | TPR | TNR | ACC | TPR | TNR | ACC | TPR | TNR | ACC | TPR | TNR | ACC |
| **Linear VAR** | | | | | | | | | | | | | | | |
| 0.10 | 100 | 92.1 | 92.9 | 98.1 | 50.3 | 55.1 | 100 | 0.0 | 10.0 | 100 | 0.0 | 10.0 | 100 | 99.9 | 99.9 |
| 0.20 | 100 | 99.9 | 99.9 | 95.8 | 78.9 | 80.6 | 100 | 0.0 | 10.0 | 100 | 0.0 | 10.0 | 100 | 100 | 100 |
| 0.30 | 100 | 100 | 100 | 91.2 | 90.9 | 91.0 | 100 | 0.0 | 10.0 | 100 | 2.9 | 12.7 | 100 | 100 | 100 |
| 0.40 | 99.6 | 100 | 100 | 81.8 | 96.0 | 94.6 | 100 | 48.9 | 54.2 | 100 | 57.4 | 61.9 | 100 | 100 | 100 |
| 0.50 | 92.7 | 100 | 99.3 | 67.6 | 98.5 | 95.4 | 79.4 | 99.9 | 97.9 | 96.9 | 100 | 99.7 | 98.7 | 100 | 99.9 |
| **Non-linear VAR** | | | | | | | | | | | | | | | |
| 0.10 | 100 | 74.2 | 76.7 | 100 | 59.3 | 63.1 | 100 | 0.0 | 9.5 | 100 | 0.0 | 9.5 | 99.5 | 57.1 | 61.1 |
| 0.20 | 98.4 | 89.2 | 90.0 | 100 | 82.9 | 84.5 | 100 | 0.0 | 9.5 | 100 | 0.0 | 9.5 | 97.4 | 99.8 | 99.5 |
| 0.30 | 94.7 | 91.7 | 92.0 | 96.3 | 89.3 | 90.0 | 100 | 0.0 | 9.5 | 100 | 85.4 | 86.8 | 92.1 | 100 | 99.2 |
| 0.40 | 89.5 | 99.4 | 98.5 | 72.1 | 91.8 | 89.9 | 99.5 | 47.9 | 52.8 | 71.1 | 100 | 97.2 | 68.9 | 100 | 97.0 |
| 0.50 | 73.2 | 100 | 97.5 | 60.5 | 95.6 | 92.2 | 47.4 | 95.7 | 91.2 | 61.1 | 100 | 96.3 | 60.5 | 100 | 96.2 |
| **Lotka-Volterra** | | | | | | | | | | | | | | | |
| 0.05 | 100 | 72.8 | 76.8 | 99.0 | 40.5 | 49.3 | 100 | 58.4 | 64.7 | 34.0 | 100 | 90.1 | 33.3 | 100 | 90.0 |
| 0.10 | 100 | 97.4 | 97.8 | 96.3 | 73.9 | 77.3 | 99.7 | 100 | 100 | 33.3 | 100 | 90.0 | 33.3 | 100 | 90.0 |
| 0.15 | 99.3 | 99.8 | 99.8 | 90.0 | 92.4 | 92.0 | 90.7 | 100 | 98.6 | 33.3 | 100 | 90.0 | 33.3 | 100 | 90.0 |
| 0.30 | 67.0 | 100 | 95.0 | 50.3 | 100 | 92.5 | 33.7 | 100 | 90.0 | 33.3 | 100 | 90.0 | 33.3 | 100 | 90.0 |
| 0.50 | 33.3 | 100 | 90.0 | 33.3 | 100 | 90.0 | 33.3 | 100 | 90.0 | 33.3 | 100 | 90.0 | 33.3 | 100 | 90.0 |
| **Lorenz96** | | | | | | | | | | | | | | | |
| 0.05 | 95.2 | 99.5 | 98.7 | 93.8 | 100 | 98.8 | 100 | 0.0 | 20.0 | 100 | 99.8 | 99.8 | 95.8 | 94.1 | 94.5 |
| 0.10 | 58.8 | 100 | 91.8 | 39.5 | 100 | 87.9 | 100 | 0.0 | 20.0 | 96.8 | 100 | 97.0 | 50.0 | 100 | 90.0 |
| 0.15 | 27.2 | 100 | 85.5 | 25.0 | 100 | 85.0 | 100 | 0.0 | 20.0 | 72.8 | 100 | 94.5 | 25.0 | 100 | 85.0 |
| 0.30 | 25.0 | 100 | 85.0 | 25.0 | 100 | 85.0 | 100 | 79.2 | 83.4 | 25.0 | 100 | 85.0 | 25.0 | 100 | 85.0 |
| 0.50 | 25.0 | 100 | 85.0 | 25.0 | 100 | 85.0 | 93.0 | 93.4 | 93.3 | 25.0 | 100 | 85.0 | 25.0 | 100 | 85.0 |

An illustration of the recovered Granger-causal connections (after "optimal" thresholding) is shown in Figure 14. Note that `NGC` can only produce the "unsigned" version of the connections and hence all its estimates are shown as positive, whereas for other methods, the entries are "signed" with red denoting the positive and blue the negative ones.

One interesting observation is that for the Lotka-Volterra system, all methods have incorrectly estimated the signs of the diagonals, in that the underlying true dependencies on their own lags are positive for the preys and negative for the predators, whereas all methods fail to identify such discrepancy — although for

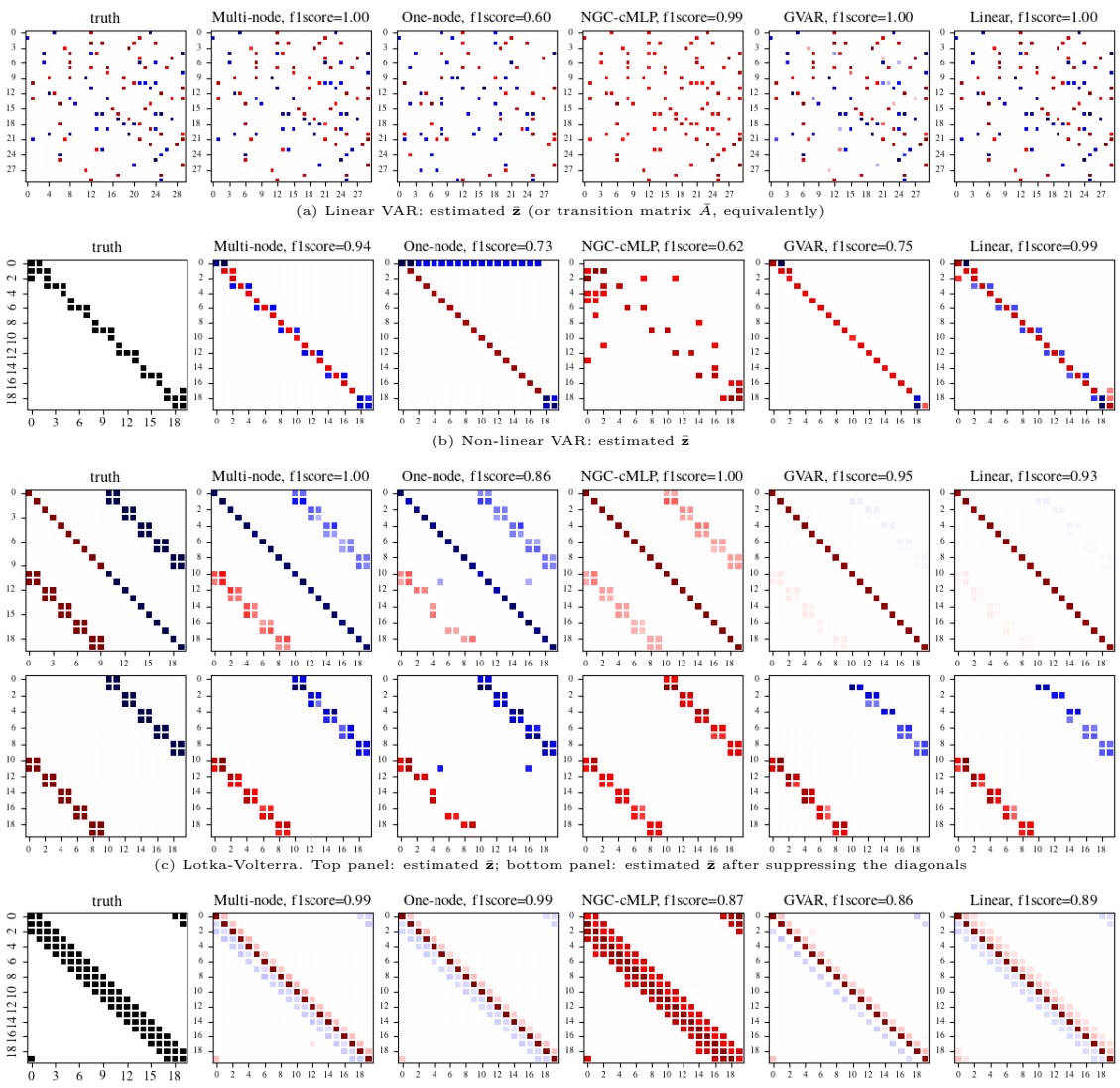

Figure 14: Estimated $\bar{\mathbf{z}}$ (after normalization) for various methods. The displayed f1score corresponds to the best attainable one (after thresholding) for each method. Red:(+); blue:(−). Note that NGC does not produce signed estimates and hence all its estimates are shown in red, with the shades corresponding to the magnitude of the entries after normalization.

the prediction model-based ones all dependencies show as positive and generative model-based ones have the opposite sign. This could be partially driven by the fact that during trajectory generation, the Runge–Kutta method (specifically, RK4) has been used and thus it renders the presence of a self-lag linear term with coefficient 1 in the recursion; in addition, a small noise term has also been injected.

For this setting, given that the estimated diagonals have dominating magnitude for GVAR and Linear, we also provide a visual display of the estimates with the diagonals suppressed.

*Remark* 5. A dichotomous behavior is observed between the unsigned and the signed estimates obtained from the code implementation of GVAR[17], with the former typically being 5-10% better (in absolute values, for reported metrics such as AUC, ACC that are between 0-100%). In all the tables, we have reported the performance of the superior one (unsigned), whereas Figure 14 is produced based on the signed estimate to show the positive/negative recovery. The best attainable F1 scores after thresholding (corresponding to the result of the specific data replicate being displayed) for these signed estimates are labeled in the title of

---

[17]Repository for GVAR: https://github.com/i6092467/GVAR

the figures; e.g., 0.75 for the non-linear VAR setting, 0.95 and 0.86 for the Lotka-Volterra and the Lorenz96 setting, respectively.

## B.3 The impact of the degree of heterogeneity

To evaluate the robustness and potential susceptibility of the proposed framework to the level of heterogeneity present across entities, we conduct additional experiments based on the Linear VAR and Non-linear VAR settings described in Section 4.1. To recap, the following dynamics are considered for each individual system of $p$ nodes, $\mathbf{x}_t = (x_{1,t}, \cdots, x_{p,t})^\top \in \mathbb{R}^p$:

- Linear VAR: $\mathbf{x}_t = A\mathbf{x}_{t-1} + \boldsymbol{\varepsilon}_t$. The Granger-causal graph $\mathbf{z}$ coincides with $A$.

- Non-linear VAR: each response coordinate $2 \leq i \leq (p-1)$ depends on the lag of its own that of two other coordinates indexed by $k_i^1$ and $k_i^3$, that is, $x_{i,t} = 0.25x_{k_i^2,t-1} + \sin(x_{k_i^1,t-1} \cdot x_{k_i^3,t-1}) + \cos(x_{k_i^1,t-1} + x_{k_i^3,t-1}) + \varepsilon_{i,t}$, with $k_i^1 < k_i^2 \equiv i < k_i^3$. The dynamics for the first and the last coordinates depend only on their respective adjacent coordinate, i.e., for $i = 1$, $x_{1,t} = 0.4x_{1,t-1} - 0.5x_{2,t-1} + \varepsilon_{1,t}$; for $i = p$, $x_{p,t} = 0.4x_{p,t-1} - 0.5x_{p-1,t-1} + \varepsilon_{p,t}$. The Granger-causal graph $\mathbf{z}$ dictates the exact locations of the $k_i^1$'s and $k_i^3$'s.

For both settings, the Granger-causal graphs $\mathbf{z}^{[m]}$'s of the $M$ entities are obtained by a "perturbation" with respect to the initial common Granger-causal graph $\bar{\mathbf{z}}^{(0)}$ (or $\bar{A}^{(0)}$ equivalently, in a linear setting), and the magnitude of such perturbation determines the degree of heterogeneity across entities and the final common graph $\bar{\mathbf{z}}$. The perturbation logic resembles the one described in Section 4.1.

Specifically, for the linear VAR setting, we let the skeleton of $\bar{A}^{(0)}$ have 30% density, that is, the support set $\mathcal{S}_{\bar{A}^{(0)}}$ is determined by independent draws from $\mathrm{Ber}(0.3)$, and the magnitude of the perturbation is controlled by the percentage of "relocated" entries. For the Non-Linear VAR setting, the magnitude of the perturbation is controlled by the number of rows whose off-diagonal entries are kept unchanged from those in the initial common Granger causal graph.[18] The sub-settings (S1-S5 with increasing degree of heterogeneity, respectively for linear and non-linear VAR setups) are depicted in Figures 15 and 16, where the percentage of relocation and the unchanged entries, respectively, are given in the sub-captions. Note that under the non-linear VAR setup, S1 and S5 correspond to the two extreme cases: no entity-level heterogeneity and almost fully heterogeneous.

We focus on generative model-based methods with a node-centric decoder, i.e., `Multi-node` (proposed framework) and `One-node`, and evaluate the performance of the estimates, obtained by training the model on different sample sizes. For the linear VAR setting, the sample size is set to 200 and 1000, while for the non-linear VAR setting to 500 and 2000. The selection of these sample sizes was based on the following three considerations: (1) non-linear dynamics are typically more challenging to learn and thus require larger networks and more samples to train; and (2) instead of choosing a "large" sample size where both methods perform well and thus little differentiation is shown, additional insights can be gained by assessing the performance of the model in settings where the available sample size is getting close to the information-theoretic limit (at the conceptual level).

Tables 5 and 6 display the results of the Linear/Non-linear VAR settings based on the same set of metrics as in Section 4.2, and they correspond to the average of 3 data replicates with the standard deviations displayed in parentheses. Major observations are: (1) for `Multi-node`, the estimation of $\bar{\mathbf{z}}$ is reasonably robust to the varying degree of heterogeneity across sub-settings. In particular, little deterioration is observed across sub-settings, although for sub-setting S5, given the very few common entries, the presented metrics become not not particularly meaningful. (2) Regarding the quality of individual entity estimates, `Multi-node` exhibits some deterioration in the non-linear setting when the model is getting close to being mis-specified (S5 versus S1-S4). (3) In the settings under consideration, where the sample size starts becoming rather small, `Multi-node` starts exhibiting an advantage over `One-node` by a wide margin. Specifically, for the estimated $\bar{\mathbf{z}}$, `One-node` shows performance degradation as the level of heterogeneity increases across sub-settings S1 to S5 (even for the linear case), and the overall performance is inferior to that of `Multi-node`. The latter is

---

[18]Recall, in the original settings presented in Section 4.1, for the linear VAR setting, the percentage of "relocated" entries is 10%; for the non-linear VAR setting, every 3rd row is left unchanged.

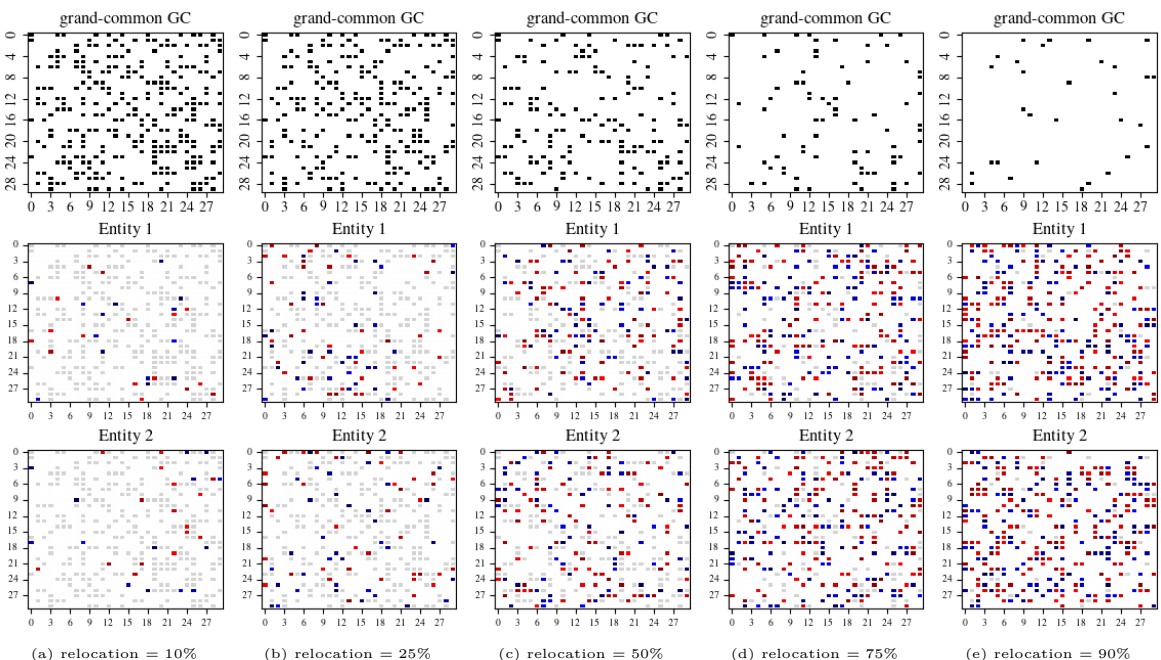

Figure 15: $30 \times 30$ Linear VAR system with a total number of $M = 20$ entities. Sub-settings are displayed vertically with increasing level of heterogeneity (from left to right). In the figure, only $\bar{\mathbf{z}}$ and $\mathbf{z}^{[1]}$, $\mathbf{z}^{[2]}$ are displayed.

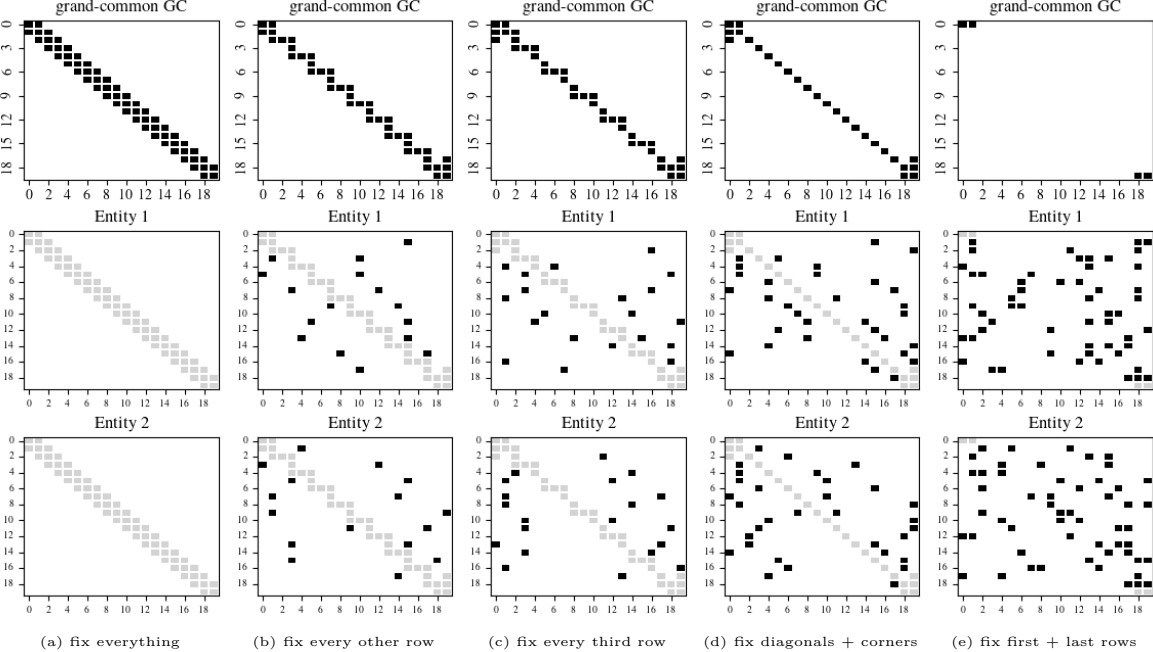

Figure 16: $20 \times 20$ Non-linear VAR system with a total number of $M = 10$ entities. Sub-settings are displayed vertically with increasing level of heterogeneity (from left to right). In the figure, only $\bar{\mathbf{z}}$ and $\mathbf{z}^{[1]}$, $\mathbf{z}^{[2]}$ are displayed. Similar to those in Section 4, as the non-linearity is induced via sinusoidal functions, we do not know the true sign of the cross lead-lag dependency; as such, the entries corresponding to entries that are present are colored in black.

somewhat expected: `Multi-node` performs joint estimation over samples across all entities and thus borrows information across; as such, it can rely on fewer number of samples[19] to attain estimates of similar accuracy.

---

[19]Here the number of samples is expressed in relative terms, that is, train size, corresponding to the number of trajectories used in training for each entity.

Table 5: Performance evaluation for the estimated $\bar{\mathbf{z}}$ and $\mathbf{z}^{[m]}$'s under settings S1-S5 of Linear VAR. Numbers are in %, and correspond to the mean results based on 5 data replicates; standard deviations are reported in the parentheses.

| | | Multi-node | | | | | One-node | | | | |
|---|---|---|---|---|---|---|---|---|---|---|---|
| | | S1 | S1 | S3 | S4 | S5 | S1 | S2 | S3 | S4 | S5 |
| **Linear VAR**; train size 200 | | | | | | | | | | | |
| common | AUROC | 100(0.0) | 100(0.0) | 100(0.0) | 100(0.0) | 100(0.0) | 90(2.1) | 90(0.6) | 90(4.7) | 83(4.3) | 85(1.7) |
| | AUPRC | 100(0.0) | 100(0.0) | 100(0.0) | 100(0.0) | 100(0.0) | 85(1.4) | 82(0.2) | 77(8.1) | 56(5.7) | 44(10.1) |
| | F1(best) | 100(0.3) | 100(0.0) | 100(0.2) | 100(0.0) | 100(0.0) | 75(2.1) | 72(1.3) | 70(8.1) | 53(6.1) | 43(10.0) |
| entity | AUROC | 94(1.4) | 94(1.2) | 94(1.4) | 95(1.6) | 95(1.6) | 88(3.1) | 89(2.4) | 89(3.1) | 88(3.4) | 88(2.6) |
| (avg) | AUPRC | 92(1.9) | 92(1.8) | 92(2.3) | 92(2.3) | 92(2.4) | 83(4.2) | 84(3.4) | 84(4.6) | 82(4.4) | 83(3.4) |
| | F1(best) | 84(2.4) | 84(2.2) | 84(2.9) | 84(2.7) | 84(2.9) | 75(4.1) | 76(3.0) | 75(4.1) | 74(4.1) | 74(3.2) |
| **Linear VAR**; train size 1000 | | | | | | | | | | | |
| common | AUROC | 100(0.0) | 100(0.0) | 100(0.0) | 100(0.0) | 100(0.0) | 97(1.3) | 96(0.6) | 95(1.3) | 91(1.2) | 93(0.6) |
| | AUPRC | 100(0.0) | 100(0.0) | 100(0.0) | 100(0.0) | 100(0.0) | 95(1.6) | 93(0.3) | 88(3.9) | 72(4.1) | 59(9.9) |
| | F1(best) | 100(0.4) | 100(0.0) | 100(0.0) | 100(0.0) | 100(0.0) | 89(1.7) | 85(0.5) | 80(6.5) | 67(3.0) | 57(8.6) |
| entity | AUROC | 95(1.4) | 95(1.3) | 95(1.5) | 95(1.7) | 95(1.6) | 94(1.4) | 94(1.4) | 94(1.5) | 94(1.6) | 94(1.6) |
| (avg) | AUPRC | 92(2.0) | 92(2.0) | 92(2.4) | 92(2.3) | 92(2.3) | 92(2.0) | 92(2.1) | 91(2.4) | 92(2.3) | 92(2.3) |
| | F1(best) | 84(2.9) | 84(2.4) | 84(2.9) | 85(2.6) | 85(2.8) | 84(2.7) | 84(2.5) | 84(3.1) | 84(2.7) | 84(2.9) |

Table 6: Performance evaluation for the estimated $\bar{\mathbf{z}}$ and $\mathbf{z}^{[m]}$'s under settings S1-S5 of Non-linear VAR. Numbers are in %, and correspond to the mean results based on 5 data replicates; standard deviations are reported in the parentheses.

| | | Multi-node | | | | | One-node | | | | |
|---|---|---|---|---|---|---|---|---|---|---|---|
| | | S1 | S1 | S3 | S4 | S5 | S1 | S2 | S3 | S4 | S5 |
| **Non-linear VAR**; train size 500 | | | | | | | | | | | |
| common | AUROC | 98(0.4) | 98(1.4) | 96(0.4) | 100(0.0) | 100(0.0) | 92(1.0) | 84(1.3) | 75(1.7) | 98(0.2) | 98(0.9) |
| | AUPRC | 81(1.4) | 84(10.9) | 73(3.2) | 98(0.4) | 100(0.0) | 69(0.1) | 64(0.6) | 49(9.0) | 89(0.6) | 46(31.5) |
| | F1(best) | 84(3.3) | 78(7.3) | 69(1.8) | 92(1.9) | 100(0.0) | 74(0.6) | 69(0.0) | 60(5.6) | 90(2.1) | 50(31.0) |
| entity | AUROC | 97(0.1) | 97(0.8) | 92(0.3) | 98(0.5) | 77(1.0) | 92(0.4) | 74(1.3) | 63(1.6) | 71(2.9) | 51(2.9) |
| (avg) | AUPRC | 79(0.4) | 82(6.0) | 67(2.1) | 88(1.9) | 41(1.9) | 68(0.6) | 54(0.2) | 39(3.4) | 52(2.6) | 18(0.9) |
| | F1(best) | 79(0.6) | 77(3.1) | 68(1.1) | 81(2.6) | 55(1.3) | 67(1.2) | 53(0.3) | 45(1.1) | 51(1.4) | 28(1.5) |
| **Non-linear VAR**; train size 2000 | | | | | | | | | | | |
| common | AUC | 99(0.1) | 100(0.2) | 98(0.3) | 100(0.0) | 100(0.0) | 95(0.1) | 95(0.0) | 95(0.4) | 99(0.2) | 100(0.0) |
| | AUPRC | 94(0.5) | 96(4.3) | 84(1.9) | 99(0.3) | 100(0.0) | 74(0.1) | 75(0.1) | 77(1.1) | 92(1.0) | 100(0.0) |
| | F1(best) | 95(0.0) | 96(1.7) | 77(1.1) | 96(1.9) | 100(0.0) | 77(0.0) | 69(0.0) | 72(2.2) | 92(0.0) | 100(0.0) |
| entity | AUROC | 99(0.1) | 99(0.2) | 95(0.6) | 99(0.1) | 80(0.6) | 95(0.3) | 93(0.2) | 90(0.3) | 93(0.5) | 73(1.1) |
| (avg) | AUPRC | 92(0.4) | 93(2.1) | 82(0.7) | 95(0.3) | 53(2.9) | 78(0.9) | 71(0.3) | 67(0.2) | 71(1.1) | 32(0.8) |
| | F1(best) | 87(0.6) | 88(0.9) | 76(1.2) | 89(0.9) | 61(1.5) | 76(0.4) | 70(1.0) | 63(0.4) | 64(1.2) | 47(0.5) |

Finally, note that in the proposed framework, at the decoder stage the Common2Entity step where the encoder distribution is merged via weighted conjugacy adjustment, the hyper-parameter $\omega$ controls the mixing percentage between the common and the entity-specific information. Conceptually, its choice varies according to the degree of heterogeneity present across entities: in the extreme case where the common structure is de facto absent, $\omega = 1$; the other end of the extreme corresponds to $\omega = 0$ when there is no heterogeneity. It is worth noting that we have observed empirically that the proposed framework is not sensitive to the choice of $\omega$, since in most cases its specific value makes little difference to the quality of the estimated $\bar{\mathbf{z}}$ and $\mathbf{z}^{[m]}$'s, as long as it was selected from a reasonable range (e.g., between $[0.25, 0.75]$). For example, in all the experiments above, we have fixed $\omega$ at 0.5.

## B.4 Some remarks on sample size

We give a brief account of the performance of the proposed framework in small sample size regimes. Note that in practical settings, model performance hinges on multiple factors, such as sample size, the size of the problem—including both the number of nodes and the number of entities, given the joint learning strategy—and how complex the temporal dynamics of the underlying systems are. The goal of this section is to provide guidance on the "minimum number of samples required"—from a practitioner's perspective—in settings of comparable size to the ones considered herein.

Specifically, we focus on the same set of time series settings for systems with non-linear dynamics considered in Section 4 and Appendix B.1, namely, the Non-Linear VAR, multi-species Lotka-Volterra, and the Lorenz96. In all three settings there are 20 nodes in their respective entity-level dynamical systems, and the collection

contains 5 or 10 entities. Recall that for the first setting the non-linear dynamics are induced through some sinusoidal function, while the other two settings are ODE-based systems.

Table 7 presents the performance evaluation of the estimated $\bar{\mathbf{z}}$ and $\mathbf{z}^{[m]}$'s based on `Multi-node`, when training sample sizes are 3000, 1000 and 500, respectively.

Table 7: Performance evaluation for $\widehat{\bar{\mathbf{z}}}$ and $\widehat{\mathbf{z}}^{[m]}$'s based on `Multi-node` under different settings with various training sample sizes. Numbers are in %, and correspond to the mean results based on 5 data replicates; standard deviations are reported in the parentheses.

| | | Non-linear VAR | | | Lotka-Volterra | | | Lorenz96 | | |
| | | 3000 | 1000 | 500 | 3000 | 1000 | 500 | 3000 | 1000 | 500 |
|---|---|---|---|---|---|---|---|---|---|---|
| common | AUROC | 98(0.1) | 97(0.2) | 96(0.4) | 100(0.0) | 97(2.2) | 90(8.0) | 99(0.4) | 95(1.3) | 89(3.5) |
| | AUPRC | 89(0.5) | 80(1.3) | 74(3.1) | 100(0.2) | 95(3.2) | 81(7.9) | 98(0.8) | 92(2.0) | 85(2.8) |
| | F1(best) | 79(1.4) | 75(1.3) | 69(1.7) | 100(0.4) | 94(4.3) | 78(4.6) | 94(0.7) | 87(2.3) | 84(1.9) |
| entity | AUROC | 96(0.5) | 94(0.6) | 92(0.7) | 88(0.8) | 81(2.1) | 64(1.4) | 92(1.5) | 87(1.3) | 85(1.3) |
| (avg) | AUPRC | 86(0.4) | 76(1.2) | 69(2.3) | 79(1.2) | 66(3.2) | 43(3.8) | 85(2.4) | 79(2.1) | 76(2.8) |
| | F1(best) | 78(0.3) | 73(1.4) | 69(1.6) | 75(1.1) | 64(4.0) | 42(2.4) | 78(2.3) | 73(2.5) | 70(3.6) |

The main observations are: (1) for the common graph $\bar{\mathbf{z}}$, as sample size reduces from 3000 to 1000, the proposed method's performance metrics stay above a reasonable range, even though a certain degradation is present, and its magnitude varies across settings. (2) For the entity-specific $\mathbf{z}^{[m]}$'s, the degradation in performance is more pronounced as the sample size reduces, and the model clearly suffers from not having access to an adequate number of samples.[20]

Based on these observations, we broadly conclude the following for practical settings of comparable size to the ones examined above: in the case where the primary focus is on the common graph $\bar{\mathbf{z}}$, the proposed framework would likely yield reasonable recovery even with about 1000 samples. On the other hand, if individual entity-level estimates are also of interest, sample sizes below 3000 would become rather challenging for the method to exhibit a satisfactory performance.

### B.5 Lotka-Volterra with perturbation: some characterization

We provide a characterization/justification for the "perturbed" Lotka-Volterra system, pertaining to how to validate a Lotka-Volterra system based on the "perturbed" interaction matrix being stable.

The general form of $p$-multi-species Lotka-Volterra equations are given by

$$\frac{\mathrm{d}x_i}{\mathrm{d}t} = r_i x_i \left(1 + \sum_{j=1}^{p} A_{ij} x_j\right), \tag{17}$$

where $r_i > 0$ is the *inherent per-capita growth rate* of species $x_i, i = 1, \cdots, p$ and $A \in \mathbb{R}^{p \times p}$ the species interaction matrix. The system considered in (13) can then be put in this canonical form, by assuming that the first $p/2$ species are preys and the last $p/2$ species predators.

Specifically, for the preys the corresponding equation in the canonical form becomes

$$\frac{\mathrm{d}x^i}{\mathrm{d}t} = \alpha x_i \left[\left(1 - \frac{1}{\eta^2} x_i\right) - \beta/\alpha \sum_{j \in \mathcal{P}_i^{\text{prey}}} x_j\right], \qquad i = 1, \cdots, p/2$$

where $r_i = \alpha$, $A_{ii} = -\frac{1}{\eta^2}$, $A_{ij} = -\beta/\alpha$ for all $j \in \mathcal{P}_i^{\text{prey}}$ otherwise 0; $\mathcal{P}_i^{\text{prey}}$ denotes the support set of the prey indexed by $i$. Analogously, for the predators the corresponding equation in the canonical form becomes

$$\frac{\mathrm{d}x_i}{\mathrm{d}t} = -\gamma x_i \left(1 - \delta/\gamma \sum_{j \in \mathcal{P}_i^{\text{predator}}} x_j\right), \qquad i = p/2 + 1, \cdots, p$$

---

[20]For these small sample size experiments, we use the same set of hyper-parameters as the ones in earlier experiments with much larger sample sizes (1e4). One can potentially expect improved performance with more carefully tuned hyper-parameters, although the improvement would likely be limited.

where $r_i = -\gamma$, $A_{ii} = 0$, $A_{ij} = -\delta/\gamma$ for all $j \in \mathcal{P}_i^{\text{predator}}$ otherwise 0; $\mathcal{P}_i^{\text{predator}}$ denotes the support set of the predator indexed by $i$.

It can be seen that fixed points of the set of equations in (17) can be found by setting $d\varkappa_i/dt = 0$ for all $i$, which translates to the vector equation

$$\mathbf{r} + A\varkappa = 0, \qquad \mathbf{r} \in \mathbb{R}^p, \varkappa \in \mathbb{R}^p, A \in \mathbb{R}^{p \times p}.$$

Consequently, fixed points exist if $A$ is invertible and are given by $\varkappa = -A^{-1}\mathbf{r}$. Note that $\varkappa_i = 0$ is a trivial fixed point. Further, the fixed point may contain both positive and negative values, which implies that there is no stable attractor for which the populations of all species are positive. The eigenvalues of $A$ determine the *stability* of the fixed point. By the stable manifold theorem, if its eigenvalues are less than 1, then the fixed point is stable. This can be easily verified once the "perturbed" Granger-causal matrix $\mathbf{z}$'s (which determines the $\mathcal{P}_i$'s and hence the corresponding $A$) are generated.

## C   Granger Causality and Graphical Models, Bayesian Hierarchical Modeling, and Linear VARs

This section comprises of three parts that provide background information on different topics mentioned in the main paper. Section C.1 illustrates how the framework of graphical models can be used to capture the concept of Granger causality. Section C.2 provides a brief overview of the Bayesian hierarchical modeling framework and outlines how it shares broad similarities to the modeling framework used in the paper. Finally, Section C.3 discusses possible ways of accomplishing the modeling task via a collection of linear VARs, either using a frequentist formulation, or a Bayesian hierarchical modeling one.

### C.1   Granger causality and graphical models

Consider a dynamical system, comprising of a $p$-dimensional stationary time series $\mathbf{x}_t := (x_{1,t}, \cdots, x_{p,t})$, with $x_{i,t}$ denoting the value of node $i$ at time $t$. Further, let $\mathbb{V} = \{x_1, \cdots, x_p\}$ denote the node set of the $p$ nodes/time series of the system.

A *Granger causal time series graph* (Dahlhaus & Eichler, 2003) has node set $V = \mathbb{V} \times \mathbb{Z}$ and edge set $E \subseteq V \times V$, wherein an edge $(x_i, t-s) \to (x_j, t) \notin E$, if and only if $s \leq 0$ or $x_{i,t-s} \perp\!\!\!\perp x_{j,t} \mid \mathcal{X}_t \setminus x_{i,t-s}$, where $\mathcal{X}_t = \{\mathbf{x}_{t'}, t' < t\}$ denotes the entire past process of the time series at time $t$, $\perp\!\!\!\perp$ probabilistic independence and $\setminus$ the set difference operator. The above definition implies that the edge set $E$ contains directed edges from past time points to present ones, only if $x_{i,t-s}$ and $x_{j,t}$ are dependent, *conditioned on* all other past nodes in $V$ excluding $x_{i,t-s}$.

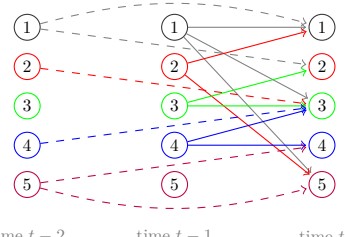

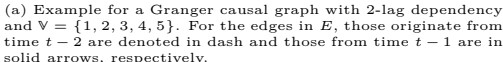

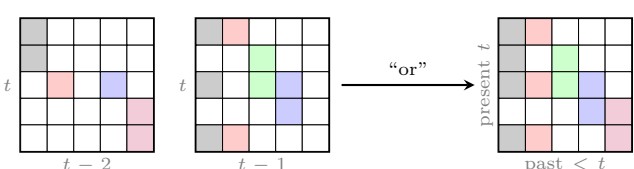

(a) Example for a Granger causal graph with 2-lag dependency and $\mathbb{V} = \{1, 2, 3, 4, 5\}$. For the edges in $E$, those originate from time $t - 2$ are denoted in dash and those from time $t - 1$ are in solid arrows, respectively.

(b) Connection matrices corresponding to the graph in Figure 17a, where columns correspond to emitters (past) and rows corresponding to receivers (present). Colored cells denote the presence of a connection. The right-most matrix corresponds to the aggregate Granger-causal connection matrix that summarizes and indicates present-past dependencies.

Figure 17: Pictorial illustration for Granger causal time series graph, aggregate Granger causal graph and their corresponding matrix representation.

An *aggregate Granger causal graph* (Dahlhaus & Eichler, 2003) has vertex set $\mathbb{V}$ and edge set $\mathcal{E}$, wherein an edge $(x_i \to x_j) \notin \mathcal{E}$ if and only if $(x_i, t-s) \to (x_j, t) \notin E$ for all $u > 0$, $t \in \mathbb{Z}$; i.e, *absence* of the edge $(x_i \to x_j)$ from the aggregate Granger causal graph implies *absence* of Granger causality from node (time

series) $x_i$ to node $x_j$, while *presence* of that edge implies that one or more time lags of node $x_i$ are Granger causal of node $x_j$.

Figure 17 illustrates pictorially both the Granger causal time series graph (Figure 17a), and the matrix representation (in the form of heatmaps) for the aggregate Granger causal graph (the right-most heatmap in Figure 17b).

In the case of a linear VAR system of order $q$ given by $\mathbf{x}_t = \sum_{k=1}^{q} A_k \mathbf{x}_{t-k} + \mathbf{e}_t$, the edge set of the Granger causal time series graph corresponds to $E = \{(A_k)_{ij} \,|\, (A_k)_{ij} \neq 0, \ i, j \in V, \ k = 1, \cdots, q\}$, while $\mathcal{E} = \{B_{ij} | B_{ij} = \mathbf{1}(\sum_{k=1}^{q}(\mathrm{abs}(A_k)_{ij}) \neq 0), \ i, j \in \mathbb{V}\}$, with $\mathbf{1}(\cdot)$ denoting the indicator function. The aggregate Granger causal graph with edge set $\mathcal{E}$ is an *unweighted* one, namely, its edges take values 0 (absence) or 1 (presence) and consequently reflect absence/presence of Granger causality between the time series.

*Remark* 6 (On the estimated Granger-causal graph). Under the proposed framework, in the binary case, the Granger connectivity graph corresponds exactly to the aggregate Granger causal graph defined above (see Section 3 and Remark 1), which is also in the same spirit as how different edge types are modeled in Kipf et al. (2018). In the continuous case, it corresponds to a *weighted* version of the aggregate Granger causal graph, wherein the weights correspond to the size of the "gate" through which the information from the past flows to the present. Admittedly, in the presence of non-linear modules (such as MLP) after the gating operation in the decoder, the weights no longer correspond to the "predictive strength" as defined in the original paper by Granger (1969). Nonetheless, at the conceptual level, the weights reflect the "strength" of the underlying relationships, as measured through the "permissible information flow".

## C.2 Bayesian hierarchical modeling

Given the prevalent usage of hierarchical modeling in the case where observational units form a hierarchy— e.g., in our motivating example, the observed time series are at the entity level and the entities form a group—we briefly review the Bayesian hierarchical modeling framework next.

Since its initial introduction in Lindley & Smith (1972) for linear models, the Bayesian hierarchical framework has been expanded and used for many other classes of statistical models. The book by Gelman et al. (2014) provides a description of the general framework and outlines the role of exchangeability for constructing prior distributions for statistical models with hierarchical structure. The framework has been operationalized and used for many statistical models, including regression and multilevel models (Gelman & Hill, 2006), time series (Berliner, 1996) and spatio-temporal models (Wikle et al., 1998), in causal analysis (Feller & Gelman, 2015), cluster analysis (Heller & Ghahramani, 2005), in nonparametric modeling (Teh & Jordan, 2010), and so forth. At the modeling level, the outline of the framework for a hierarchy comprising of two levels is as follows. Data for entities $m = 1, \cdots, M$ are generated according to some probability distribution

$$p(\varkappa^{[m]}; \theta^{[m]}, \phi) = p(\varkappa^{[m]} | \theta^{[m]}) \cdot p(\theta^{[m]} | \phi) \cdot p(\phi).$$

$\theta^{[m]}$'s are entity-specific parameters, and they are assumed to be generated exchangeably from a common population, whose distribution is governed by a common parameter $\phi$, and can be specified as $p(\theta^{[m]} | \phi)$. The common parameter $\phi$ can be fairly complex (for an example, see Section C.3) and possesses a prior distribution $p(\phi)$, which depending on the nature of $\phi$ can be fairly involved. The prior distribution for the parameter $(\theta^{[m]}, \phi)$ that governs the data generation mechanism for entity $m$ jointly, can then be characterized by $p(\theta^{[m]}, \phi) = p(\theta^{[m]} | \phi) p(\phi)$.

Note that the above specification exhibits differences to the generative process of a multi-level VAE presented in Section 2.2. Specifically, in the VAE specification, there are observed and latent random variables, modeled according to a probability distribution with *fixed* parameters $\theta^\star$, whereas in the Bayesian hierarchical modeling formulation, the parameters of the data generating distribution are *random* variables themselves and respect a hierarchical specification as previously mentioned.

## C.3 Modeling via a collection of linear VARs

We illustrate how the modeling task at hand can be handled when the dynamics are assumed linear. In particular, the dynamical systems can be characterized by a collection of linear VAR models; we show how

the common structure can be modeled by decomposing the transition matrix or using hierarchical modeling, respectively in a frequentist and a Bayesian setting. For ease of exposition, in the sequel, we assume the collection of linear VAR models have lag of order 1, and they are given by

$$\mathbf{x}_t^{[m]} = A^{[m]}\mathbf{x}_{t-1}^{[m]} + \varepsilon_t, \qquad m = 1, \cdots, M.$$

**Frequentist formulation.** Suppose that the transition matrices can be decomposed as $A^{[m]} = A_0 + B^{[m]}$, i.e., into a *common* component $A_0$ and an *entity-specific* one $B^{[m]}$. For model identifiability purposes, an "orthogonality" constraint is imposed; for example, in the form of $A_0 B^{[m]} = \mathbf{0} \in \mathbb{R}^{p \times p}$. In settings where the transition matrices $A^{[m]}$ are additionally assumed sparse (see, e.g., the numerical experiments in Section 4), such a constraint is typically in the form of $\text{support}(A_0) \cap \text{support}(B^{[m]}) = \emptyset$, namely that the matrices $A_0$ and $B^{[m]}$ do not share non-zero entries.

**Bayesian hierarchical modeling formulation.** We consider a collection of linear VAR models as above. The probability distribution of the data is $p\big(\{\mathbf{x}_t^{[m]}\}_{m=1}^M | \{A^{[m]}\}_{m=1}^M, \phi\big)$, where $\phi$ is a vector of additional parameters specified next. To construct the prior distribution of the model parameters $(\{A^{[m]}\}, \phi)$ we proceed as follows. Note that at the modeling level, a simple hierarchy is defined for each $(i,j)$-th element of the transition matrix across all $M$ entities/models; i.e., we consider $p^2$ such hierarchies *independently*. To use the Bayesian hierarchical modeling framework, let $\vec{c}_{ij} = (A_{ij}^{[m]}, \cdots, A_{ij}^{[M]})'$ be an $M$-dimensional vector containing the $(i,j)$-th element of all $M$ transition matrices. The following distributions are imposed on $\vec{c}_{ij}$'s and the parameters associated with their priors:

$$\vec{c}_{ij} \mid (\Psi, \tau_{ij}) \sim \mathcal{N}(0, \tau_{ij}\Psi); \tag{18}$$
$$\tau_{ij} \sim \text{Gamma}(M + 1/2, \lambda_{ij}),$$
$$\Psi \sim \text{Inverse Wishart}(S_0, \gamma_0).$$

The prior distributions in (18) are independent over index $(i,j)$; $\tau_{ij}$ is an $(i,j)$-element specific scaling factor, and $\Psi$ an $M \times M$ matrix that captures similarities between the $M$ models. The parameters $\lambda_{ij}, S_0, \gamma_0$ can be either fixed to some pre-specified values (e.g., a fixed $S_0$ can reflect prior knowledge on the *similarity* between the $M$ models), or equipped with diffuse prior distributions. Further, note that if $\Psi \equiv I$ the identity matrix, then the above specification reduces to the Bayesian group lasso of Kyung et al. (2010). Based on the above exposition, it can be seen that $\phi := (\Psi, \{\tau\}_{ij}, i, j = 1, \cdots, p)$. In summary, we have the following two-level modeling specification: at the first level, we have the data distribution, while at the second level the distribution on the elements of the transition matrices that are "coupled" across the $M$ models through $\phi$ and its prior distribution specification. Obviously, more complicated prior specifications can be imposed, for example by "coupling" whole rows of the transition matrices $A^{[m]}$ across the entities.

## D Additional Results for the EEG Dataset

The estimated common Granger-causal connections based on `One-node` and `NGC` are depicted in Figures 18 and 19, respectively. The increase in the overall Granger causal connectivity in the EC session compared to that in the EO session observed for `Multi-node` and `GVAR` is also present in the results from `One-node`, whereas it is reversed in the results of the `NCG`. Further, the observed increase in the overall connectivity pattern between the EO session compared to the EC session, exhibits differences between the left and right parts of the brain, something also observed in the results of `Multi-node`. Further, note that `NCG` does not produce signed estimates and hence all Granger causal connections are colored grey in Figure 19. This limitation of the method can hinder scientific insights that could be obtained from the analysis of a dataset by `NGC`.

## E On Respecting the Sign Distinction of the Connections

This section provides some explanation to how the proposed methodology (Multi-node)—modulo estimation error that can introduce inaccuracies—recovers the sign of the underlying truth *up to a complete sign flip*,

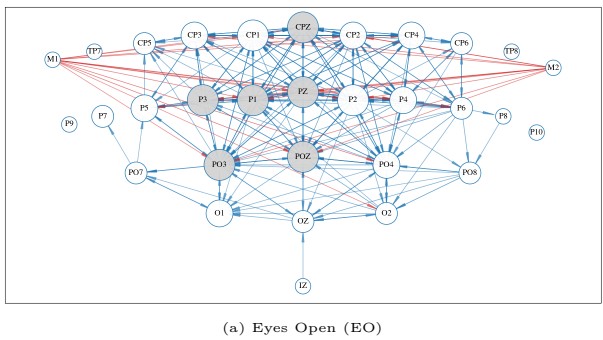 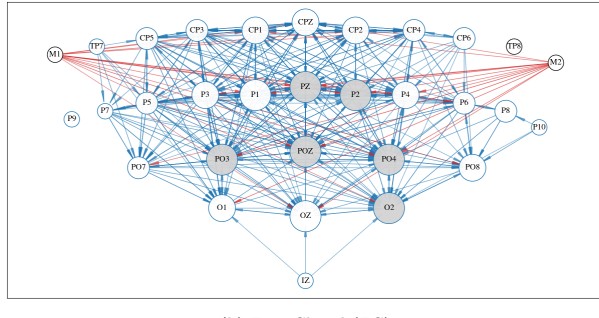

(a) Eyes Open (EO)               (b) Eyes Closed (EC)

Figure 18: `One-node` results: estimated **common** Granger-causal connections for EO (left panel) and EC (right panel) after normalization and subsequent thresholding at 0.50. Red edges correspond to positive connections and blue edges correspond to negative ones; the transparency of the edges is proportional to the strength of the connection. Larger node sizes correspond to **higher in-degree** (incoming connectivity), and the top 6 nodes are colored in gray.

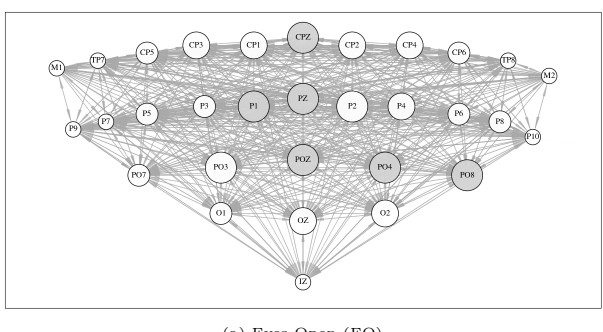 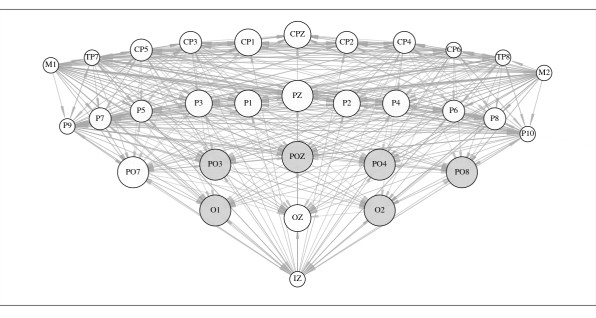

(a) Eyes Open (EO)               (b) Eyes Closed (EC)

Figure 19: `NGC` results: estimated **common** Granger-causal connections for EO (left panel) and EC (right panel) after normalization and subsequent thresholding at 0.45. All edges are colored gray, since `NGC` does not provide signed estimates of Granger causal connections. The transparency of the edges is proportional to the strength of the connection. Larger node sizes correspond to **higher in-degree** (incoming connectivity), and the top 6 nodes are colored in gray.

that is,

$$\text{SIGN}(\hat{\mathbf{z}}) = (\pm)\text{SIGN}(\mathbf{z}); \tag{19}$$

with $\text{SIGN}(\cdot)$ operating in an entry-wise fashion on $\mathbf{z}$ or $\hat{\mathbf{z}}$. In (19), $\mathbf{z}$ generically refers either to the grand-common Granger-causal graph $\bar{\mathbf{z}}$ or entity specific ones $\mathbf{z}^{[m]}$, and $\hat{\mathbf{z}}$ is the corresponding estimate. This is equivalent to saying that there is no guarantee that for each individual entry, $\text{sign}(z_{ij}) = \text{sign}(\hat{z}_{ij})$ always holds; however, all positive (negative) signed connections are identified as having the same sign. In this regard, the signs of the estimates obtained from the procedure can be interpreted in a meaningful way, in that the positive/negative connections can be *differentiated*; see Figure 20 for an illustration.

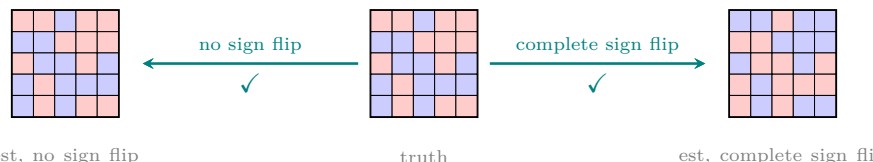

est, no sign flip         truth        est, complete sign flip

Figure 20: Pictorial illustration for the concept of "up to complete sign flip". In both the no-sign-flip and the complete-sign-flip case, the estimate always "groups" the positive/negative connections together in a way that is in accordance with the truth.

As a result of (19), the following also readily holds for any two entries indexed by $(i_1, j_1)$ and $(i_2, j_2)$, $\forall\, i_1, j_1, i_2, j_2 \in \{1, \cdots, p\}$:

$$\text{sign}(z_{i_1 j_1})\text{sign}(z_{i_2 j_2}) = \text{sign}(\hat{z}_{i_1 j_1})\text{sign}(\hat{z}_{i_2 j_2});$$

i.e., if two connections have the same/opposite signs in $\mathbf{z}$, they continue having the same/opposite signs in $\hat{\mathbf{z}}$. We shall refer to this property as "respecting the sign distinction".

The goal of this section is to provide some intuition on how the above mentioned is enabled through the encoder-decoder learning—in particular, in the presence of non-linear modules. Note that the subsequent arguments do not constitute a formal end-to-end proof.

In the sequel, we focus on the single-entity case and ignore modules related to the coupling between entity-level graphs and their grand-common counterpart, as these modules are not pertinent to this specific discussion. Concretely, the relevant modules in the ensuing discussion are:

- $q(\mathbf{z}^{[m]}|\varkappa^{[m]})$ as captured by (enc-a) and (enc-b) combined; i.e., the "Trajectory2Graph" encoder.
- $p(\varkappa^{[m]}|\mathbf{z}^{[m]})$ as captured by (dec-b); i.e., the "Graph2Trajectory" decoder.

The superscript $[m]$ will be omitted henceforth.

**Outline of the argument.** The argument consists of two parts:

1. The decoder, by utilizing a *shared* MLP across all response coordinates, ensures that the sign distinction is respected across the rows, along any column. Specifically, see, e.g., equations (9) and (10), wherein the MLP and the subsequent operations (in particular, their parameters) are shared by *all* response coordinates $i$'s.

2. The encoder, in the case of supervised training, disallows any *partial* (row or column) sign flip.

(1) and (2) jointly ensure that $\hat{\mathbf{z}}$ respects the sign of $\mathbf{z}$ up to a *complete* sign flip, and this is operationalized via the end-to-end training where the parameters are jointly learned and the data likelihood maximized.

At the high level, the shared MLP mechanism in the decoder ensures that it will not generate estimates that show "row sign flip" relative to the underlying truth. Specifically, for any fixed column, if one looks at the estimates along the columns (i.e., vertically) and across the rows, the estimates would respect their sign distinction in a pairwise fashion. However, it does not preclude cases where along the rows (i.e., horizontally) and across the columns, signs in the estimates can be flipped (i.e., column sign flip). On the other hand,

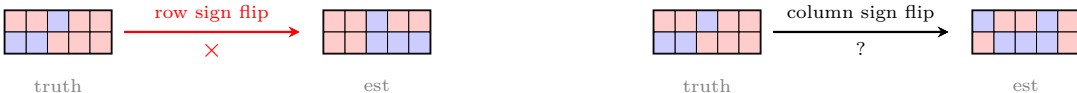

(a) Example for row sign-flip: signs of the 2nd row is flipped (right versus. left)

(b) Example for column sign-flips: signs of the 1st, 3rd and 4th columns are flipped (right versus. left)

Figure 21: Pictorial illustration for the concept of "row sign flip" and "column sign flip", picking the first two rows from Figure 20. Note that the former is prohibited by the shared MLP mechanism in the decoder construction.

to generate estimates that recover the sign of the underlying truth up to a complete sign flip, both row and column sign flips need to be precluded. The latter is facilitated by the encoder module during the end-to-end training: when the decoder fixes the "sign-orientation" vertically across the rows, the encoder would favor estimates that do not exhibit any partial sign flip during learning.

The details for each component are given next.

### E.1 Decoder

**Claim:** By using a shared MLP across all response coordinates, for any fixed column $j \in \{1, \cdots, p\}$, the decoder respects the sign distinction across the rows of $\mathbf{z}$, that is,

$$\text{sign}(\widehat{z}_{i_1 j})\text{sign}(\widehat{z}_{i_2 j}) \equiv \text{sign}(z_{i_1 j})\text{sign}(z_{i_2 j}), \qquad \forall i_1, i_2 \in \{1, \cdots, p\}. \tag{20}$$

The same cannot be guaranteed, however, if different MLPs are used for different response coordinates.

For illustration purposes, we focus on the case where the feature dimension is 1 (i.e., classical time series setting). Consider a simple two-layer MLP whose hidden layer has $h$ neurons. Let $f_{\text{MLP}} : \mathbb{R}^p \mapsto \mathbb{R}$ be represented as

$$f_{\text{MLP}}(\mathbf{u}) = W^{(2)}\sigma\big(W^{(1)}\mathbf{u} + b^{(1)}\big) + b^{(2)}, \qquad \mathbf{u} \in \mathbb{R}^p;$$

$W^{(1)} \in \mathbb{R}^{h \times p}, b^{(1)} \in \mathbb{R}^{h \times 1}, W^{(2)} \in \mathbb{R}^{1 \times h}, b^{(2)} \in \mathbb{R}$; $\sigma(\cdot)$ is some activation function. Specifically in the Graph2Trajectory decoder, the function input of the MLP is in the form of $\mathbf{u}_{i,t-1}$, whose $j$th coordinate is given by $x_{j,t-1} \circ z_{ij}$, assuming the absence of any numerical embedding (see, e.g., expressions in (9) with superscript $[m]$ dropped). To further simplify notation, we ignore subscript $t-1$, and let $\mathbf{y} = (y_1, \cdots, y_p) \in \mathbb{R}^p$ denote the time-$t$ target. Effectively, at decoding time, an approximation of the following form is considered for all timestamps:

$$
\begin{aligned}
y_i &\approx f_{\mathrm{MLP}}(\mathbf{u}_i), \qquad \forall\, i = 1, \cdots, p \\
&= W^{(2)} \sigma \left( \begin{bmatrix} W^{(1)}_{11} & W^{(1)}_{12} & \cdots & W^{(1)}_{1p} \\ \vdots & \vdots & \ddots & \vdots \\ W^{(1)}_{h1} & W^{(1)}_{h2} & \cdots & W^{(1)}_{hp} \end{bmatrix} \begin{bmatrix} x_1 \circ z_{i1} \\ \vdots \\ x_p \circ z_{ip} \end{bmatrix} + b^{(1)} \right) + b^{(2)},
\end{aligned}
\tag{21}
$$

where $x_1, \cdots, x_p$ are inputs directly available through training data, $(z_{i1}, \cdots, z_{ip})'$ constitutes the $i$th row of matrix $\mathbf{z}$. Crucially, $f_{\mathrm{MLP}}$ is shared across all $i$'s.

In the actual end-to-end learning, $z_{ij}$'s are sampled from a distribution whose parameters are dictated by the encoding step. The parameters of the encoders are jointly learned with those of the decoders, by minimizing the reconstruction error and the KL term. Here to further delineate the issue pertaining specifically to whether with the use of a shared MLP, the *learned* $z_{ij}$'s can respect the sign distinction, we ignore the encoding step, and simplifies the question as follows:

*Can the learning procedure—by minimizing the prediction error based on* (21)— *that jointly learns the $W$'s, $b$'s and entries of $\mathbf{z}$'s give rise to learned $\widehat{z}_{ij}$'s, such that the $\widehat{z}_{ij}$'s respect the sign distinction?*

The answer is affirmative for any fixed column $j = 1, \cdots, p$. To see this, expand the matrix product in (21), which gives (here we ignore approximation error and assume the model is well-specified):

$$
y_i = \sum_{s=1}^{h} W^{(2)}_s \sigma \left( \sum_{j=1}^{p} \left( W^{(1)}_{sj} \circ z_{ij} \right) x_j + b^{(1)} \right) + b^{(2)}.
$$

The predicted $\widehat{y}_i$ is given by

$$
\widehat{y}_i = \sum_{s=1}^{h} \widehat{W}^{(2)}_s \sigma \left( \sum_{j=1}^{p} \left( \widehat{W}^{(1)}_{sj} \circ \widehat{z}_{ij} \right) x_j + \widehat{b}^{(1)} \right) + \widehat{b}^{(2)},
$$

where $\widehat{W}^{(1)}, \widehat{W}^{(2)}, \widehat{b}^{(1)}$ and $\widehat{b}^{(2)}$ are estimated weights and bias terms. By minimizing the prediction error, $\widehat{y}_i$ is close to $y_i$, for *any* values of $x_1, x_2, \cdots, x_p$ and for all $i$'s. This amounts to having the estimated coefficients in front of the $x_j$'s sufficiently close to the truth—in particular, modulo estimation error, the following holds:

$$
W^{(1)}_{sj} z_{ij} = \widehat{W}^{(1)}_{sj} \widehat{z}_{ij}, \qquad \text{for all } i = 1, \cdots, p.
\tag{22}
$$

This further gives

$$
(W^{(1)}_{sj})^2 z_{i_2 j} z_{i_2 j} = (\widehat{W}^{(1)}_{sj})^2 \widehat{z}_{i_1 j} \widehat{z}_{i_2 j}, \qquad \forall\, i_1, i_2 \in \{1, \cdots, p\},
\tag{23}
$$

and therefore (20) follows since $(\widehat{W}^{(1)}_{sj})^2 > 0$.

Note that in the case where different MLPs are used for different response coordinates, (23) becomes $(W^{(i_1,1)}_{sj} W^{(i_2,1)}_{sj}) z_{i_2 j} z_{i_2 j} = (\widehat{W}^{(i_1,1)}_{sj} \widehat{W}^{(i_2,1)}_{sj}) \widehat{z}_{i_1 j} \widehat{z}_{i_2 j}$, which no longer leads to (20).

**Toy data experiments.** To verify this empirically, we consider a toy data example, where the trajectories are generated according to a 2-dimensional linear VAR system, that is,

$$
\mathbf{x}_t = A \mathbf{x}_{t-1} + \mathbf{e}_t, \qquad \text{where } A = \begin{bmatrix} 0.5 & -0.25 \\ -0.25 & 0.5 \end{bmatrix};
\tag{24}
$$

coordinates of $\mathbf{e}_t$ are drawn i.i.d. from $\mathcal{N}(0, 0.5)$. Note that given the linear setup, the transition matrix corresponds precisely to the true Granger-causal graph, and therefore $\mathbf{z} \equiv A$.

We run end-to-end training based on two configurations of the decoder:

(a) a single MLP shared across all response coordinates;

(b) separate MLPs for different response coordinates.

In both configurations, the MLPs are 2-layer ones with a hidden layer of dimension 64. The experiment is run over a single data replicate but repeated using 10 independent seeds.

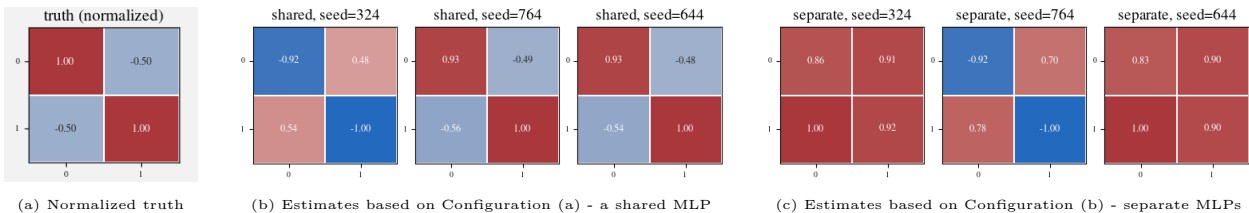

(a) Normalized truth     (b) Estimates based on Configuration (a) - a shared MLP     (c) Estimates based on Configuration (b) - separate MLPs

Figure 22: Toy data experiment decoder results: heatmaps for $\mathbf{z}$ (truth, normalized) and $\hat{\mathbf{z}}$ (estimates, normalized) under the shared and separate-MLP configurations. Panel (a) corresponds to $\mathbf{z}$ after normalization; panel (b) correspond to normalized $\hat{\mathbf{z}}$ (from runs with different seeds) obtained under Configuration (a); panel (c) correspond to normalized $\hat{\mathbf{z}}$ obtained under Configuration (b).

Figure 22 displays the estimated $\mathbf{z}$ corresponding to 3 different seeds for each configuration. Amongst all 10 runs, Configuration(a) preserves the sign distinction at all times—in this particular case, diagonals in $\hat{\mathbf{z}}$ always have the same sign and anti-diagonals have the opposite. Note that results from run seed 324 (left-most figure in Figure 22b) correspond to the case where the estimate yields a *complete* sign flip of the underlying truth. For Configuration (b), it fails in 2 out of the 10 runs—showing 2 failures (seed 324 and 644) and 1 success (seed 764) in Figure 22c, as the estimates can fail to preserve the sign distinction amongst the edges.

## E.2 Encoder

**Claim:** the encoder is able to perform "effective" learning based on labels up to a *complete* sign flip, but learning becomes problematic when the labels entail any *partial* sign flip.

Similar to the case of the decoder, to delineate the issue pertaining to the encoder, instead of considering end-to-end training where the two models are jointly learned, we consider a simplified setting, where we use the encoder module for a supervised learning task, based on data whose true generating mechanism is associated with the Granger causal graph $\mathbf{z}$. The question posed is the following:

*The true trajectories are generated based on $\mathbf{z}$. For a supervised learning task where the training labels are provided and the learning is enabled by the encoder module, is the encoder able to perform "effective" learning,*

1. *when the label used during training is some partial (column or row) sign flip of $\mathbf{z}$?*

2. *when the label used during training is a complete sign flip of $\mathbf{z}$, namely $-\mathbf{z}$?*

This is explored via synthetic data experiments, where the data generating mechanism is identical to the one considered in Section E.1.

Concretely, let $\mathbf{z}^\sharp$ denote the quantity that is provided as the target (label) during the supervised training; note that the data is generated according to (24), with $\mathbf{z} \equiv A = \begin{bmatrix} 0.5 & -0.25 \\ -0.25 & 0.5 \end{bmatrix}$, irrespective of the labels provided. The following four training scenarios are considered:

(a) No sign flip: $\mathbf{z}^\sharp = \begin{bmatrix} 0.5 & -0.25 \\ -0.25 & 0.5 \end{bmatrix}$, that is, $\mathbf{z}^\sharp = \mathbf{z}$;

(b) Complete sign flip: $\mathbf{z}^\sharp = \begin{bmatrix} -0.5 & 0.25 \\ 0.25 & -0.5 \end{bmatrix}$, that is, $\mathbf{z}^\sharp = -\mathbf{z}$;

(c) Column sign flip: $\mathbf{z}^\sharp = \begin{bmatrix} 0.5 & 0.25 \\ -0.25 & -0.5 \end{bmatrix}$, that is, $\mathbf{z}^\sharp_{:,1} = \mathbf{z}_{:,1}$, $\mathbf{z}^\sharp_{:,2} = -\mathbf{z}_{:,2}$;

(d) Row sign flip: $\mathbf{z}^{\sharp} = \begin{bmatrix} 0.5 & -0.25 \\ 0.25 & -0.5 \end{bmatrix}$, that is, $\mathbf{z}^{\sharp}_{1,:} = \mathbf{z}_{1,:}$, $\mathbf{z}^{\sharp}_{2,:} = -\mathbf{z}_{2,:}$.

We run encoder-only training for the above four scenarios. Results[21] are displayed in Figure 23, with the estimated $\mathbf{z}$ displayed in the top panel and the label $\mathbf{z}^{\sharp}$ used for supervision during training displayed in the bottom panel.

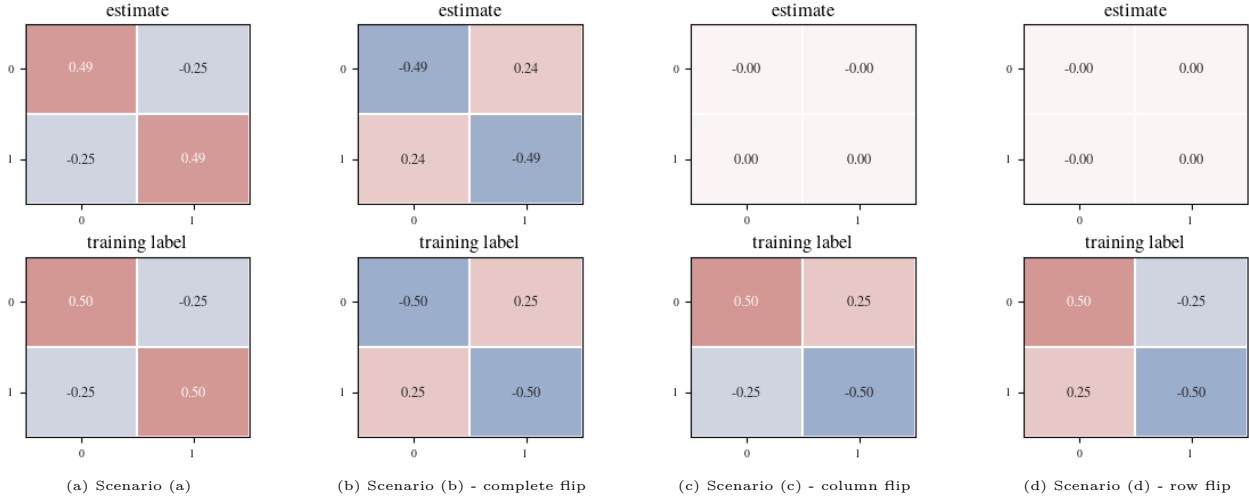

Figure 23: Toy data experiment encoder-only results: heatmaps for $\hat{\mathbf{z}}$ (estimates, top panel) and and $\mathbf{z}^{\sharp}$ (training label, bottom panel) for scenarios (a) to (d) respectively. Note that the underlying ground truth (i.e., the $\mathbf{z}$ that governs the dynamics of the trajectories) for all these experiments are identical to the one in Scenario (a).

As the results show, the encoder learns almost perfectly (relative to the provided labels) in scenarios (a) and (b), despite the latter being a complete sign flip. On the other hand, it struggles to learn in the case of partial sign flips (i.e., Scenarios (c) and (d)), as manifested by the essentially-zero estimated values. This empirically corroborates our claim.

Finally, it is worth noting that the claim examined in this subsection is under the supervised learning setup, namely, it establishes the fact that the encoder only permits no or complete sign flip, under a setting where the training target is explicitly provided. In practice, the learning is end-to-end, that is, there is no "real" supervision on the encoder available. As such, at the conceptual level, the learning relies on the decoder to fix the vertical sign-orientation as well as the encoder to preclude potential row sign flip—our experiment results in Section E.1 also corroborates this.

## F    Generalization to Multiple Levels of Grouping

We discuss the generalization of the proposed framework to the case where multiple levels of grouping are present and the corresponding group-common graphs at different levels of the hierarchy are of interest.

Consider $L$-levels of *nested* grouping where the group assignments become increasingly granular as the level index increases. Specifically, there is a single level-0 group that encompasses all entities, and $M$ (degenerate) level-$L$ groups, with each group $m$ having a singleton member being the entity $m$; all other levels are cases in between – see also Figure 24 for a pictorial illustration. Note that the case discussed in the main manuscript corresponds to the special case with $L = 1$. As an example for the case of $L = 2$ levels, consider the data analyzed in Section 5. Suppose that the subjects can be partitioned into 3 groups according to their ages — e.g., less than 30 years old, 30-60 years old, over 60. In such a setting, the single level-0 group comprises of all subjects; the level-1 groups correspond to subjects falling into different age strata; the level-2 groups are the subjects themselves. The quantities of interest are the connectivity patterns shared by subjects within their respective groups at all levels.

---

[21] Here we are displaying results for the test data; the results for training data lead to the same conclusion qualitatively.

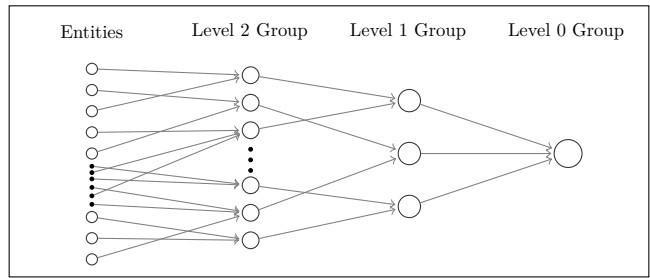

Figure 24: Diagram for a 3-level grouping. Neurons corresponds to $G_k^l$'s that collects the indices of the entities belonging to that group. Solid lines with arrows indicate how small groups from an upper level form larger groups at a lower level.

Let $\mathcal{G}^l := \{G_1^l, \cdots, G_{|\mathcal{G}^l|}^l\}$ denote the collection of groups of level $l$; each $G_k^l$ is the index set for the entities belonging to group $k$ at level $l$ and the group membership is non-overlapping, that is, $G_{k_1}^l \cap G_{k_2}^l = \emptyset, \forall\, k_1, k_2 \in \{1, \cdots, |\mathcal{G}^l|\}$. The quantities of interest are the entity-specific graphs $\mathbf{z}^{[m]}$, as well as the group-level common structure for all groups at all levels, that is $\bar{\mathbf{z}}^{G_k^l}$, denoting the group-common structure amongst all entities that belong to the $k$th group, with level-$l$ grouping; $l = 0, \cdots, L-1$ indexes the group level; $k = 1, \cdots, |\mathcal{G}_l|$ indexes the group id within each level. Finally, we let $\bar{\mathbf{z}} \equiv \bar{\mathbf{z}}^{G^0}$, which is consistent with its definition in the main text and it corresponds to the grand-common structure across all entities.

Without getting into the details of each step, the end-to-end learning procedure can be summarized in Figure 25. Compared with the two-level case, the generalization amounts to additional intermediate encoded/decoded distributions in the form of $q_\phi(\mathbf{z}^{[G_k^{l-1}]}|\mathbf{z}^{[G_k^l]})$, $p_\theta(\mathbf{z}^{[G_k^l]}|\mathbf{z}^{[G_k^{l-1}]})$ and $p_\theta(\mathbf{z}^{[G_k^l]}|\cdot)$ (post conjugacy adjustment/merging information); $l = 2, \cdots, L; k = 1, \cdots, |\mathcal{G}_l|$.

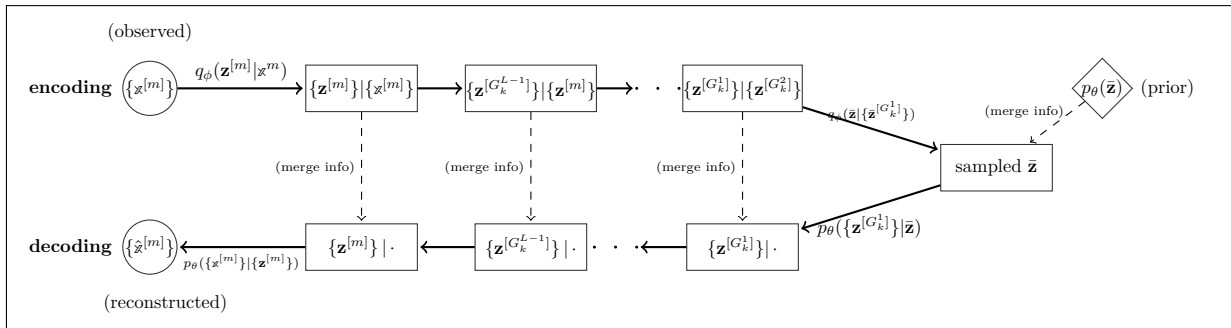

Figure 25: Diagram for the end-to-end encoding-decoding procedure in the presence of multiple levels of grouping.

