# OpenReview forum: "A VAE-based Framework for Learning Multi-Level Neural Granger-Causal Connectivity"
_TMLR — Accepted by TMLR_

### Review · Reviewer_8Gsn · 2023-12-13

**Summary Of Contributions:**

The paper presents a general framework for learning Granger causal structure from multiple related time series (entities) that share some common causal structure. The framework is based on a hierarchical model and uses a variational autoencoder (generative model) to jointly learn causal relations and shared causal structure between multiple entities. The framework is presented in general terms and detailed for two cases (Normal and Bernoulli causal structure). The primary contribution is the extension to multiple related entities, which allows learning/generalizing across entities with similar but not identical causal structure.

**Audience:**

Yes

**Claims And Evidence:**

Yes

**Requested Changes:**

Critical changes marked with *:

Abstract

Would it be possible to make it more explicit in the abstract what examples of applications could be?

1. Introduction

The introduction mentions expressing Granger causality through a graphical model. It might benefit the reader with a visual example.

The decomposition of transition matrices in the linear case could be explicated mathematically, to make the point more clear.

"...by postulating a hierarchical model..." again, could the be made more explicit, e.g. with an illustration, graphical model, or mathematical description?

2. Related work

Regarding the supervised approach, "...by aggregating information from the coefficients...": Could you include a bit more intuitive description for the motivation/idea behind this approach in contrast with the following section.

Could you explicate how the "stability-based procedure" is used to select the final connections in this line of work?

To make the description of the generative model-based approaches complete / self-contained consider including details/examples of how $p(x_{t+1}|<_t,\dots,x_1,z)$ would be defined. This might also be an opportunity to introduce notation that is needed later in the paper?

The mathematical definition of the ELBO is very general and does not give much information to the reader (no matter if the reader is familiar with variational inference or not). Consider writing a bit more explicitly how the encoder, decoder, and prior could be defined (this is related to the previous point).

The point that vector-valued nodes are problematic in the generative approach could be substantiated better. Why, and how can it be handled?

Does the reconstruction error term in eq. (1) depend on $z_1,\dots,\z_L$ or only $z_1$?

The purpose and implications of the "conjugacy adjustment" could be described in more detail (although I recognize that the level of detail is limited in the original paper).

* "---although it was not mentioned in the original paper..." I believe exactly this point is mentioned (section below eq. 19 in in the Sønderby paper) and is the main motivation provided.

3. The proposed framework

"...two layer ... multiple layers..." This is unclear

"...in the latter case, it can correspond to the values of node features." Unclear - what does it mean that node values correspond to node features?

The Granger-causal connection matrix could be more explicitly defined. In which sense does it define a graph - I assume a weighted graph, since it is mentioned that values are scalar: what do the weights correspond to? Similarly, "common substructure" is not clearly defined. (This is partially addressed later, e.g. in Remark 1 - I suggest addressing this earlier)

In sec. 3.2.1 the hidden representation h is referred to without being properly defined.

The message passing part of the encoder should be more clearly defined in the main text. It is difficult to understand without referring to Kipf et al. and should be more self contained in my view. How and on which graph does the GNN operate in this setting, when the graph is a latent variable?

At times it is difficult to assess how the different components that goes in the model are defined in practice. Just as an example, the function in Common2Entity $f_{\bar z\rightarrow z}^1(\cdot)$ is not defined anywhere in the main text or appendix as far as I can tell.

"Similar to the function ... and its choice can be rather simple." Could you explicate this, please. I suppose this could be chosen as the identity function? What do you use in practice? This is a good example of generality standing in the way of clarity.

The weight $\omega$ is introduced without much discussion. The only prior discussion was in the context of the ladder VAE where $\omega=.5$ was used motivated by a prior-posterior update. Is the proposed algorithm valid for any choice of the weight? By "toggling" do you mean discretely switching between 0 and 1, or varying the weight within the range? If the latter, I suggest using a different word, and if the former I suggest describing the motivation in more detail.

Consider making a graphical illustration of the Graph2Trajectory construction.

"...one needs to partition them to “short” ones of length T..." Why is this needed? Is this something you choose to do for computational efficiency reasons?

I do not understand the remark "...signed estimates and thus positive/negative Granger causal connections..." The $z^{[m]}_{ij}$ are vectors used to gate embeddings of the trajectories, and outputs are next processed by an MLP. How can the sign be interpreted in a meaningful manner as positive/negative connections?

* "The performance is overall on-par with GVAR..." Does GVAR not perform somewhat better overall?

**Strengths And Weaknesses:**

Strengths
- The proposed method is novel to my knowledge, builds on well established techniques that are combined in a meaningful manner.
- Mathematical details are for the most part rigorously described.
- The findings are supported empirically both through simulated data experiments and a EEG data set.

Weaknesses
- Readability could be significantly improved. Concepts are for the most part presented in general terms and subsequently made specific for different modeling choices. While this is in some sense a strength, it also makes the paper more difficult to follow that necessary. I suggest keeping the structure, but spending some effort on clarity when introducing concepts.
- Software implementation is not provided for the review.
- Performance of the method is not significantly better than existing methods, and it is not clearly analyzed or easy to understand why.
- The experiments on EEG only provide some qualitative evidence for the merits of the method. It would be interesting with a more detailed understanding of the results and a direct comparison with other methods.
- Are there any other real data sets/benchmarks that could be used for comparison with existing methods? My familiarity with the research topic and the current state of the art is not exhaustive, so I am unable to suggest any specific appropriate benchmarks for conducting accurate comparisons.

---

### Review · Reviewer_eJKM · 2024-01-02

**Summary Of Contributions:**

The major contribution of this manuscript is to look at the case where there are multiple realizations of a common shared graph, where each realization may have some differences in the underlying graph.  This is useful in cases such as identifying brain dynamics, where each subject has a related but distinct realization.  This is handled by using a hierarchical generation process in a multi-layer variational auto-encoder.

**Audience:**

Yes

**Broader Impact Concerns:**

None.

**Claims And Evidence:**

Yes

**Requested Changes:**

As mentioned above, a more complete description of the related work in hierarchical time series models and this work's relationship to them should be added.

I dislike the terminology used on the "supervised" methods.  They aren't really supervised as that is a standard approach for a time-series model.  Under that terminology, a linear VAE and the generative models would really also be supervised, whereas we usually limit supervision to mean some external label to the time series.  Please change the terminology, or add clarifications.

Fix neuroimaging terminology.

As mentioned above, I think that performing more experiments to characterize performance as a function of how different entities are and under model misspecification would better characterize the model.

**Strengths And Weaknesses:**

## Strengths

Overall, this idea makes sense in the context of relevant medical studies where there is a lot of subject-to-subject variability.

Provided results show that the proposed methods can better identify "grand common" graphs, which is useful in many scientific studies.

The overall idea is clear and well-described.

## Weaknesses

The related work lacks discussion of hierarchical processes for time series models.  As that is relatively well-studied in the Bayesian literature, albeit not with neural networks, and highly relevant, a significant discussion on the literature and this work's relationship to it should be added.

The experiments do not clearly demonstrate under what conditions the proposed methods will work better. I think that it would be useful to show how the performance varies as a the heterogeneity of the entities increase, as it varies from each entity having the same graph to being completely different.  That would allow much better characterization of when this may improve results.  Additionally, what happens when the model is misspecified or poorly specificed.

For section 5, it would be useful to highlight what differences in the results occur due to the new methodology, and discuss how that impacts scientific conclusions and robustness (e.g., do you get difference results?  Are the results more reliable/reproducible?).

## Minor Issues

As a minor thing, only a single entity is shown in the results.  I would suggest considering showing 2 entities to give a better sense of the heterogeneity.

In Figure 6, it is challenging to compare the inferred graphs to the true graphs as they are on separate pages and involve a lot of scrolling.  I'd suggest marking the true graph somehow on Figure 6.

Note that EEG is not neuroimaging.  Please correct the terminology.

---

### Review · Reviewer_Ti39 · 2024-01-20

**Summary Of Contributions:**

This paper proposes a new VAE-based framework for assessing Granger-causal connectivity among a collection of entities. Assuming that the entities have some common, underlying connectivity, with entity-specific differences, the method jointly learns both the common and entity-specific connectivity graphs. The paper evaluates its method using simulated data and provides a demonstration on a real dataset of multiple EEG subjects.

**Audience:**

Yes

**Broader Impact Concerns:**

No ethical concerns that would necessitate a broader impact statement.

**Claims And Evidence:**

Yes

**Requested Changes:**

I believe addressing each of the points in the weaknesses section, above, with justifications/explanations of a few sentences or a paragraph, as appropriate, would strengthen the paper.

**Strengths And Weaknesses:**

**Strengths:**
1. The exposition is largely clear; I particularly liked Section 2.1 (in the most recent edit), which draws a contrast between "predictive" and "generative" approaches.
2. The VAE-based framework is explained clearly and the various components are mostly well-described.
3. There are sufficient simulations in the main text and the appendix to convince the reader that the method works.
4. The real-data application shows that the method can be applied to a case with realistic network, subject, and sample sizes.
5. I thank the authors for additional details in the appendix, which answered several of my questions, e.g., accounting for multiple lags within this framework.
6. The main take-aways that follow the big results (tables/figures) are extremely helpful.

**Weaknesses:**
1. The notation in Section 2.2 was slightly confusing due to the reuse of $z$ to mean layer activations. Perhaps this could be changed to a different alphabet to reduce confusion.
2. If I understand correctly, the (enc-a) part of the encoder is actually the workhorse of the whole framework, which infers a graph from time-series measurements. I was slightly disappointed that this was relegated to the appendix; even there, it would help to have a few more details on what the embedding function $emb(x)$ is, and how precisely the two MLP functions are constructed.
3. In general, Granger causal frameworks are called "causal" because it suggests the past of node A is helpful for predicting the future of node B (in addition to the past of node B). Here, the decoder imposes this causal constraint, however the encoder does not - is there a reason for this choice? (e.g., does it not grant any extra benefit, or would it be hard to implement?)
4. The proposed method appears to involve fitting a large number of parameters. How many samples does this method need in order to provide reasonable estimates? Are sample sizes measured in terms of number of time points or number of trials (e.g., repeated presentations of the same stimulus in a neuroscientific context)? I understand that Table 5 (appendix B.3) talks a little bit about the effect of different sample sizes, but it would help to have a concise paragraph in the discussion section about this.

---

### Author Response · Authors · 2024-02-26
**Acknowledgements to the Action Editor and Reviewers**

We thank the Action Editor and three reviewers for their careful review of the manuscript, and for their constructive comments and suggestions.

We have uploaded the camera-ready version of the manuscript, and included the link to the code repository.

---

### Decision · Action_Editor_Rqe7 · 2024-02-20

**Recommendation:** Accept as is

**Comment:**

All 3 reviewers agree that the manuscript satisfies the standard and criteria of TMLR. The authors integrated all the comments from the reviews in the revision.

**Audience:**

Yes.

**Claims And Evidence:**

Yes.